# Air quality modeling intercomparison and multiscale ensemble chain for Latin America

**Jorge E. Pachón**[1], **Mariel A. Opazo**[2], **Pablo Lichtig**[3], **Nicolas Huneeus**[TS1][2], **Idir Bouarar**[4], **Guy Brasseur**[4], **Cathy W. Y. Li**[4], **Johannes Flemming**[5], **Laurent Menut**[6], **Camilo Menares**[2], **Laura Gallardo**[2], **Michael Gauss**[7], **Mikhael Sofiev**[8], **Rostilav Kouznetsov**[8], **Julia Palamarchuk**[8], **Andreas Uppstu**[8], **Laura Dawidowski**[3], **Nestor Y. Rojas**[9], **Maria de Fatima Andrade**[TS2][10], **Mario E. Gavidia-Calderón**[10], **Alejandro H. Delgado Peralta**[10], and **Daniel Schuch**[10,11]

[1]Department of Environmental Engineering, Universidad de La Salle, Bogotá, 111711, Colombia

[2]Department of Geophysics and Center for Climate and Resilience Research (CR2), Universidad de Chile, Santiago, 8320000, Chile

[3]Comisión Nacional de Energía Atómica – CNEA, Buenos Aires, C1429BNP, Argentina

[4]Max Planck Institute for Meteorology, 20146 Hamburg, Germany

[5]European Centre for Medium-Range Weather Forecasts – ECMWF, 53175 Bonn, Germany

[6]Laboratoire de Météorologie Dynamique, Palaiseau, 91128, France

[7]Norwegian Meteorological Institute, Oslo, 0313, Norway

[8]Finnish Meteorological Institute, Helsinki, 00560, Finland

[9]Department of Chemical and Environmental Engineering, Universidad Nacional de Colombia, Bogotá, 111321, Colombia

[10]Instituto de Astronomia, Geofísica e Ciências Atmosféricas, Universidade de São Paulo, São Paulo, 05508-09B, Brazil

[11]Civil and Environmental Engineering, Northeastern University, Boston, MA 02115, USA

**Correspondence:** Guy Brasseur (guy.brasseur@mpimet.mpg.de)

**Abstract.** A multiscale modeling ensemble chain has been assembled as a first step towards an air quality analysis and forecasting (AQF) system for Latin America. Two global and three regional models were tested and compared in retrospective mode over a shared domain (120–28° W, 60° S–30° N) for the months of January and July 2015. The objective of this experiment was to understand their performance and characterize their errors. Observations from local air quality monitoring networks in Colombia, Chile, Brazil, Mexico, Ecuador and Peru were used for model evaluation. The models generally agreed with observations in large cities such as Mexico City and São Paulo, whereas representing smaller urban areas, such as Bogotá and Santiago, was more challenging. For instance, in Santiago during wintertime, the simulations showed large discrepancies with observations. No single model demonstrated superior performance over others or among pollutants and sites available. In general, ozone and $NO_2$ exhibited the lowest bias and errors, especially in São Paulo and Mexico City. For $SO_2$, the bias and error were close to 200 %, except for Bogotá. The ensemble, created from the median value of all models, was evaluated as well. In some cases, the ensemble outperformed the individual models and mitigated extreme over- or underestimation. However, more research is needed before concluding that the ensemble is the path for an AQF system in Latin America. This study identified certain limitations in the models and global emission inventories, which should be addressed with the involvement and experience of local researchers.

## 1 Introduction

Latin America has some of the most populated urban areas in the world. Notably, Mexico City and São Paulo have populations exceeding 20 million, while Lima, Bogotá, Rio de Janeiro and Buenos Aires have more than 10 million inhab-

itants each (United Nations, 2018). These densely populated regions often experience air pollution events due to large emission sources and due to atmospheric conditions. Other major cities, such as Santiago and Medellín, with a population of ∼ 7 million and ∼ 3.5 million, respectively, are also affected by poor air quality. This urban air pollution not only has long-lasting effects on the health of the population but also has a significant negative impact on the environment and possibly the regional climate (Busch et al., 2023; Gouveia et al., 2018; Molina et al., 2015; Rodríguez-Villamizar et al., 2018; Romieu et al., 2012). Latin America could greatly benefit from an air quality analysis and forecasting (AQF) system that informs the public about air pollution episodes and supports policy actions.

To better understand the causes of air pollution events in Latin America, it is important to consider the local emission sources. In addition to the usual urban pollution sources (e.g., industrial facilities, residential heating, energy production and transportation sectors), plumes from biomass burning and long-range dust transport can occasionally reach major cities. In northern South America, increased pollution levels in the dry season have been associated with biomass burning (Ballesteros-González et al., 2020; Casallas et al., 2023; Mendez-Espinosa et al., 2019) and dust from the Sahara (Mendez-Espinosa et al., 2020). The latter source also affects the Caribbean and central Mexico in early spring (Kramer and Kirtman, 2021; Ramírez-Romero et al., 2021). Also, in the context of climate and land use change, wildfires are a recurrent phenomenon in southern South America (Resquin et al., 2018; de la Barrera et al., 2018; Sarricolea et al., 2020). The Amazon is the largest forest in the world and a significant source of biogenic volatile organic compounds (BVOCs), precursors of CO, ozone and secondary aerosols (Nascimento et al., 2022; Zimmerman et al., 1988).

Air quality management in Latin America and the Caribbean (LAC) CE1 has been traditionally focused on surveillance and building emission inventories (Franco et al., 2019). Modeling activities for LAC are less frequent than North America, Europe or Asia, mainly due to limited computing resources and scarce information of emission sources. Of more than 30 regional AQF systems identified worldwide, only one exists in Latin America (Zhang et al., 2012). In addition to the restrictions already mentioned, LAC has other challenges: complex terrain where cities are situated in the valleys and canyons of the Andes, varying meteorological conditions due to their proximity to mountains and coastlines, deep convection in the tropics, extensive biomass burning in the Orinoco and Amazon basins, and the presence of densely populated megacities and urban areas, among others. Despite limitations on applying air quality models in LAC, regional models have been successfully implemented since 2000.

The Coupled Chemistry Aerosol and Tracer Transport model to the Brazilian development of the Regional Atmospheric Modeling System (CCATT–BRAMS) was developed in the region (Longo et al., 2013) to investigate the impact of the Amazonian wildfires on air quality in major Brazilian cities (Pereira et al., 2011; Freitas et al., 2011). The North American Community Multiscale Air Quality (CMAQ) model, coupled with the Weather Research and Forecasting (WRF) meteorological model, has been used in Colombia and Brazil to predict pollutant concentrations and assess reduction strategies (Albuquerque et al., 2019; East et al., 2021; Pérez-Peña et al., 2017; Nedbor-Gross et al., 2018; Pachón et al., 2018). The WRF model coupled with chemistry (WRF–Chem) online has been actively used to study the impact of regional sources on air quality in urban centers across Colombia (Ballesteros-González et al., 2020, 2022; Casallas et al., 2024; González et al., 2018; Mendez-Espinosa et al., 2019), Chile (Saide et al., 2016) and São Paulo (Gavidia-Calderón et al., 2024). CHIMERE (Menut et al., 2013) and MATCH (Andersson et al., 2015) models have been applied in Chile to assess pollutant chemical transformation and dispersion as well as emission reduction strategies (Gallardo et al., 2002; Lapere, 2018; Lapere et al., 2021; Mailler et al., 2017). Additionally, CAMS reanalysis data have been compared against air quality observations, observing well-captured temporal trends for $PM_{10}$, $PM_{2.5}$ and $SO_2$ but not for $NO_x$ (Casallas et al., 2024).

This work conducts the first model intercomparison effort and ensemble construction for Latin America, which was assembled under the Prediction of Air Pollution in Latin America and the Caribbean (PAPILA) project (https://papila-h2020.eu/papila, last access: 14 August 2024). The aim of PAPILA was to develop an AQF system for the region with increasing capabilities in major cities. This objective is in line with the Global Air Quality Forecasting and Information System (GAFIS) initiative that supports the implementation of AQF systems, especially in countries and regions where they do not exist, such as Africa and South America (WMO, 2022). This article presents a retrospective (hindcast) analysis. Section 2 presents model descriptions, emission inventories utilized in the models and observations employed for model evaluation. In Sect. 3 we analyze the model performance and conduct intercomparisons for each pollutant ($NO_2$, $O_3$, CO, $SO_2$, $PM_{2.5}$). We also discuss the season variability of predictions and the analysis of large vs. small urban areas. Finally, Sect. 4 summarizes our findings and outlines directions for future development.

## 2 Methodology

The model intercomparison and construction of the ensemble required relevant activities: the execution of global and regional models in a common domain, harmonization of the model output, ensemble construction, collection of air quality observations, analysis of temporal and spatial variability, and model evaluation.

## 2.1 Description of the models and modeling setup

For the model intercomparison, two global models (CAMS and SILAM) and three regional models (CHIMERE, WRF–Chem, EMEP MSC-W) were selected based on the expertise of the research groups working on the PAPILA project (Table 1). WRF–Chem was implemented by two different groups, the Max Planck Institute for Meteorology (MPIM) in Germany and the University of São Paulo (USP) in Brazil, with different setups. It is worth noting that the early simulations analyzed hereby do not represent the best performance of each model in the LAC region or over individual urban areas. The different models are briefly described in the following paragraphs.

The Copernicus Atmosphere Monitoring Service (CAMS) provides state-of-the-art global atmospheric composition data based on the IFS (Integrated Forecasting System) model of the European Centre for Medium-Range Weather Forecasts (ECMWF) (Inness et al., 2019). The chemical mechanism of the IFS is an extended version of the Carbon Bond 2005 (CB05) and complements the MACC aerosol module (Flemming et al., 2017; Morcrette et al., 2009). The CAMS reanalysis data used for this project are a combination of satellite observations of atmospheric composition and the IFS modeling setup. Anthropogenic emissions from the MACC/CityZen (MACCity) inventory (Granier et al., 2011) and biomass-burning emissions from the Global Fire Assimilation System (GFAS) v1.2 (Kaiser et al., 2012) were used in the simulations (Table 1). The biogenic emissions were simulated offline by the Model of Emissions of Gases and Aerosols from Nature (MEGAN) version 2.1 (Guenther et al., 2006) using an offline emission inventory (ECCAD, 2021). CAMS has been extensively evaluated against ozonesondes, aircraft profiles, surface observations and global satellite retrievals (Flemming et al., 2015).

The System for Integrated Modeling of Atmospheric Composition (SILAM, http://silam.fmi.fi, last access: 14 August 2024) is a chemical transport model for global-to-local simulations of atmospheric composition and air quality that was developed at the Finish Meteorological Institute (FMI) (Sofiev, 2002; Kouznetsov and Sofiev, 2012; Sofiev et al., 2010, 2006, 2015). Briefly, SILAM employs the Carbon Bond Mechanism IV (CBM-IV) for gas-phase chemistry (Gery et al., 1989). For further details on the model characteristics, refer to METEO-FRANCE (2020). For this work, the SILAM simulations were driven by the meteorological IFS model of the ECMWF. Anthropogenic emissions were adopted from the CAMS global emission inventory v2.1, whereas biomass-burning emissions were generated by the Integrated Monitoring and Modelling System for Wildland Fires (IS4FIRES) (http://is4fires.fmi.fi, last access: 3 July 2024) (Sofiev et al., 2009; Soares and Sofiev, 2014). The biogenic emissions were simulated offline by MEGAN v2.1 (Guenther et al., 2006), particularly isoprene and monoterpene emissions computed for the year 2010, as found on the MEGAN website (Table 1). The model has been extensively evaluated in numerous international retrospective studies (Marécal et al., 2015; Kukkonen et al., 2012; Blechschmidt et al., 2020; Petersen et al., 2019) and real-time operational applications. SILAM is included in the regional European forecasting system provided by CAMS together with CHIMERE and EMEP MSC-W and eight other models (Colette et al., 2020).

CHIMERE is a Eulerian chemical transport model (CTM). It is able to perform simulations from urban to hemispheric scale (Lapere, 2018; Lapere et al., 2021; Mailler et al., 2017; Menut et al., 2021). The model can be used online (with WRF only) or offline (with several meteorological models). The model characteristics are published elsewhere (METEO-FRANCE, 2020). For this study, the meteorological forcing is the IFS global simulation provided by the ECMWF. The biogenic emissions are calculated online using MEGAN v2.1 (Guenther et al., 2006) using the 30 s horizontal-resolution database. Fire emissions are those of CAMS (Kaiser et al., 2012) and reformatted for CHIMERE using the dedicated preprocessor (Menut et al., 2021). The mineral dust is calculated online using the Alfaro and Gomes (2001) scheme, and the sea salt emissions are also calculated online using the Monahan (1986) scheme. $NO_x$ values by lightning are calculated using the scheme described in Menut et al. (2020). CHIMERE is used for analysis and forecast in tens of countries around the world and at various spatial scales, including the CAMS forecast. More specifically for Latin America, it was used for several studies about anthropogenic emissions, deposition of black carbon on snow, and indirect effects and impact of megafires on cloud formation (Lapere et al., 2021; Mailler et al., 2017; Lapere, 2018). For this exercise, CHIMERE was run for the 31 d of January and July 2015. However due to problems in the output files, 15 d of data were missing (5 d from 14 to 18 January and 10 d from 11 to 19 July and 9 July).

The EMEP MSC-W model ("EMEP model" hereafter) is an offline chemical transport model that was developed at the Norwegian Meteorological Institute (MET Norway). It is used to simulate photo-oxidants as well as organic and inorganic aerosols in scales ranging from local to global (Simpson et al., 2012). Details regarding the model characteristics can be found in METEO-FRANCE (2020). For this study the model was driven by meteorological data from the IFS model of the ECMWF. Gas-phase chemistry from the "EMEP scheme" comprises 70 species and 140 reactions (Andersson-Sköld and Simpson, 1999; Simpson et al., 2012), inorganics from the MARS equilibrium module (Binkowski and Shankar, 1995), and organics from the CBM-Z mechanism (Zaveri and Peters, 1999). Emissions from forest and vegetation fires are taken from the Fire INventory from NCAR (FINN v1.0) (Wiedinmyer et al., 2011). Biogenic emissions of isoprene and (if required) monoterpenes are calculated in the model for every grid cell (Simpson et al., 2012). The EMEP model has for several decades

**Table 1.** Description of the models included in the ensemble.

| Institution and model | Model type | Vertical resolution | Grid cells | Projection | Initial and Boundary conditions | Chemical mech. – Aerosol model | Meteorology | Emissions |
|---|---|---|---|---|---|---|---|---|
| ECMWF–CAMS | Global | 60 levels up to 0.1 hPa (65 km). Lowest level ~20 m | 450 × 460 | lat–long | IC for meteorology from the ECMWF's operational analysis and for chemistry from the previous forecast. | Chemistry: CB05 – Aerosol: AER bulk | C-IFS | CAMS-REG-AP_v3.1/2016 provided by CAMS_81. MACCity GFAS v1.2 Biogenic MEGAN 2.1 |
| FMI – SILAM | Global and regional offline v5.7 | 25 layers of varying depth up to 5.25 Pa. Lowest level ~20 m | 590 × 526 | lat–long | IC and BC from a global SILAM simulation with a coarser resolution. | Chemistry: Gas-phase CBM-IV | C-IFS ECMWF | CAMS-GLOB-ANT-v2.1 IS4FIRES v1.0. Biogenic MEGAN 2.1 |
| LMD & UCL – CHIMERE V2017r4.2 | Regional offline | Variable, 8 levels from the surface up to 500 hPa. 7 levels below 2 km | 450 × 455 | Lambert conformal | BC CAMS-Global IFS IC previous day forecast. | Chemistry: SAPRC 07 A – Aerosol: 10 bins and 15 species LMDz-INCA (gas, aerosols) and GOCART (mineral dust) | C-IFS | Anthropogenic EDGAR-HTAP Biogenic online MEGAN v2.1 |
| UCL – EMEP | Global and regional offline | 20 layers up to 100hPa with approximately 10 in the Planetary Boundary layer | 450 × 460 | lat–long | BC CAMS-Global IFS IC previous day forecast. | Chemistry: CBMZ – Aerosol: EMEP scheme MARS | C-IFS | Anthropogenic EDGAR-HTAP FINN v1.0 Biogenic Included with internal calculation of emissions |
| MPIM – WRF–Chem v3.6.1 | Regional online | 36 vertical levels up to 50hPa Lowest level ~50 m | 455 × 450 | Mercator | BC and IC for meteorology from GFS, for Chemistry from CAM-Chem. | Chemistry: MOZART 4 – Aerosol: GOCART | WRF | CAMS-GLOB-ANT v4.2 FINN v1.5 Biogenic MEGAN v2.1 |
| USP – WRF–Chem v3.9.1 | Regional online | 35 vertical levels up to 50hPa Lowest level ~50 m | 455 × 450 | Mercator | BC and IC for meteorology from GFS, for Chemistry from CAM-Chem. | Chemistry: MOZART – Aerosol: GOCART | WRF | CAMS-GLOB-ANT v5.3 FINN v1.5 Biogenic MEGAN v2.1 |

Abbreviations: FMI – Finnish Meteorological Institute, ECMWF – European Center for Weather and Modeling Forecast, LMD – Laboratoire de Météorologie Dynamique, MPIM – Max Planck Institute for Meteorology, UCL – University of Chile and USP – University of São Paulo, EDGAR – Emissions Database for Global Atmospheric Research, CBMZ – Carbon Bond Mechanism version Z.

been the main tool for underpinning air quality policies under the United Nations Economic Commission for Europe (UN-ECE) convention on long-range transboundary air pollution. However, it should be noted that the runs for this study were the very first EMEP model simulations ever conducted on a regional scale for LAC and should thus be considered only as a first demonstration of model capabilities. For PAPILA, the EMEP model was run by the modeling team at the University of Chile in Santiago with some support by MET Norway.

WRF–Chem is the Weather Research and Forecasting (WRF) model coupled with chemistry, developed at the National Center for Atmospheric Research (NCAR) with the purpose of simulating urban- to regional-scale fields of trace gases and particulates. The air quality and meteorological components share the same transport and physics scheme, as well as the same horizontal and vertical grids (Fast et al., 2006; Grell et al., 2005). The MPIM WRF–Chem uses version 3.6.1 to simulate meteorology and chemistry simultaneously online in South America at $\sim 20\,\text{km}$ horizontal resolution and 36 vertical levels extending from the surface to $\sim 21\,\text{km}$ altitude. The gas-phase chemistry is represented by version 4 of the Model for Ozone and Related Chemical Tracers (MOZART-4) chemical scheme (Emmons et al., 2010). The Goddard Chemistry Aerosol Radiation and Transport (GOCART) bulk aerosol module coupled with MOZART is used in this study to consider the aerosol processes (Chin et al., 2002; Ginoux et al., 2001). Boundary and initial conditions for the meteorology were set up from the Global Forecast System (GFS) and for the chemical species concentrations from CAM-Chem. The anthropogenic emissions were from CAMS-GLOB-ANT v4.2, which consists of $0.1° \times 0.1°$ grid maps of several species, including $CO$, $SO_2$, $NO$, non-methane volatile organic compounds (NMVOCs), $NH_3$, black carbon (BC) and organic carbon (OC) CE2. Daily varying emissions of trace species from biomass burning were taken from the FINN v1.5 dataset (Wiedinmyer et al., 2011). Biogenic emissions of trace species from terrestrial ecosystems are calculated online using MEGAN v2.04 (Guenther et al., 2006). Further details on the MPIM WRF–Chem model settings can be found in Bouarar et al. (2019).

WRF–Chem run by the USP (version 3.9.1) uses similar characteristics to those previously described with a horizontal resolution $\sim 22\,\text{km}$ and 35 vertical layers. Some differences from the MPIM configuration are the version of global emissions of CAMS-GLOB-ANT v5.3 (ECCAD, 2020), the speciation of the chemical boundary condition from the CAM-Chem model (Buchholz et al., 2019; Emmons et al., 2010) and the speciation of FINN v1.5 emissions, which are suitable for simulation over São Paulo. For this exercise, WRF–Chem did not include Mexico City in the modeling domain.

CHIMERE, IFS, EMEP, WRF–Chem, LOTOS-EUROS and SILAM models are used in an ensemble mode to configure the MarcoPolo–Panda prediction system in Asia (Brasseur et al., 2019; Petersen et al., 2019). It has been ob-

served that, under specific circumstances, a model ensemble can outperform individual models, demonstrating the potential benefits of this approach. With the desire to replicate the experience in Latin America, the selected models were applied in a common domain, defined by the southeastern corner at 119°54′ W, 59°54′ S and the northeastern corner at 28°6′ W, 29°54′ N. The models were run at a spatial resolution of $\sim 0.2° \times 0.2°$ ($\sim 20 \times 20\,\text{km}$). Input meteorology and emissions were up to the modeling group (Table 1). The simulation period covers January (Southern Hemisphere summer) and July (Southern Hemisphere winter) of 2015. The modeling data are CE3 available in a public repository (Pachón et al., 2024).

## 2.2 Model evaluation

The performance of the models was assessed by comparing the simulated concentrations with the average of the observations for each available city, pollutant and considered period. The observation's average was constructed by computing the arithmetic mean of all air quality stations available in the network within the city's polygon. On the other hand, the simulated concentrations for the models were estimated as the average of the models' closest grid point to the location of each station that is within the city's polygon for every city and pollutant considered in this study. This results in a weighted average of the model where the weight is given by the number of stations that measure the pollutant closest to each grid point, resulting in the same geographical sampling for the observations and the models, thus reducing any potential station's sampling bias to the best of our abilities. This approach was chosen with the objective of assessing the model performance in cities rather than for each air quality station separately. It is outside the scope of this work to conduct an intra-urban variability study of the model performance given the chosen resolution of $0.2° \times 0.2°$. The model evaluation was focused on nitrogen dioxide ($NO_2$), ozone ($O_3$), carbon monoxide (CO), sulfur dioxide ($SO_2$), and particulate matter less than 2.5 µm ($PM_{2.5}$) and less than 10 µm ($PM_{10}$).

For each period, pollutant and city, the model evaluation included the following metrics: model / observations ratio, mean bias (BIAS), modified normalized bias (MNBIAS), root mean square error (RMSE), fractional gross error (FGE) and correlation coefficient ($R$). The formulas were replicated from the MarcoPolo–Panda project (Petersen et al., 2019) and are presented in Table A1 in Appendix A. These evaluation metrics were computed for all models and the ensemble using the Modelling and Observation System and Analysis Tool, CE4 MOSPAT (Huneeus and Opazo, 2024).

## 2.3 Air quality monitoring networks in Latin America

Several air quality monitoring networks (AQMNs) are available throughout Latin America, especially in major cities. However, worldwide access to the datasets can be difficult

due to language barriers and the lack of a centralized platform. A comprehensive list of AQMNs in Latin America was assembled for the PAPILA project (https://papila-h2020.eu/observations, last access: 14 August 2024). For the year 2015, we collected air quality data for 12 cities in Mexico, Colombia, Ecuador, Peru, Chile, Brazil and Uruguay. Only stations with a minimum of 75 % data completeness were considered when calculating the city average of the observations, resulting in eight cities with enough data to use for this study. This data completeness requirement considers a minimum of 75 % of days available for each period, as well as a minimum of 75 % of hourly data to construct their daily average. We focus in this study on the four major cities (from north to south) Mexico City, Bogotá, São Paulo and Santiago (Fig. 1). However, data of all available cities were used in the model evaluation (Tables B1 through B8 in Appendix B).

## 3   Results

Simulated concentrations of all pollutants from all models were compared against observations from every city and for both periods (January and July) in 2015. In this section, we present results from the model evaluation, the spatial and temporal variability of simulated fields, and the impact of large versus small urban areas in the model intercomparison.

### 3.1   Model evaluation

The following results are presented for every pollutant: analysis of observations from AQMNs, simulated concentrations by the models, comparison of evaluation metrics and discussion of model performance, including the ensemble and analysis of model variation.

### 3.1.1   Nitrogen dioxide – $NO_2$

**Observations**

The number of stations per city recording $NO_2$ during January and July 2015 varies between 7 in Bogotá and 24 in Mexico City (Appendix B). The highest daily average concentration of $NO_2$ is observed in Santiago during winter at around 30 ppb (Fig. 2). This can be attributed to adverse meteorological conditions and emissions from transportation and residential combustion in the surrounding municipalities (Mazzeo et al., 2018; Saide et al., 2016), whereas in the summer $NO_2$ levels fall to 11 ppb. The second largest values are shown in Mexico City and São Paulo with daily average $NO_2$ levels of 27 and 20 ppb, respectively, due to the heavy use of fossil fuels in transportation and power generation. The lowest levels of $NO_2$ are measured in Bogotá with 16.4 ppb on average.

**Model performance**

In Bogotá and Santiago, $NO_2$ is underestimated by the ensemble members (Fig. 2). In Santiago, the mean of the models is 10.3 ppb in summer and 22.1 ppb in winter, lower than the mean of the observations. Similarly, in Bogotá the mean of the modeled values is 6.6 ppb, much lower than observations. In contrast, in São Paulo and Mexico City, the models both over- and underpredict the ambient concentrations, and the averages of the modeled fields (23.6 and 30.3 ppb, respectively) are on the same order of magnitude as the observations.

São Paulo and Mexico City exhibit the lowest MNBIAS and FGE for $NO_2$ (Table A2). The correlation between the models and observations hovers around 0.7, which is larger than the goal benchmark proposed for this pollutant ($R \geq$ 0.6) (Zhai et al., 2024).

In Santiago, the MNBIAS is mostly negative during both seasons except in the SILAM and EMEP models, which resulted in a positive bias. The degree to which the models underestimate the observations is notably higher in winter than in summer and with a larger FGE (Table A2). The correlation between models and observations in Santiago is larger in summer than in winter, with some models achieving the criteria benchmark ($R > 0.5$) (Zhai et al., 2024). In Bogotá, the MNBIAS values are large and consistently negative, and the FGE varies between 50 % and 156 % (Table A2). Despite these lower scores, the correlation between observations and models is moderate, around 0.6 in January, meeting criteria benchmarks and demonstrating that certain models can successfully replicate the temporal variations but not the magnitude of the pollutant.

The adequate performance in São Paulo and Mexico City may be attributed to an accurate portrayal of the temporal and spatial variability that is achieved in large urban areas like these ($> 3500\,km^2$), which encompass at least nine model cells ($20\,km \times 20\,km$). The lower simulated $NO_2$ levels in Bogotá likely stems from an underestimation of emissions. A study by Rojas et al. (2023) utilized local data to estimate on-road emissions in Colombia and revealed substantial underestimation of $NO_x$ emissions by global inventories such as EDGAR 6.1, CAMS and the Community Emissions Data System (CEDS). Their findings recommend adjustments to the emission factors used for $NO_x$, particularly for heavy-duty and passenger vehicles, followed by a recalculation of the resulting emissions. The underestimation of $NO_2$ can also be noted in other cities such as Medellín, Guadalajara, Lima and Quito (Fig. 8). These cities, along with Bogotá, possess urban areas ranging from 235 to $890\,km^2$ and are confined within one or two cells of the models ($20\,km \times 20\,km$). It is possible that the average of observations is heavily influenced by local sources, in which case a finer modeling resolution is required to accurately capture the spatial variability of air pollution.

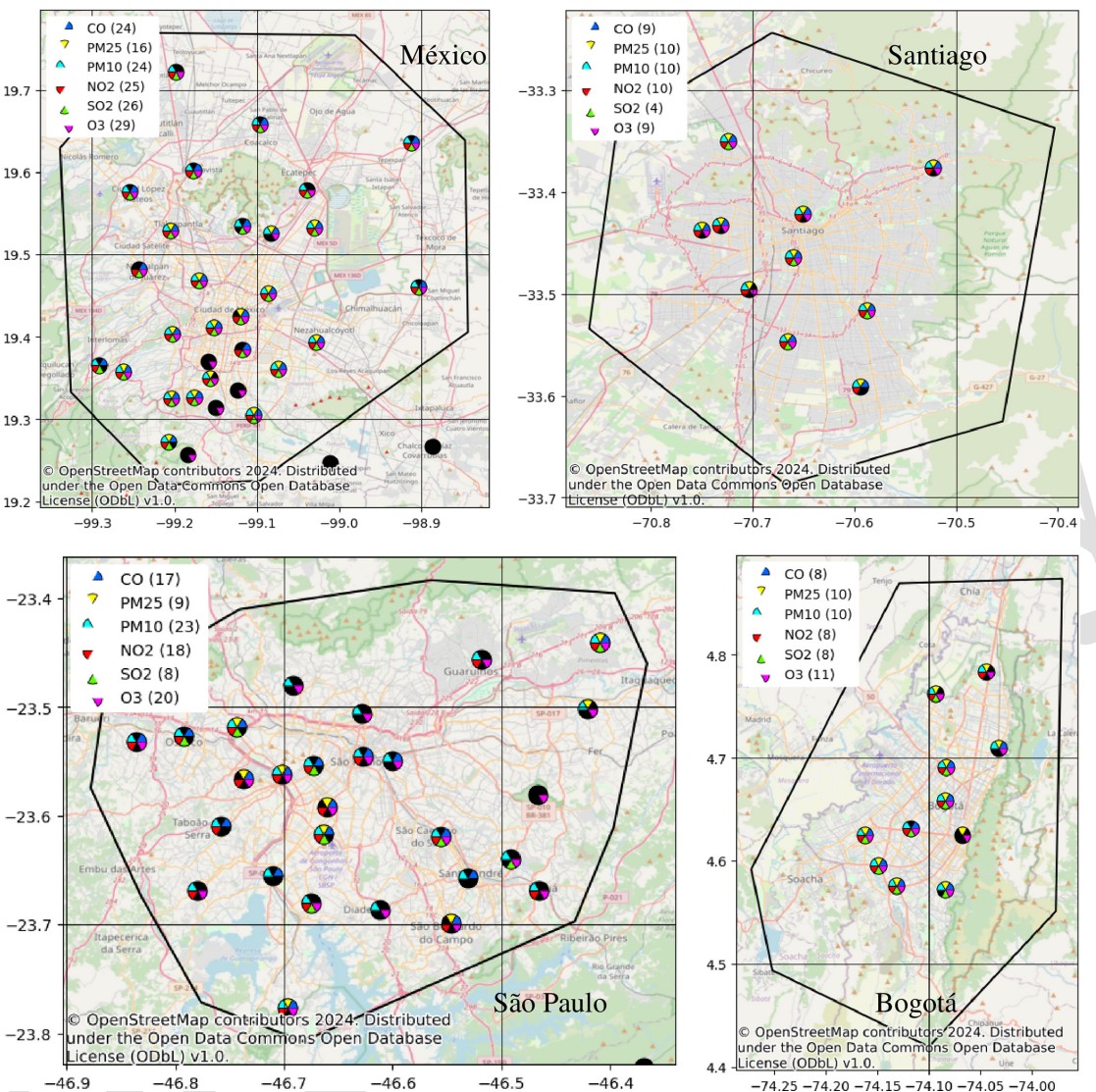

**Figure 1.** Location of air quality stations in major Latin American cities (Santiago, Bogotá, Mexico City, São Paulo) alongside the city's definition for computing the modeled city average. © OpenStreetMap contributors 2024. Distributed under the Open Data Commons Open Database License (ODbL) v1.0.

## Model intercomparison

For $NO_2$, CAMS underestimates the observations in the four cities, whereas SILAM underestimates this pollutant in Bogotá, Mexico City and São Paulo (only in July) and overestimates the observations in Santiago and in São Paulo (in January). CAMS displayed larger MNBIAS and FGE than SILAM. In general, SILAM reproduces at least 80 % of the $NO_2$ levels, with the exception of Bogotá, where only 30 % of the $NO_2$ levels are simulated. The correlation coefficient is better for SILAM ($R \sim 0.6$) than for CAMS ($R \sim 0.3$).

The results from regional models are very diverse. In general, WRF–MPI, CHIMERE and EMEP have lower values of MNBIAS and FGE for $NO_2$ in São Paulo and Mexico City (Table A2). In São Paulo, except for WRF–USP, regional models tend to overestimate $NO_2$ with a MNBIAS between 20 % and 70 %. WRF–USP reproduces about 76 % of $NO_2$ concentrations. In Mexico City, the tendency of regional models is to overestimate the $NO_2$ levels (MNBIAS: 10 % to 75 %). In Santiago, CHIMERE achieves the lowest MNBIAS ($-2$ %) in January but not in July ($-119$ %). In Bogotá, the MNBIAS in regional models remains consistently negative.

In Fig. 2, the model variation is visible. In Santiago in winter the range of $NO_2$ values is 48 ppb, which corresponds to a coefficient of variation (CV) of 71 % (Table A8); this contrasts with the range in summer of 15 ppb (CV = 49 %). Other large variations are observed in Mexico City in July (range 54 ppb, CV 57 %) and São Paulo (range 32 ppb, CV

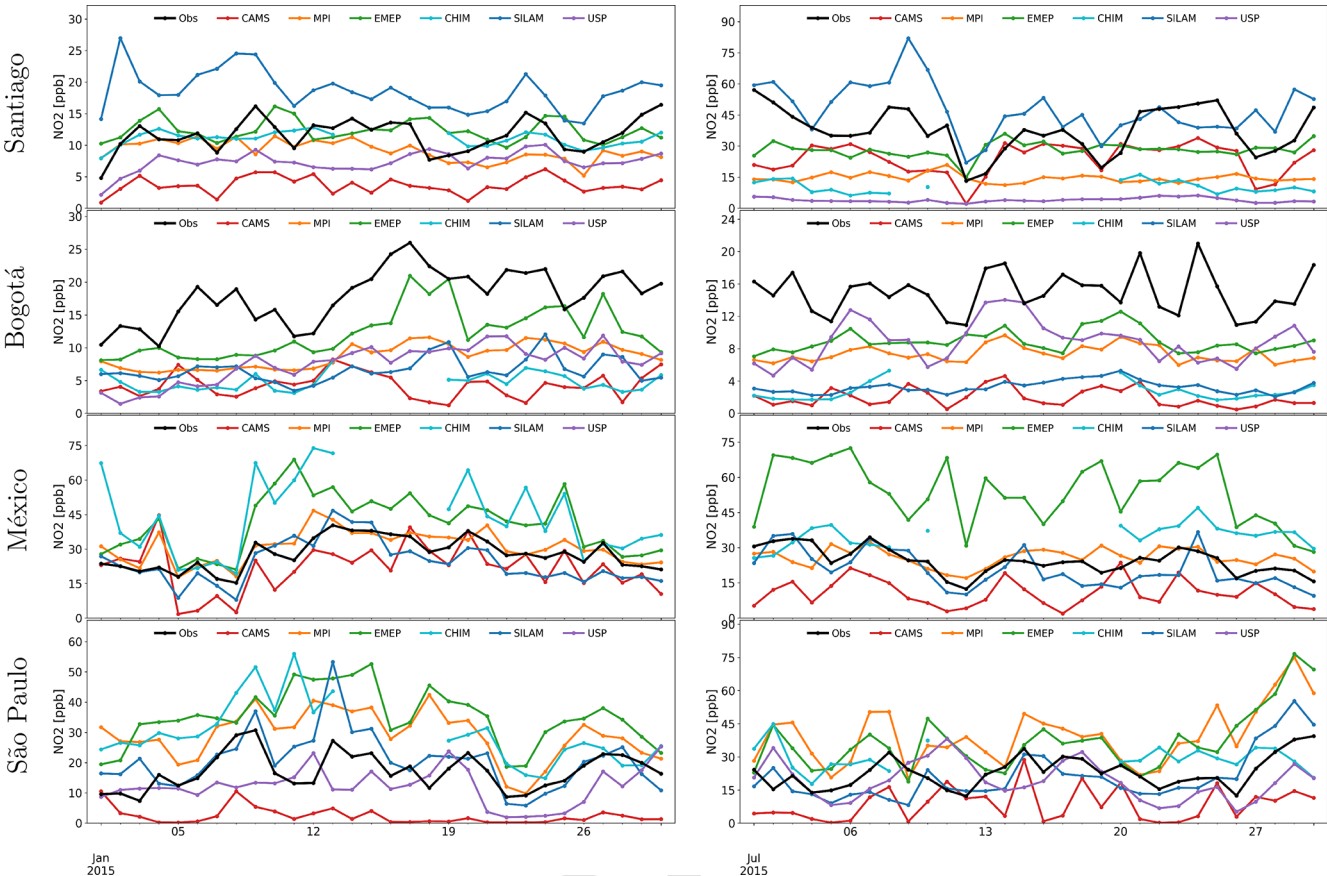

**Figure 2.** Observed (black) and simulated NO$_2$ daily mean concentrations in Santiago, Bogotá, Mexico City and São Paulo for January (left) and July (right) 2015.

46 % to 58 %). It is interesting to note the case of Bogotá, where all models consistently underestimate NO$_2$, but the model variation is the lowest (8 ppm with CVs of 39 % and 56 %).

## 5 Ensemble performance

The median ensemble underestimates NO$_2$ concentrations in Bogotá and to a lesser extent in Santiago. This is consistent with the underestimation trend by most of the models. The ensemble in these two cities has some of the lowest MN-BIAS, FGE and $R$, but they are not always better than individual models (Table A2). On the contrary, in Mexico City and São Paulo, the ensemble median outperforms the models for NO$_2$. In summer and winter, the ensemble presents the lowest FGE in both cities. The correlation coefficient range is between 0.5 and 0.8 within the criteria benchmark $R > 0.5$ (Zhai et al., 2024). The MNBIAS values are also the lowest (−2.9 % to 17.7 %).

### 3.1.2 Ozone – O$_3$

**Observations**

The number of stations per city recording O$_3$ during January and July 2015 varies between 9 in Santiago and 29 in Mexico City (Appendix B). The highest observed ozone concentration was in Mexico City in July with an average of 31 ppb. However, this value is significantly lower than the surface ozone concentrations reported in the MAM (March–April–May) season, with values larger than 70 ppb (Barrett and Raga, 2016; Silva-Quiroz et al., 2019). The second largest ozone value occurs in São Paulo during January with daily averages of 24 ppb. This is probably due to an abundance of ozone precursors, in particular volatile organic compounds (VOCs) from the use of biofuels in the transportation sector (de Fatima Andrade et al., 2017; Gavidia-Calderón et al., 2024) and biogenic VOCs (Martins et al., 2006). Santiago experiences a marked seasonal cycle of ozone concentrations with summer values of approximately 22 ppb and winter concentrations around 3.6 ppb. This seasonal difference has been observed in other studies (Seguel et al., 2024). In Bogotá, ozone concentrations are the lowest and below 13 ppb.

## Model performance

In the four cities, simulations of $O_3$ are mainly overestimated (Fig. 3). In the summer in São Paulo and Mexico City, simulations can reach up to 100 ppb, which is significantly above the observations. In Santiago in the winter, the mean of models ($\sim$20 ppb) is significantly larger than observations, indicating that the models have difficulty reproducing low values of this secondary pollutant. In the summer, ozone estimates are much closer to observations. Similarly, in Bogotá, models estimate an average of 17 ppb, which is on the same order of magnitude as the observations.

The overestimation of $O_3$ in Santiago might be related to the underestimation of $NO_2$ previously described and the inadequate titration of ozone. Ozone formation in Santiago has been found to be VOC-limited (Seguel et al., 2020). This situation is also observed in Bogotá, where most models overestimate $O_3$ with a MNBIAS between 25 % and 80 % (Table A3). In contrast, in Mexico and São Paulo, the models that overestimate $NO_2$ also overestimate $O_3$. This complex situation is explained by the nonlinearities in the formation of ozone (Grewe, 2004). In general, correlation coefficients for $O_3$ are very low ($R < 0.3$), especially in São Paulo and Mexico City, indicating the challenge to adequately reproduce the spatial and temporal variability of this pollutant. Only in Santiago in January is the criteria benchmark for $O_3$ ($R > 0.5$) achieved by some models (Emery et al., 2017).

## Model intercomparison

In the case of global models, CAMS underestimates $O_3$ in the four cities except in Santiago during winter. Additionally, CAMS tends to have low correlation levels along with a higher bias and error (Table A3). SILAM displays a lower bias and error compared to CAMS. However, just like with CAMS, SILAM significantly overestimates $O_3$ levels in Santiago during the winter. In Bogotá, SILAM underestimates $O_3$ to a lesser extent than CAMS, with a larger FGE in July (74 %) than in January (22 %).

In São Paulo, daytime concentrations of ozone are generally overestimated by most models (except for CAMS). The largest overprediction of $O_3$ (MNBIAS from 30 % to 90 %) is associated with overestimation of $NO_2$, especially for MPI, EMEP and CHIMERE models. For the models with $NO_2$ levels in reasonable agreement with observations (SILAM, USP), the ozone overprediction is lower (MNBIAS < 25 %). Among the regional models, EMEP and WRF–MPI consistently overestimate $O_3$ levels in all cities, with relatively high MNBIAS and FGE. In contrast, WRF–USP proves particularly suitable for São Paulo, achieving some of the lowest FGE. CHIMERE also performs well in Santiago in the summer, likely owing to local adjustments and parameterizations tailored to these specific cities.

Figure 3 shows a relatively large model variation for ozone. The largest ozone variability is shown in Mexico City in summertime with a range of 62 ppb and a CV of 72 % (Table A8). This wide variability is caused by the simulation of the EMEP model (71 ppb) and CAMS (9.6 ppb), which represent the extreme cases of over- and underestimation. In a similar manner, in Bogotá, São Paulo and Santiago, the CVs are 61 %, 49 % and 47 %, respectively, explained by the strong underestimation of CAMS and severe overestimation by EMEP and WRF–MPI.

## Ensemble performance

In Santiago in January, the median ensemble showed one of the lowest MNBIAS and FGE, surpassed only by CHIMERE (Table A3), and achieved the criteria benchmark for this pollutant ($R > 0.75$) TS3 (Emery et al., 2017). In July, the overestimation of ozone by most models impacts the performance of the ensemble, which also overestimates $O_3$ concentrations. In Bogotá, the ensemble has some of the best scores for MNBIAS and FGE and represents an intermediate value between all models. In São Paulo, in wintertime, the ensemble has superior metrics (MNBIAS $\sim -2$ %) TS4 compared to any individual model, while in the summer the ensemble overestimates the observations as most models do. In Mexico City, the ensemble median performs better than all individual models with a MNBIAS between 4 % (summer) and 13 % (winter) and a FGE less than 32 %. Similar to the individual models, for most of the cases, the correlation coefficient for the ensemble does not meet any of the benchmarks (Emery et al., 2017).

### 3.1.3 Carbon monoxide – CO

## Observations

The number of stations per city recording CO during January and July 2015 varies between 7 in Bogotá and 24 in Mexico City (Appendix B). CO levels are generally below 1.0 ppm for all cities (Fig. 4). However, in Santiago during winter some values surpass 1.5 ppm due to a combination of adverse meteorological conditions and emissions from the transportation sector and residential combustion, commonly employed for heating in neighboring municipalities (Saide et al., 2016; Gallardo et al., 2012).

There is a slight increase of CO in São Paulo in July with respect to January, due to the atmospheric conditions where lower winds and lower boundary layer increased the primary pollutant concentration during winter. Additionally, biomass burning from wildfires, which begin in July and peak in August and September for the southern part of the Amazon rainforest, can bring more CO (Marlier et al., 2020). Likewise, larger CO concentrations in Bogotá in January are part of the wildfire season in northern South America lasting from the end of December until April (Mendez-Espinosa et al., 2019).

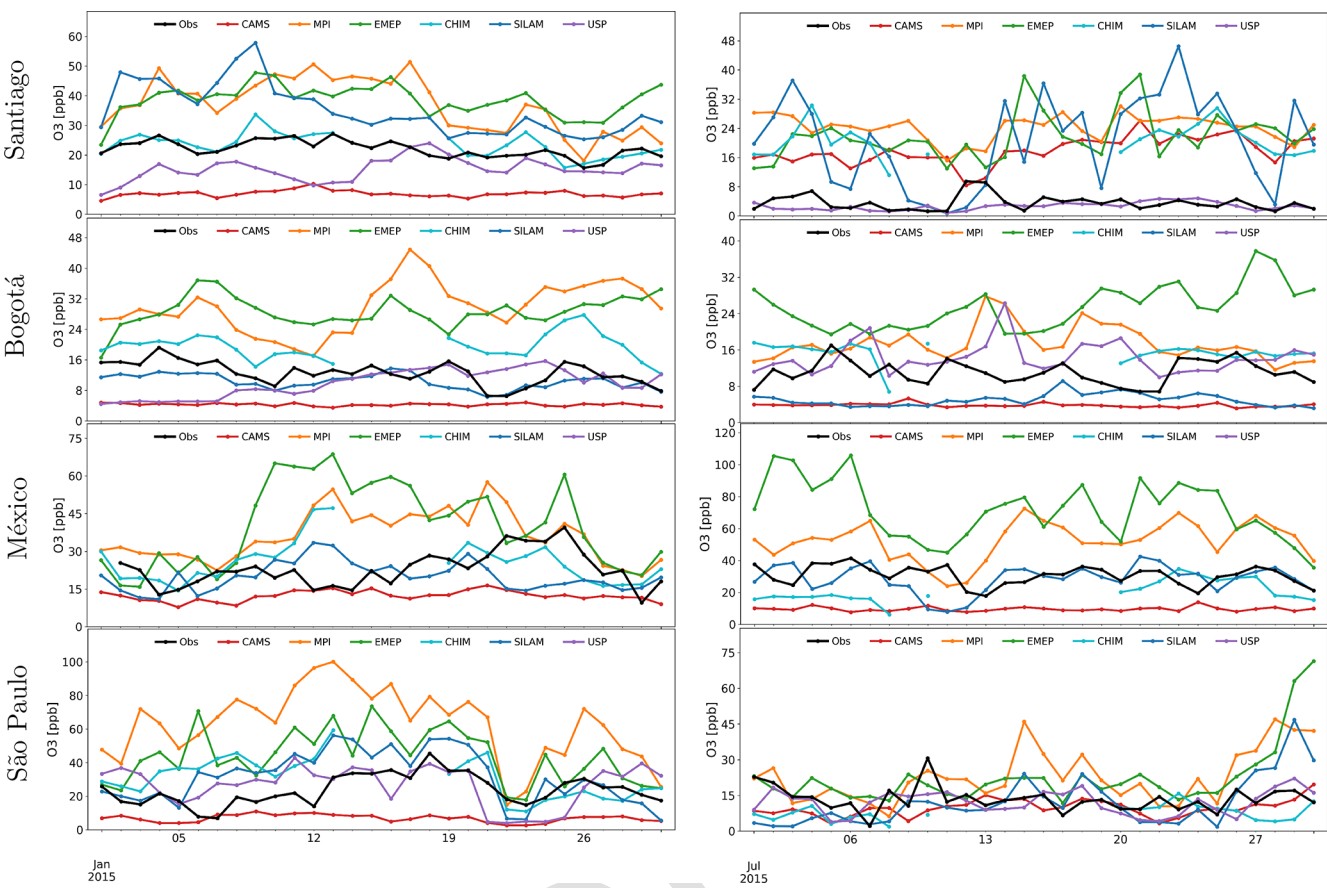

**Figure 3.** Observed (black) and simulated $O_3$ daily mean concentrations in Santiago, Bogotá, Mexico City and São Paulo for January (left) and July (right) 2015.

## Model performance

Santiago records the largest simulated value of CO in winter with peak of 5.0 ppm (Fig. 4). The second largest values are observed in Mexico City with values around 3.0 ppm. In both cases, models severely overestimate the observations with some MNBIAS larger than 100 % (Table A4). São Paulo displays intermediate values with an average CO of 0.5 ppm, and Bogotá has the lowest modeled values with an average of 0.27 ppm.

CO simulations in Santiago, São Paulo and Mexico City both over- and underpredict observations (Fig. 4). However, in Santiago in winter only the SILAM model overpredicts CO values (MNBIAS 98 %); the other models underpredict the values (MNBIAS between −152 % and −1 %). This situation could be explained by emissions, synoptic conditions or the models' simulation of the boundary layer (Mazzeo et al., 2018). In Bogotá, all models consistently underestimate the CO with a MNBIAS between −50 % and −131 % (Table A4). In January correlation coefficients for CO hover around 0.6, achieving benchmarks ($R > 0.4$) (Zhai et al., 2024). This result demonstrates the model's capability to reproduce the time variability of this pollutant in Bogotá,

even if the levels are under- or overestimated. The same situation is observed in Mexico City and São Paulo, where goal ($R > 0.6$) and criteria ($R > 0.4$) benchmarks are often achieved (Zhai et al., 2024).

The underestimation in Bogotá is similar to that observed for $NO_2$, which we attributed to a shortfall in emissions. According to the local inventory, CO emissions are predominantly attributed to mobile sources (99 %), with motorcycles contributing to 45 % of these emissions, automobiles accounting for 36 % and the remainder originating from other vehicles (SDA – Secretaría Distrital de Ambiente, 2018). Notably, it has been identified that motorcycle emissions are underestimated in Colombia (Rojas et al., 2023). The significant rise in the number of motorcycles in the country and their declining condition is not accurately reflected in global emission inventories, such as EDGAR 6.1.

Observed CO mixing ratios are also underestimated in cities such as Medellín, Guadalajara, Quito and Lima (Fig. 8), which might be explained by the coarse resolution of the model not capturing the local characteristics. It is possible that issues with CO emissions in global inventories or excess of OH radicals in photochemistry also contribute to

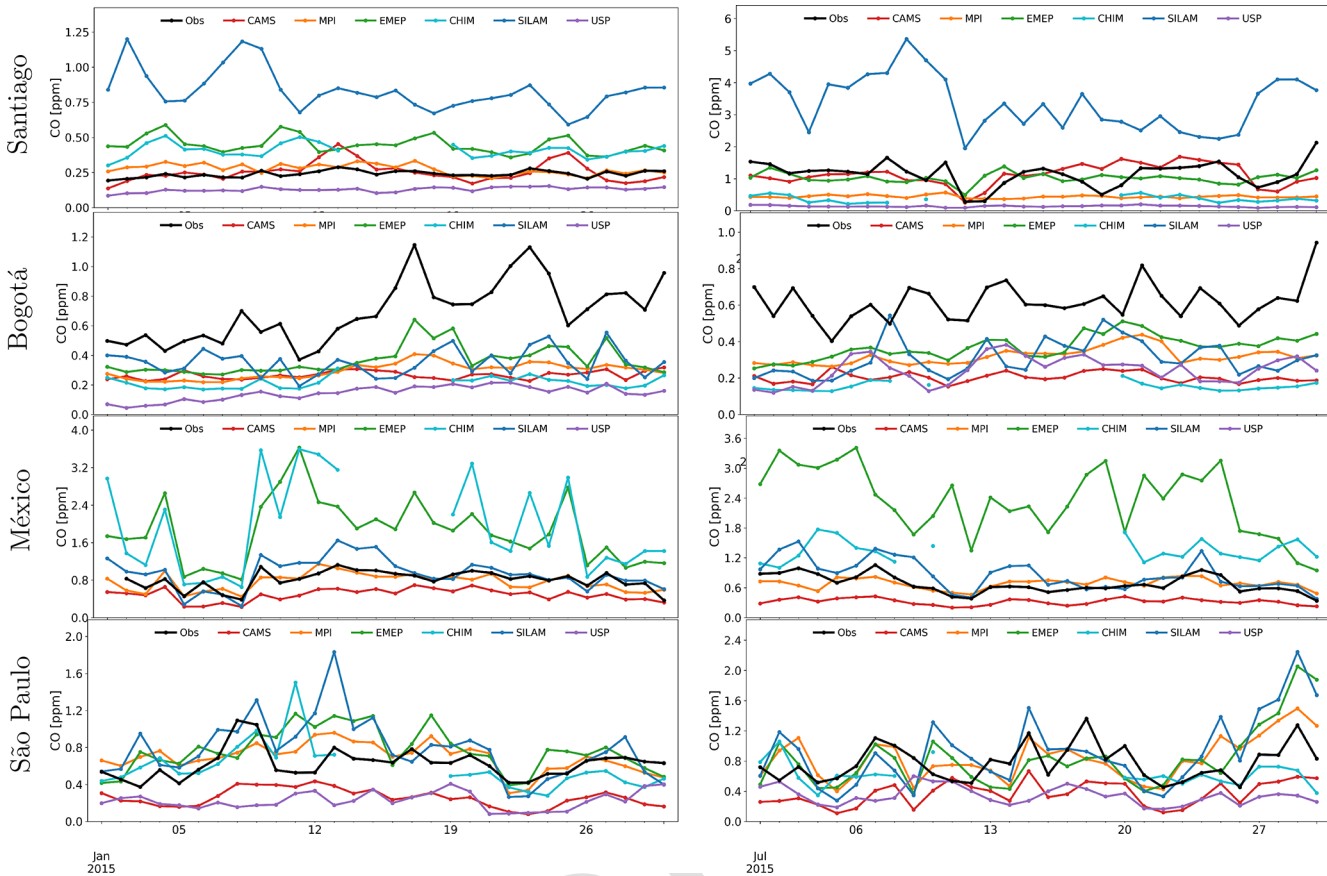

**Figure 4.** Observed (black) and simulated CO daily mean concentrations in Santiago, Bogotá, Mexico City and São Paulo for January (left) and July (right) 2015.

this trend. In addition, a major source of atmospheric CO is the oxidation of BVOCs (Worden et al., 2019), which are significantly underestimated in the Southern Hemisphere (Zeng et al., 2015).

In São Paulo, five out of six models slightly underestimate CO with a relatively high correlation coefficient. The simulated concentrations for daily values range from 0.1 to 2.0 ppm, similar to that found in other studies (Deroubaix et al., 2024). Nevertheless, concentrations exceeding 1.2 ppm are simulated only for certain days (13 January and 30 July) and are probably due to wood burning (Fig. C1).

## Model intercomparison

Global models, particularly CAMS, tend to underestimate CO levels in Bogotá, São Paulo and Mexico City with a MN-BIAS $< -50$ %. In Santiago, CAMS adequately simulates CO levels with a MNBIAS $< \pm 2.5$ % and a FGE $< 25$ %. The correlation coefficient achieves the criteria benchmark ($R > 0.4$) proposed by Zhai et al. (2024). SILAM underestimates CO in Bogotá (model / observations of $\sim 0.6$) and overestimates it in Santiago, while it performs relatively well in São Paulo and Mexico City (MNBIAS $< 22$ %). Corre-

lation coefficients meet the criteria and goal benchmarks ($R > 0.4$ and $R > 0.6$) proposed by Zhai et al. (2024).

When it comes to regional models, WRF–USP consistently underestimates CO levels with a high bias (MN-BIAS $< -60$ %) and error (FGE $> 60$ %). WRF–MPI has better performance, especially in São Paulo and Mexico City (MNBIAS $< \pm 15$ %), and correlation coefficients within the goal benchmark (Zhai et al., 2024). EMEP and CHIMERE largely overestimate observations in Mexico City, while in São Paulo they closely match observations. In Santiago, these models tend to overpredict CO in the summer and underpredict it during the winter.

The largest model variation is observed in Santiago during wintertime with a range of 3.2 ppm and CV of 106 % (Table A8). Mexico City also shows large variation in summer (CV: 72 %) and winter (CV: 56 %). Bogotá and São Paulo present less variation between model results.

## Ensemble performance

In winter in Santiago and Bogotá in both periods, the ensemble follows the underestimation pattern of all models (Table A4). In São Paulo there are models with better perfor-

mance than the ensemble, but the ensemble results are reasonable, with a MNBIAS close to $-15\%$ and an $R$ of approximately 0.7. In Mexico City, the overestimation of CO by the EMEP and CHIMERE models (MNBIAS > 60 %) is
reduced in the ensemble (MNBIAS $\sim 15\%$).

### 3.1.4  Sulfur dioxide – SO$_2$

**Observations**

The number of stations per city recording SO$_2$ during January and July 2015 varies between 4 in Santiago and 26 in
Mexico City (Appendix B). The largest concentration of SO$_2$ is observed in Mexico City with values between 3.0 ppb (January) and 4.4 ppb (July) due to volcanic emissions (de Foy et al., 2009) and the heavy consumption of coal in power generation and cement production, especially in the proximity of the "Tula–Vito–Apasco" industrial area (SEMARNAT
and INECC, 2020). On the other hand, SO$_2$ in Bogotá, Santiago and São Paulo is lower, with concentrations ranging from 1.0 to 1.8 ppb (Fig. 5).

**Model performance**

The largest simulation is shown in Mexico City, with an average of 45 ppb SO$_2$, followed by São Paulo, with a mean concentration of 8.5 ppb. In Santiago, the average SO$_2$ value is 8.5 ppb. The lowest modeled values are found in Bogotá, with an average of 0.97 ppb (Table A5).
The models' simulated SO$_2$ exhibits significant discrepancies when compared to the observations, with severe overestimation in Santiago, Mexico City and São Paulo (Fig. 5), with a MNBIAS reaching up to 190 % and a FGE up to 200 % (Table A5). On the contrary, for Bogotá the predicted SO$_2$
values are in reasonable alignment with the observations, except for the WRF–Chem USP simulation, which drastically underestimates SO$_2$ (MNBIAS: $-200\%$) (Table A5).
  The overestimation of SO$_2$ levels could stem from issues within global emission inventories. In fact, an overestima-
tion of SO$_2$ emissions in CAMS was observed for Buenos Aires and Santiago when compared to the PAPILA inventory (Castesana et al., 2022). These emissions primarily originate from the energy and industrial sectors, where the sulfur content in coal appears to significantly contribute to this overes-
timation.
  The good performance in Bogotá might be related to lower SO$_2$ emissions apportioned in the city. In fact, the vast majority of SO$_2$ emissions ($\sim 90\%$) in Colombia originate from the industrial and energy production sectors (IDEAM, 2020).
However, these facilities are typically located outside major urban areas. Bogotá contributes only 1.5 % of the total national SO$_2$ emissions (de Ambiente, 2018).

**Model intercomparison**

CAMS and SILAM severely overestimate SO$_2$ in Mexico City, São Paulo and Santiago with a MNBIAS and a FGE
larger than 100 %. In Bogotá, both global models underestimate SO$_2$ concentrations (MNBIAS from $-56\%$ to $-80\%$) but with lower FGE ($< 80\%$) than CAMS. In January, correlation coefficients met the criteria benchmark ($R > 0.35$) suggested by Zhai et al. (2024).
  The performance of regional models for SO$_2$ is quite diverse. WRF–USP severely underestimates SO$_2$ in all cities (MNBIAS close to $-200\%$). In Santiago, Mexico City and São Paulo the models overestimate SO$_2$ in a similar fashion to global models. In Bogotá, EMEP and WRF–MPI show the
lowest MNBIAS ($< 16\%$).
  The largest model variation for SO$_2$ is found in Mexico City, where the range of models is 200 ppb, and the CV is larger than 150 % (Table A8). In Santiago and São Paulo, the model variation is close to a CV of 95 %. In Bogotá, the
variation is the lowest (CV: $\sim 75\%$).

**Ensemble performance**

In Mexico City, Santiago and São Paulo, SO$_2$ is overestimated by all models, except the USP. Therefore, the median ensemble also overestimates SO$_2$ concentration and does not
represent any improvement in the evaluation metrics (Table A5). In Bogotá, the ensemble tends to underestimate the concentrations (MNBIAS: $\sim -55\%$) to a lesser extent than individual models.

### 3.1.5  Fine particulate matter – PM$_{2.5}$                         75

**Observations**

The number of stations per city recording PM$_{2.5}$ during January and July of 2015 varies between 9 in Bogotá and 16 in Mexico City (Appendix B). The largest PM$_{2.5}$ concentrations are found in Santiago during the Southern Hemisphere
winter, with daily values around 56 µg m$^{-3}$. This can be attributed to adverse meteorological conditions and emissions from transportation and residential combustion in the surrounding municipalities (Mazzeo et al., 2018; Saide et al., 2016). The second largest values are shown in Mexico City,
with an average of 23 µg m$^{-3}$ due to local emission sources. In São Paulo, PM$_{2.5}$ levels are larger in July (19 µg m$^{-3}$) than in January (16 µg m$^{-3}$), due to the impact of wildfires from the Amazon basin and sugarcane burning (de Fatima Andrade et al., 2017). In Bogotá, PM$_{2.5}$ concentrations
are the lowest in July (13 µg m$^{-3}$) due to the influence of trade winds (Pachón et al., 2018) but with larger values in January (19 µg m$^{-3}$) due to biomass-burning events and frequent thermal inversions (Ramírez et al., 2018).

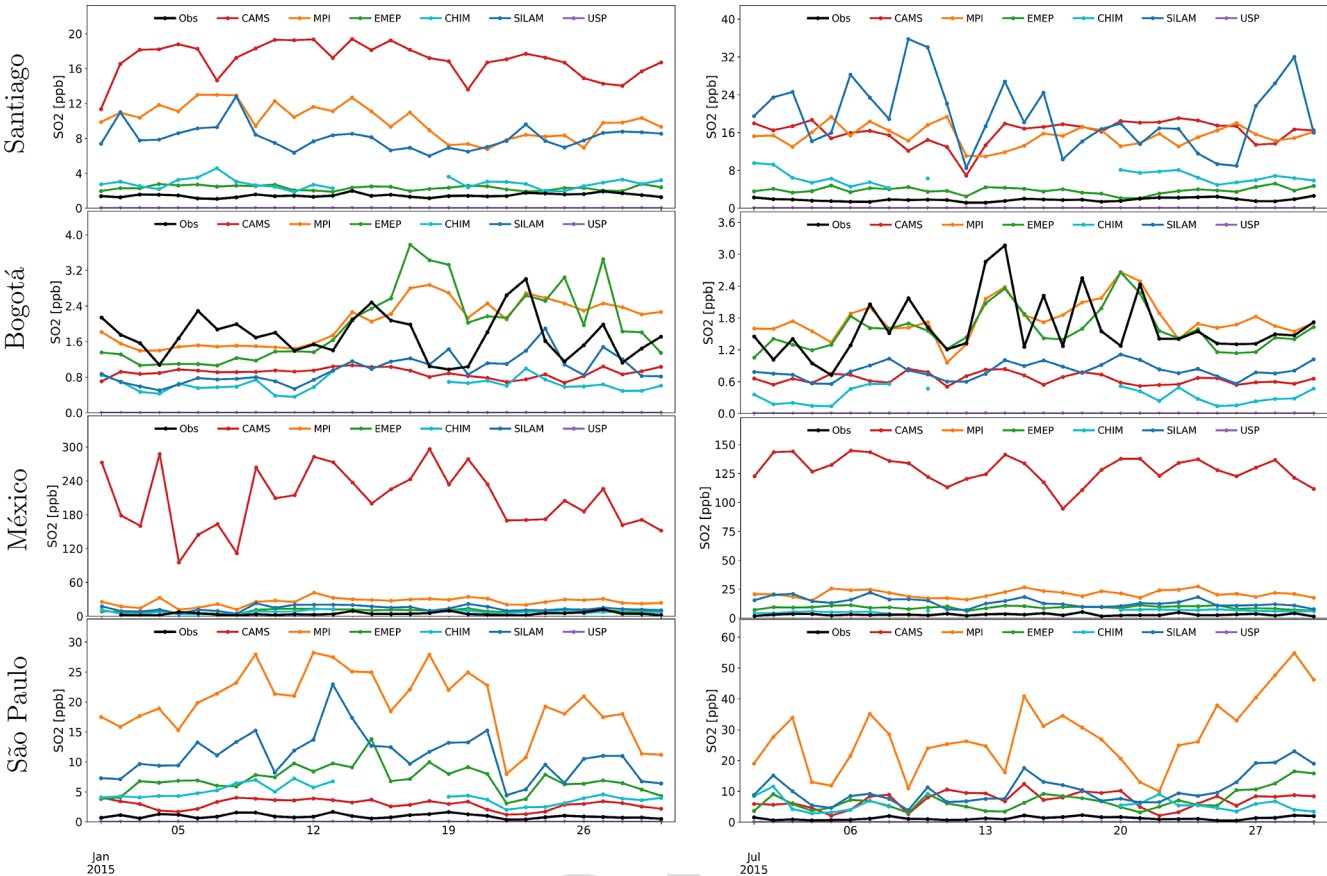

**Figure 5.** Observed (black) and simulated $SO_2$ daily mean concentrations in Santiago, Bogotá, Mexico City and São Paulo for January (left) and July (right) 2015.

## Model performance

In Santiago in wintertime, the mean of the models is larger than observations, whereas in summer the simulations are mostly below observations. In Mexico City, simulated values are approximately double the observations. In São Paulo, $PM_{2.5}$ is under- and overpredicted by the models. In Bogotá, most of the simulations are below the observations (Fig. 6).

In Santiago, Bogotá and Mexico City, models over- and underpredict $PM_{2.5}$ (Table A6). In São Paulo, overestimation is observed in all models with the exception of WRF–USP and may be linked to an excess of fire emissions, as suggested by other studies (Deroubaix et al., 2024). The MNBIAS varies from 39 % to 120 % except for WRF–USP, whose MNBIAS is negative (Table A6). The correlation coefficients for $PM_{2.5}$ are in some cases larger than the goal ($R > 0.7$) or criteria ($R > 0.4$) benchmarks proposed by Emery et al. (2017). It is worth noting the cases of Mexico City in January and São Paulo in July, where most models achieve the goal metric. In smaller urban areas like Medellín, Lima and Quito (Fig. 8), most models tend to underestimate observations, potentially due to the coarse resolution of the models.

Hourly simulations of $PM_{2.5}$ are useful to understand the discrepancies between model and observations. In Fig. C2, we show the hourly data and model outputs. In São Paulo, the highest $PM_{2.5}$ concentrations are simulated by SILAM on 13 January ($> 320 \,\mu g \, m^{-3}$) and 30 July ($> 400 \,\mu g \, m^{-3}$), which corresponds to days with high simulated CO values as well (Fig. C1 in Appendix C) and may indicate an overestimation of biomass burning by the IS4FIRES module in SILAM. From 15 to 30 January there is also an excess of $PM_{2.5}$ from SILAM.

In Mexico City, the highest $PM_{2.5}$ concentrations are simulated by the CAMS model with about $250 \,\mu g \, m^{-3}$ in January and $160 \,\mu g \, m^{-3}$ in July (Fig. C3), which are severely overestimated. The large $PM_{2.5}$ values are distributed in the whole period rather than specific days and do not correspond to high CO concentrations to suspect the influence of fires. This situation might indicate a local and continuous source of $PM_{2.5}$.

## Model intercomparison

Both global models consistently overestimate $PM_{2.5}$ in Santiago, São Paulo and Mexico City, but they behave differently

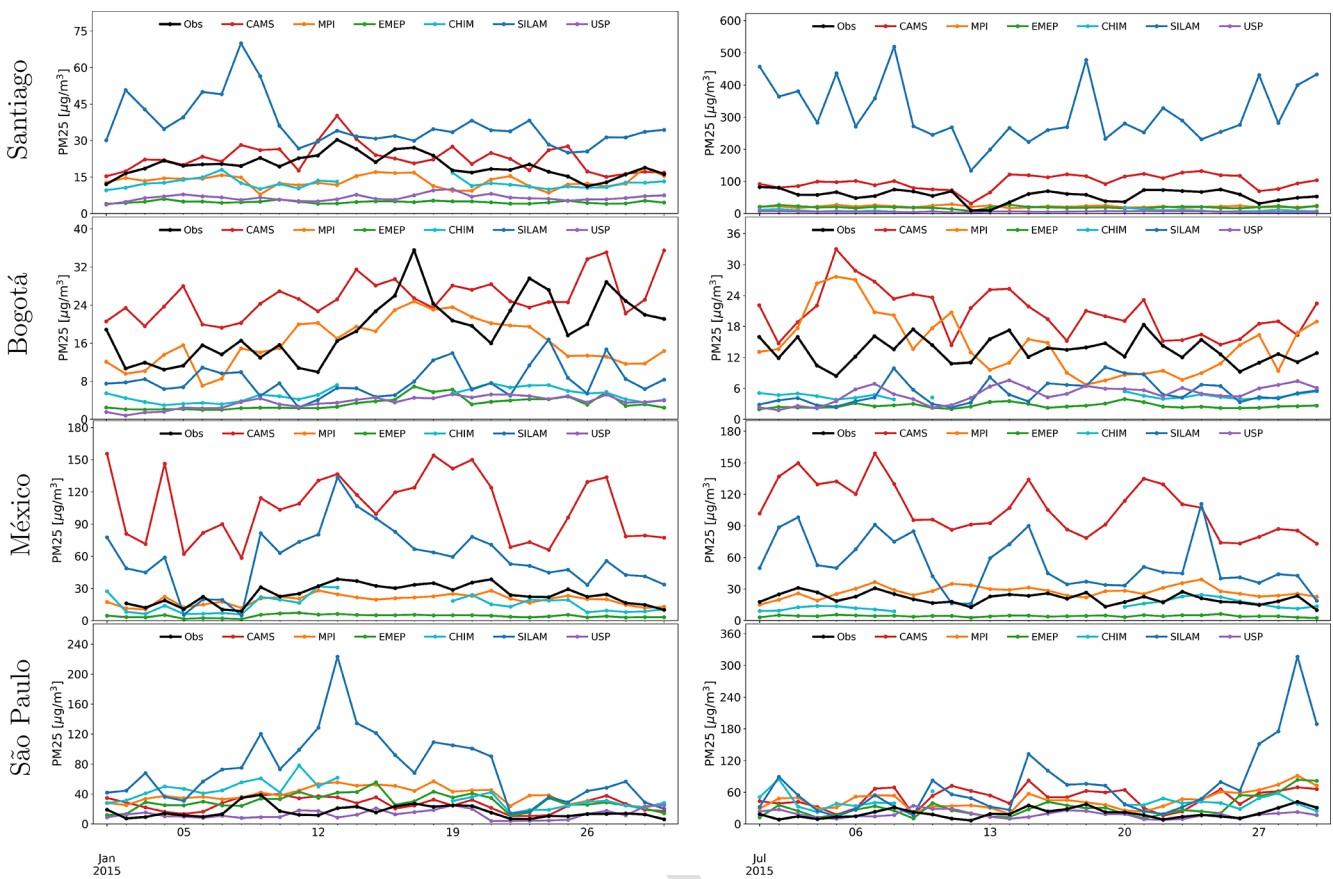

**Figure 6.** Observed (black) and simulated PM$_{2.5}$ daily mean concentrations in Santiago, Bogotá, Mexico City and São Paulo for January (left) and July (right) 2015.

in Bogotá. In Mexico City, CAMS has a greater overestimation than SILAM, but in São Paulo and Santiago SILAM values are larger (Fig. 6). In Bogotá, CAMS overestimates PM$_{2.5}$ (MNBIAS $\sim$ 37 %), whereas SILAM underestimates it (MNBIAS $\sim$ −85 %). The SILAM correlation coefficient meets the criteria benchmark suggested by Emery et al. (2017).

Among the regional models, EMEP shows the largest underestimation (MNBIAS $<$ −110 %) in all cites, except in São Paulo, where the model overestimates PM$_{2.5}$ but is within the criteria benchmark (MNBIAS $<$ ±60 %) (Boylan and Russell, 2006) and with a correlation coefficient ($R >$ 0.4) that meets the criteria benchmark by Emery et al. (2017) in July (Table A6). WRF–USP heavily underestimates in Bogotá and Santiago but performs well in São Paulo with the lowest errors. This difference in behavior might be explained by a good adaptation of the model's inputs to the city. The WRF–MPI model meets goal benchmarks for MNBIAS and FGE in Bogotá and Mexico City.

The largest model variation is observed in Mexico City and Santiago during wintertime with a CV greater than 100 % (Table A8). Santiago in summer and Bogotá present

intermediate values (CV of 70 to 80 %), whereas São Paulo shows the least variability between models (CV $<$ 56 %).

### Ensemble performance

Considering the large underestimation of most models in Bogotá and Santiago, the ensemble displays lower bias and error than some of the individual models (Table A6). In Mexico City, the ensemble outperforms models with a MN-BIAS of −5 % in January and +30 % in July, achieving the goal benchmark suggested by Boylan and Russell (2006), as well as the correlation coefficient ($R >$ 0.8) in January. For São Paulo, all models tend to overestimate PM$_{2.5}$, so it follows that the ensemble presents the same behavior as MN-BIAS $>$ 67 %. The correlation coefficient meets the criteria benchmark ($R >$ 0.4) in both periods (Emery et al., 2017).

### 3.2 Spatial seasonal variability of predictions

For all pollutants, models and periods, maps of mean concentrations were constructed to visualize the spatial differences (Appendix D). In order to summarize the results, other spatial plots were also prepared: median ensemble (Fig. 7), median absolute deviation (Fig. E1 in Appendix E) and mean stan-

dard deviation (Fig. E2). In Fig. 7, pollution hot spots are clearly visible around major urban areas, in particular in São Paulo on the southeastern coast and Mexico City in the northwestern part of the continent. São Paulo and Mexico City each cover a significant area, of approximately $3600\,\mathrm{km^2}$, spanning at least nine modeling cells ($400\,\mathrm{km^2}$ each). This extensive coverage offers some spatial representation of the physical and chemical atmospheric processes. Other regions highlighted on the maps include Lima and Santiago on the Pacific coast, Buenos Aires along the southern shore of the Río de la Plata, and cities in the northern part of South America like Quito, Bogotá, Medellín and Caracas. However, most of these cities are encompassed by six or fewer modeling cells, limiting the potential for significant spatial variation.

The temporal seasonality can also be observed in Fig. 7. The left and right panels show results for January and July, corresponding to the Southern Hemisphere summer and winter, respectively. For $SO_2$, major hot spots appear in Mexico City, São Paulo and the surrounding areas, and the Pacific coast in Chile. The $SO_2$ concentrations are associated with volcanic emissions and the use of coal in power generation, cement production and copper smelting, which are active in both summer and winter (Huneeus et al., 2006; SEMARNAT and INECC, 2020). Similarly, $NO_2$ hot spots are common in major urban areas due to transportation emissions.

In January, the median ensemble shows high concentrations of $PM_{10}$ in several areas. In the south of Argentina, the concentrations are primarily due to dust from the Patagonia desert areas (Gassó and Torres, 2019). In the north of Brazil and the Guianas, increased $PM_{10}$ levels are most likely associated with fires in the Orinoco basin during the dry season (Hernandez et al., 2019). In a similar manner, $PM_{2.5}$ concentrations show an increase in the northern part of Brazil due to biomass burning. Large concentrations of $PM_{2.5}$ in São Paulo in both January and July are probably caused by overestimation of fires, as previously discussed.

During the austral summer, the southeastern part of Brazil (including São Paulo) displays high concentrations of ozone that were simulated mainly by the regional models WRF–Chem and EMEP and the global SILAM (Fig. D3 in Appendix D). Several studies have shown the influence of urban plumes of $NO_2$ on the Amazon rainforest, rich in BVOCs, with the consequent generation of ozone (Rafee et al., 2017; Nascimento et al., 2022). In January, simulated $O_3$ concentrations are also high in Mexico City during winter, a situation that has been observed in other studies (Barrett and Raga, 2016). There is a maximum of CO in the area between north of Argentina, south of Bolivia, Paraguay and south of Brazil, probably related to fires and the abundance of BVOCs.

In July, during the austral winter, concentrations of CO, $PM_{2.5}$ and $PM_{10}$ are significant in Santiago due to transportation and residential heating emissions under adverse meteorological conditions. $PM_{10}$ concentrations are high in the Caribbean and central Mexico, primarily due to the transport of Saharan dust into these urban areas (Kramer and Kirtman, 2021; Ramírez-Romero et al., 2021). Similarly, along the Pacific coast between Chile and Peru, increased $PM_{10}$ is probably explained by anthropogenic emissions of copper smelters in connection with strong eastern wind events (Huneeus et al., 2006). Large concentrations of $O_3$ are visible in Mexico City that are associated with clear skies under high-pressure atmospheric conditions (Barrett and Raga, 2016). Elevated $O_3$ values in the Andes between northern Chile and central Peru might be explained by the abundance of VOCs from metropolitan regions and industrial zones (Seguel et al., 2020).

The median absolute deviation maps (Fig. E1) and the standard deviation maps (Fig. E2) display spatial differences between model simulations. In particular, for particulate matter ($PM_{10}$ and $PM_{2.5}$) a notorious dissimilarity is observed in northern Brazil in January, Venezuela in July and the south of Argentina in both periods. The reason for this disagreement is the simulation of the WRF–MPI model, which contributes with significant PM mass in the mentioned zones, probably due to an overestimation of fires in the northern part of the continent and dust in the southern areas. In July, CO showed large differences in the Colombian and Peruvian Amazon, mostly driven by the EMEP model. This situation might be related to an incorrect estimation of BVOC emissions as precursors of CO in forested areas. The inadequate simulation of $NO_2$ by the CAMS model, explained in Sect. 3.1.1, is the cause of the large standard deviation of model results for this pollutant.

## 3.3 Large versus small urban areas

The coarse resolution used in the modeling systems ($0.2° \times 0.2°$) poses challenges in adequately representing the intricate topography and diverse meteorological conditions of the different cities in LAC. Capturing these physical phenomena can be very difficult and requires a finer scale with much greater computational demand. In the last few years, emission inventories for LAC at high spatial and temporal resolution have been constructed (Castesana et al., 2022; Álamos et al., 2022; Puliafito et al., 2015, 2017; Rojas et al., 2023), and it is expected they will complement existing global emission inventories at coarse resolution. We observe that, in large urban areas ($> 3500\,\mathrm{km^2}$), the models tend in general to have a lower and positive MNBIAS compared to medium-size ($600 < \mathrm{area} < 3600\,\mathrm{km^2}$) or small-size ($\mathrm{area} < 600\,\mathrm{km^2}$) cities (Fig. 8). For example, for Mexico City and São Paulo, the two largest cities in LAC, the mean of the models shows the lowest MNBIAS and FGE for CO ($-27\%$ to $29\%$) and $NO_2$ ($-6\%$ to $6\%$), while in other cities they display a larger and negative MNBIAS and FGE (Tables A2 and A4). The discrepancies in $NO_2$ have a corresponding impact in the overestimation of $O_3$. For particulate matter, a similar pattern is observed, with a positive MNBIAS for larger urban areas and a negative MNBIAS

for medium and small cities. High-resolution simulations are necessary to resolve the spatial variation, but unfortunately global models at high performance are scarce in the Southern Hemisphere (Zhang et al., 2023).

Although the size of cities can influence the performance of the models at coarse resolution, other challenging features for models exist. For instance, Bogotá and Santiago have several challenges in terms of topography and meteorology (Mazzeo et al., 2018; Nedbor-Gross et al., 2017; Reboredo et al., 2015), and local emissions are not always accounted for in global inventories (Castesana et al., 2022; Huneeus et al., 2020; Osses et al., 2022; Rojas et al., 2023). Ideally, we would have access to more cities of various sizes to make this determination with more certainty; unfortunately, local measured data were only available for the cities we considered.

## 4 Conclusions and future developments

This study performed the first intercomparison and model evaluation effort in Latin America with the idea to develop an AQF system that can inform the public about air pollution episodes and support policy actions. Despite the limitations of air quality and emissions data, as well as computing resources, the scientific community in Latin America, with international support, has achieved significant progress in air quality modeling and in understanding the fate and transport of pollutants in the region. For instance, the impacts of Saharan dust, biomass burning from the Orinoco and the Amazon basins, and biogenic VOCs of the Amazon rainforest are becoming better understood through modeling.

Several challenges still exist. In addition to the intricate topography and diverse meteorological conditions, limitations are found in anthropogenic, volcanic and biogenic emissions; in spatial and temporal profiles; in land use and vegetation types; and in other data that are relevant for the calculation of wildfire emissions. This last source is crucial in the region under a climate change scenario, for which adequate parameterization of biomass burning is necessary. The boundary conditions of the models can be improved, which are especially important for long-lived species. The experience of local researchers who have been implementing air quality models for several years can greatly benefit international efforts, such as global emission inventories and the recently launched WMO GAFIS initiative.

At this first stage of development, interesting and insightful findings were identified for the region. Despite the fact that some of the models were still in an early phase for regional implementation, most models could adequately reproduce air quality observations with the best performance observed for nitrogen dioxide in Mexico City and São Paulo. These enormous urban areas ($> 3500\,km^2$) outperformed Bogotá and Santiago, which are cities between 500 and $1000\,km^2$. This suggests an accurate portrayal of the temporal and spatial variability in large cities with the current model resolution ($0.2° \times 0.2°$) and the need for a finer model domain in smaller cities that could capture circulation and emission features. At the moment, high-resolution global simulations in the Global South remain rare.

The ensemble median was evaluated on its potential to outperform individual models. In certain periods and cities, the ensemble performed better than any individual models, for example, when the errors of the models compensate for each other but not when the errors are recurring in all the models. The results varied per city, pollutant and period. Before defining whether the ensemble is the correct approximation for an AQF system, more research is necessary. This work only looked at 2 months (1 in summer and 1 in winter); a thorough analysis of one entire annual cycle with sufficient spinup time should be conducted. More observations should also be included for model calibration and evaluation. For 2015, only eight cities in LAC had data that complied with quality and completeness criteria. In recent years, more AQ networks have been implemented and data are more publicly available.

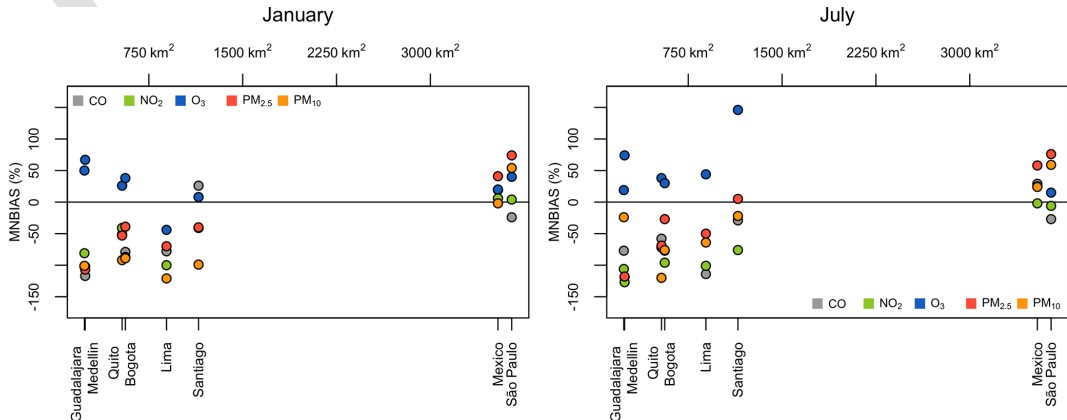

**Figure 7.** Spatial variability of simulated $PM_{10}$, $PM_{2.5}$, $O_3$ and CO in LAC for January and July 2015 (based on the median of the models).

**Figure 8.** MNBIAS estimated for large and small urban areas.

**Appendix A: Evaluation scores**

**Table A1.** Metrics used for model evaluation.

| Metric | Formula for each city, model and month |
| --- | --- |
| Model / observation ratio | $\text{ratio} = \frac{\overline{m}}{\overline{O}} = \frac{\sum_d m_{\mathrm{d}}}{\sum_d O_{\mathrm{d}}}$ |
| Mean bias | $\text{BIAS} = \frac{1}{N}\sum_d (m_{\mathrm{d}} - O_{\mathrm{d}})$ |
| Modified normalized bias | $\text{MNBIAS} = \frac{2}{N}\sum_d \frac{m_{\mathrm{d}} - O_{\mathrm{d}}}{m_{\mathrm{d}} + O_{\mathrm{d}}}$ |
| Fractional gross error | $\text{FGE} = \frac{2}{N}\sum_d \left|\frac{m_{\mathrm{d}} - O_{\mathrm{d}}}{m_{\mathrm{d}} + O_{\mathrm{d}}}\right|$ |
| Root mean square error | $\text{RMSE} = \sqrt{\frac{1}{N}\sum_d (m_{\mathrm{d}} - O_{\mathrm{d}})^2}$ |
| Correlation coefficient | $R = \frac{1}{N}\frac{\sum_d (m_{\mathrm{d}} - \overline{m})(O_{\mathrm{d}} - \overline{O})}{\sigma_{\mathrm{m}}\sigma_O}$ |
| Coefficient of variation | $\text{CV} = \frac{\sigma_{\mathrm{m}}}{\overline{m}}$ or $CV = \frac{\sigma_O}{\overline{O}}$ |
| **Metric** | **formula for each pixel and hour** |
| Median of models (ensemble) | $\text{MED} = \text{median}(\{m_i\})$ with $i \in N, 1 \geq i \geq 6$ |
| **Metric** | **formula for each pixel** |
| Median absolute deviation | $\text{MAD} = \text{median}(\{|m_{i,d} - \text{MED}_d|\})$ $i \in N, 1 \leq i \leq 6$ and $d \in N, 1 \leq d \leq 31$ |
| **Metric** | **formula for each city, day and model** |
| Observation | $O_{\mathrm{d}} = \sum_{j,k} g_{A_{j,k}} M_{j,k}$ with $\sum_{j,k} g_{A_{j,k}} = 1$ $j,k \in N$ representing a specific pixel $g_{A_{j,k}}$ proportion of area of the pixel with the area of the polygon of the city |

$O_{\mathrm{d}}$ and $m_{\mathrm{d}}$ are the observation and modeled value for each day. $\overline{m}$ is the mean of the models for each month and $\overline{O}$ the mean of the observations for each city. $\sigma_{\mathrm{m}}$ denotes the standard deviation for each model. $N$ is the number of model–observation pairs available for each month.

**Table A2.** $NO_2$ model evaluation scores (January and July).

| $NO_2$ | City | Ensemble | | Mean | | CAMS | | MPI | | EMEP | | CHIM | | SILAM | | USP | |
|---|---|---|---|---|---|---|---|---|---|---|---|---|---|---|---|---|---|
| | | Jan | Jul | Jan | Jul | Jan | Jul | Jan | Jul | Jan | Jul | Jan | Jul | Jan | Jul | Jan | Jul |
| Model / observations | Santiago | 0.81 | 0.44 | 0.89 | 0.58 | 0.31 | 0.63 | 0.78 | 0.39 | 1.06 | 0.75 | 0.94 | 0.27 | 1.60 | 1.26 | 0.63 | 0.10 |
| | Bogotá | 0.40 | 0.34 | 0.42 | 0.38 | 0.24 | 0.13 | 0.48 | 0.49 | 0.68 | 0.60 | 0.26 | 0.18 | 0.37 | 0.22 | 0.42 | 0.59 |
| | Mexico | 1.04 | 0.97 | 1.15 | 1.18 | 0.78 | 0.44 | 1.11 | 1.06 | 1.46 | 2.21 | 1.57 | 1.44 | 0.87 | 0.85 | | |
| | São Paulo | 1.18 | 1.06 | 1.24 | 1.11 | 0.14 | 0.37 | 1.69 | 1.72 | 2.04 | 1.61 | 1.72 | 1.28 | 1.20 | 0.92 | 0.71 | 0.83 |
| MNBIAS [%] | Santiago | −18.8 | −73.5 | −9.7 | −47.9 | −106 | −46.7 | −22.6 | −83.3 | 7.2 | −24.5 | −2.2 | −118 | 47.0 | 24.6 | −44.6 | −160 |
| | Bogotá | −85.8 | −97.8 | −81.6 | −88.7 | −121 | −156 | −68.7 | −67.0 | −39.2 | −49.6 | −111 | −138 | −92.1 | −128 | −84.5 | −52.2 |
| | Mexico | 1.7 | −2.9 | 12.0 | 17.0 | −35.0 | −83.7 | 10.4 | 7.3 | 35.5 | 74.0 | 44.0 | 30.8 | −18.2 | −20.3 | | |
| | São Paulo | 17.7 | 6.8 | 22.8 | 10.8 | −157 | −107 | 51.2 | 51.1 | 69.1 | 44.3 | 54.0 | 28.3 | 14.1 | −14.7 | −40.6 | −24.2 |
| RMSE [ppb] | Santiago | 3.11 | 23.64 | 2.45 | 18.36 | 8.23 | 17.17 | 3.58 | 25.67 | 3.07 | 14.65 | 2.27 | 32.10 | 7.53 | 15.15 | 4.88 | 35.30 |
| | Bogotá | 11.22 | 10.16 | 10.89 | 9.55 | 14.41 | 13.22 | 9.73 | 7.98 | 6.60 | 6.63 | 12.85 | 12.44 | 11.83 | 11.92 | 10.81 | 6.92 |
| | Mexico | 5.06 | 3.97 | 7.13 | 5.81 | 10.26 | 14.93 | 5.22 | 5.23 | 15.87 | 31.52 | 21.45 | 12.25 | 6.48 | 6.01 | | |
| | São Paulo | 5.90 | 7.03 | 6.59 | 6.94 | 15.92 | 16.31 | 13.74 | 19.38 | 19.57 | 18.07 | 15.56 | 12.38 | 7.70 | 8.82 | 8.70 | 10.81 |
| FGE | Santiago | 0.24 | 0.74 | 0.18 | 0.48 | 1.06 | 0.48 | 0.27 | 0.84 | 0.21 | 0.36 | 0.16 | 1.19 | 0.47 | 0.30 | 0.46 | 1.60 |
| | Bogotá | 0.86 | 0.98 | 0.82 | 0.89 | 1.22 | 1.56 | 0.69 | 0.67 | 0.39 | 0.50 | 1.12 | 1.39 | 0.92 | 1.28 | 0.84 | 0.52 |
| | Mexico | 0.14 | 0.13 | 0.18 | 0.19 | 0.43 | 0.84 | 0.14 | 0.17 | 0.36 | 0.74 | 0.45 | 0.36 | 0.24 | 0.26 | | |
| | São Paulo | 0.24 | 0.23 | 0.26 | 0.24 | 1.58 | 1.09 | 0.51 | 0.52 | 0.69 | 0.47 | 0.56 | 0.41 | 0.29 | 0.34 | 0.56 | 0.45 |
| $R$ | Santiago | 0.57 | 0.26 | 0.64 | 0.53 | 0.65 | 0.44 | 0.32 | 0.02 | 0.08 | 0.11 | 0.55 | 0.24 | 0.51 | 0.52 | 0.44 | 0.57 |
| | Bogotá | 0.74 | 0.25 | 0.76 | 0.31 | −0.12 | 0.44 | 0.74 | 0.34 | 0.66 | 0.15 | 0.14 | 0.15 | 0.56 | 0.43 | 0.65 | 0.17 |
| | Mexico | 0.80 | 0.73 | 0.78 | 0.73 | 0.58 | 0.40 | 0.80 | 0.48 | 0.69 | 0.52 | 0.68 | −0.21 | 0.81 | 0.80 | | |
| | São Paulo | 0.63 | 0.49 | 0.62 | 0.60 | 0.36 | 0.47 | 0.54 | 0.62 | 0.48 | 0.52 | 0.49 | −0.20 | 0.67 | 0.64 | 0.29 | 0.18 |

* Ensemble: based on the median value of the models; mean: arithmetic mean of the models; CAMS: Copernicus Atmosphere Monitoring Service's (CAMS); MPI: WRF–Chem executed by MPIM; EMEP: European Monitoring and Evaluation Programme; CHIM: CHIMERE transport model; SILAM: System for Integrated modeling of Atmospheric composition; USP: WRF–Chem executed by University of São Paulo.

**Table A3.** $O_3$ model evaluation scores (January and July).

| $O_3$ | City | Ensemble | | Mean | | CAMS | | MPI | | EMEP | | CHIM | | SILAM | | USP | |
|---|---|---|---|---|---|---|---|---|---|---|---|---|---|---|---|---|---|
| | | Jan | Jul | Jan | Jul | Jan | Jul | Jan | Jul | Jan | Jul | Jan | Jul | Jan | Jul | Jan | Jul |
| Model / observations | Santiago | 1.21 | 4.51 | 1.19 | 4.95 | 0.32 | 5.01 | 1.67 | 6.78 | 1.75 | 6.12 | 1.06 | 5.72 | 1.60 | 5.76 | 0.69 | 0.74 |
| | Bogotá | 1.24 | 1.20 | 1.35 | 1.22 | 0.34 | 0.35 | 2.36 | 1.57 | 2.27 | 2.30 | 1.54 | 1.38 | 0.82 | 0.45 | 0.78 | 1.29 |
| | Mexico | 1.12 | 1.05 | 1.19 | 1.23 | 0.54 | 0.31 | 1.58 | 1.70 | 1.76 | 2.33 | 1.12 | 0.68 | 0.87 | 0.93 | | |
| | São Paulo | 1.39 | 0.97 | 1.44 | 1.11 | 0.29 | 0.72 | 2.65 | 1.71 | 1.84 | 1.67 | 1.30 | 0.58 | 1.34 | 0.90 | 1.13 | 0.89 |
| MNBIAS [%] | Santiago | 18.8 | 127 | 16.8 | 133 | −104 | 133 | 47.7 | 149 | 54.2 | 142 | 6.1 | 147 | 44.0 | 118 | −38.0 | −20.8 |
| | Bogotá | 23.5 | 19.9 | 31.4 | 21.3 | −95.2 | −94.0 | 79.7 | 43.8 | 77.7 | 78.0 | 43.6 | 32.3 | −18.7 | −73.9 | −25.9 | 24.8 |
| | Mexico | 12.6 | 4.1 | 17.5 | 20.0 | −56.1 | −102 | 45.3 | 50.2 | 49.3 | 77.0 | 9.1 | −42.1 | −12.4 | −10.3 | | |
| | São Paulo | 30.0 | −3.4 | 35.1 | 11.0 | −104 | −29.6 | 86.4 | 46.8 | 56.8 | 45.6 | 32.3 | −50.8 | 19.6 | −30.3 | 4.5 | −13.0 |
| RMSE [ppb] | Santiago | 5.62 | 13.44 | 4.84 | 14.88 | 15.29 | 15.17 | 16.39 | 21.13 | 17.01 | 19.66 | 3.23 | 17.76 | 14.85 | 20.92 | 8.52 | 2.55 |
| | Bogotá | 4.08 | 4.05 | 5.63 | 4.08 | 8.84 | 7.70 | 18.57 | 8.12 | 16.87 | 15.43 | 7.53 | 5.63 | 3.16 | 6.77 | 6.28 | 5.69 |
| | Mexico | 9.62 | 9.38 | 10.78 | 11.45 | 12.62 | 21.93 | 16.81 | 24.96 | 24.98 | 44.77 | 11.24 | 14.21 | 10.60 | 10.19 | | |
| | São Paulo | 30.0 | 7.06 | 14.58 | 7.45 | 19.00 | 7.30 | 43.50 | 14.24 | 24.62 | 16.06 | 16.30 | 9.90 | 14.31 | 10.84 | 12.89 | 6.24 |
| FGE | Santiago | 0.19 | 1.30 | 0.17 | 1.34 | 1.04 | 1.34 | 0.48 | 1.49 | 0.54 | 1.42 | 0.11 | 1.48 | 0.44 | 1.32 | 0.40 | 0.51 |
| | Bogotá | 0.25 | 0.27 | 0.33 | 0.28 | 0.95 | 0.94 | 0.80 | 0.45 | 0.78 | 0.78 | 0.44 | 0.40 | 0.22 | 0.74 | 0.46 | 0.34 |
| | Mexico | 0.31 | 0.23 | 0.34 | 0.27 | 0.57 | 1.02 | 0.45 | 0.53 | 0.56 | 0.77 | 0.31 | 0.50 | 0.39 | 0.29 | | |
| | São Paulo | 0.43 | 0.41 | 0.43 | 0.39 | 1.06 | 0.50 | 0.88 | 0.58 | 0.57 | 0.52 | 0.49 | 0.71 | 0.47 | 0.65 | 0.52 | 0.43 |
| $R$ | Santiago | 0.68 | −0.22 | 0.76 | −0.06 | 0.42 | −0.40 | 0.76 | −0.20 | 0.59 | −0.19 | 0.74 | 0.33 | 0.64 | 0.17 | −0.20 | −0.06 |
| | Bogotá | 0.42 | −0.19 | 0.05 | −0.36 | 0.15 | −0.12 | 0.02 | −0.29 | −0.16 | −0.12 | 0.56 | −0.13 | 0.69 | −0.09 | −0.53 | −0.23 |
| | Mexico | 0.07 | 0.02 | 0.09 | 0.09 | 0.26 | −0.00 | 0.32 | 0.14 | 0.08 | 0.13 | 0.04 | −0.19 | −0.26 | 0.11 | | |
| | São Paulo | 0.52 | 0.16 | 0.48 | 0.18 | 0.16 | −0.01 | 0.40 | 0.29 | 0.39 | 0.13 | 0.08 | −0.33 | 0.61 | 0.16 | 0.26 | 0.28 |

* Ensemble: based on the median value of the models; mean: arithmetic mean of the models; CAMS: Copernicus Atmosphere Monitoring Service's (CAMS); MPI: WRF–Chem executed by MPIM; EMEP: European Monitoring and Evaluation Programme; CHIM: CHIMERE transport model; SILAM: System for Integrated modeling of Atmospheric composition; USP: WRF–Chem executed by University of São Paulo.

**Table A4.** CO model evaluation scores (January and July).

| CO | City | Ensemble | | Mean | | CAMS | | MPI | | EMEP | | CHIM | | SILAM | | USP | |
|---|---|---|---|---|---|---|---|---|---|---|---|---|---|---|---|---|---|
| | | Jan | Jul | Jan | Jul | Jan | Jul | Jan | Jul | Jan | Jul | Jan | Jul | Jan | Jul | Jan | Jul |
| Model / observations | Santiago | 1.31 | 0.55 | 1.62 | 0.98 | 1.05 | 0.97 | 1.14 | 0.38 | 1.86 | 0.89 | 1.70 | 0.32 | 3.47 | 2.94 | 0.54 | 0.12 |
| | Bogotá | 0.36 | 0.41 | 0.40 | 0.45 | 0.38 | 0.33 | 0.42 | 0.52 | 0.52 | 0.60 | 0.31 | 0.25 | 0.51 | 0.50 | 0.21 | 0.40 |
| | Mexico | 1.14 | 1.18 | 1.45 | 1.61 | 0.60 | 0.48 | 0.94 | 1.01 | 2.28 | 3.48 | 2.40 | 1.99 | 1.16 | 1.28 | | |
| | São Paulo | 0.87 | 0.80 | 0.89 | 0.85 | 0.42 | 0.48 | 1.11 | 1.08 | 1.23 | 1.08 | 0.94 | 0.81 | 1.26 | 1.19 | 0.36 | 0.44 |
| MNBIAS [%] | Santiago | 26.5 | −51.4 | 47.6 | 3.4 | 2.2 | −0.8 | 12.6 | −81.4 | 59.4 | −6.1 | 52.4 | −108 | 109 | 98.5 | −60.3 | −151 |
| | Bogotá | −90.0 | −83.3 | −83.1 | −76.1 | −84.8 | −99.9 | −78.0 | −62.8 | −60.2 | −49.6 | −99.7 | −118 | −63.0 | −68.3 | −130 | −87.3 |
| | Mexico | 10.4 | 17.8 | 32.6 | 48.2 | −49.8 | −69.0 | −5.8 | 3.9 | 73.9 | 108 | 72.3 | 59.0 | 9.8 | 21.6 | | |
| | São Paulo | −15.0 | −22.8 | −12.7 | −17.2 | −84.7 | −73.3 | 10.4 | 6.1 | 18.9 | 1.8 | −8.1 | −15.3 | 17.2 | 8.9 | −94.2 | −76.2 |
| RMSE [ppm] | Santiago | 0.08 | 0.64 | 0.15 | 0.38 | 0.06 | 0.40 | 0.05 | 0.80 | 0.22 | 0.40 | 0.18 | 0.96 | 0.61 | 2.38 | 0.11 | 1.08 |
| | Bogotá | 0.47 | 0.38 | 0.45 | 0.36 | 0.47 | 0.43 | 0.43 | 0.31 | 0.37 | 0.27 | 0.49 | 0.48 | 0.39 | 0.33 | 0.58 | 0.39 |
| | Mexico | 0.18 | 0.15 | 0.45 | 0.43 | 0.34 | 0.38 | 0.14 | 0.16 | 1.20 | 1.77 | 1.40 | 0.67 | 0.23 | 0.26 | | |
| | São Paulo | 0.17 | 0.27 | 0.17 | 0.26 | 0.39 | 0.44 | 0.18 | 0.28 | 0.27 | 0.38 | 0.25 | 0.29 | 0.31 | 0.43 | 0.43 | 0.50 |
| FGE | Santiago | 0.26 | 0.59 | 0.48 | 0.30 | 0.18 | 0.27 | 0.15 | 0.85 | 0.59 | 0.32 | 0.52 | 1.08 | 1.09 | 0.98 | 0.60 | 1.52 |
| | Bogotá | 0.90 | 0.83 | 0.83 | 0.76 | 0.85 | 1.00 | 0.78 | 0.63 | 0.60 | 0.50 | 1.00 | 1.19 | 0.63 | 0.69 | 1.31 | 0.87 |
| | Mexico | 0.16 | 0.19 | 0.34 | 0.48 | 0.50 | 0.69 | 0.16 | 0.19 | 0.74 | 1.09 | 0.72 | 0.59 | 0.21 | 0.24 | | |
| | São Paulo | 0.26 | 0.32 | 0.24 | 0.31 | 0.85 | 0.74 | 0.23 | 0.30 | 0.28 | 0.36 | 0.27 | 0.30 | 0.30 | 0.41 | 0.94 | 0.76 |
| R | Santiago | 0.36 | 0.20 | 0.22 | 0.28 | 0.54 | 0.38 | 0.17 | 0.32 | −0.04 | 0.21 | 0.30 | −0.00 | −0.16 | 0.24 | 0.50 | 0.02 |
| | Bogotá | 0.72 | 0.27 | 0.71 | 0.28 | 0.07 | 0.18 | 0.77 | 0.36 | 0.60 | 0.32 | 0.42 | 0.22 | 0.29 | 0.19 | 0.67 | 0.11 |
| | Mexico | 0.85 | 0.87 | 0.73 | 0.85 | 0.78 | 0.68 | 0.75 | 0.49 | 0.59 | 0.65 | 0.74 | −0.04 | 0.85 | 0.82 | | |
| | São Paulo | 0.50 | 0.48 | 0.49 | 0.48 | 0.57 | 0.62 | 0.46 | 0.44 | 0.31 | 0.37 | 0.36 | 0.07 | 0.54 | 0.45 | 0.09 | 0.15 |

* Ensemble: based on the median value of the models; mean: arithmetic mean of the models; CAMS: Copernicus Atmosphere Monitoring Service's (CAMS); MPI: WRF–Chem executed by MPIM; EMEP: European Monitoring and Evaluation Programme; CHIM: CHIMERE transport model; SILAM: System for Integrated modeling of Atmospheric composition; USP: WRF–Chem executed by University of São Paulo.

**Table A5.** SO$_2$ model evaluation scores (January and July).

| SO$_2$ | City | Ensemble | | Mean | | CAMS | | MPI | | EMEP | | CHIM | | SILAM | | USP | |
|---|---|---|---|---|---|---|---|---|---|---|---|---|---|---|---|---|---|
| | | Jan | Jul | Jan | Jul | Jan | Jul | Jan | Jul | Jan | Jul | Jan | Jul | Jan | Jul | Jan | Jul |
| Model / observations | Santiago | 3.38 | 4.82 | 4.69 | 5.82 | 11.64 | 8.99 | 6.87 | 8.52 | 1.59 | 2.09 | 1.93 | 3.65 | 5.55 | 10.93 | 0.01 | 0.01 |
| | Bogotá | 0.58 | 0.50 | 0.64 | 0.56 | 0.52 | 0.40 | 1.17 | 1.10 | 1.13 | 0.98 | 0.36 | 0.20 | 0.56 | 0.50 | 0.00 | 0.00 |
| | Mexico | 3.53 | 4.85 | 12.43 | 12.46 | 46.93 | 42.11 | 5.83 | 6.92 | 2.12 | 3.02 | 1.65 | 1.98 | 3.12 | 4.44 | | |
| | São Paulo | 6.27 | 5.69 | 8.13 | 8.01 | 3.08 | 5.73 | 21.01 | 22.41 | 7.46 | 5.68 | 4.70 | 4.64 | 11.57 | 8.43 | 0.01 | 0.01 |
| MNBIAS [%] | Santiago | 107 | 130 | 140 | 140 | 168 | 159 | 147 | 157 | 45.4 | 69.3 | 61.2 | 109 | 138 | 162 | −196 | −197 |
| | Bogotá | −50.2 | −63.5 | −42.4 | −53.3 | −58.8 | −79.9 | 16.0 | 12.9 | 7.2 | 0.4 | −89.2 | −127 | −55.8 | −62.0 | −199 | −199 |
| | Mexico | 111 | 131 | 169 | 170 | 191 | 190 | 141 | 149 | 76.2 | 101 | 57.0 | 69.8 | 102 | 124 | −196 | −196 |
| | São Paulo | 144 | 139 | 156 | 153 | 101 | 136 | 181 | 181 | 151 | 134 | 129 | 128 | 167 | 154 | −193 | −193 |
| RMSE [ppb] | Santiago | 3.58 | 7.05 | 5.46 | 8.76 | 15.67 | 14.51 | 8.79 | 13.64 | 0.94 | 2.11 | 1.53 | 4.85 | 6.79 | 19.21 | 1.46 | 1.82 |
| | Bogotá | 0.87 | 0.93 | 0.83 | 0.84 | 0.96 | 1.09 | 0.75 | 0.49 | 0.97 | 0.48 | 1.15 | 1.15 | 0.93 | 0.95 | 1.79 | 1.70 |
| | Mexico | 12.44 | 12.10 | 52.40 | 35.18 | 207.29 | 125.97 | 22.42 | 18.35 | 6.08 | 6.33 | 4.62 | 3.39 | 10.63 | 11.19 | | |
| | São Paulo | 5.26 | 6.17 | 7.04 | 9.28 | 2.11 | 6.34 | 19.66 | 28.76 | 6.49 | 6.66 | 3.71 | 5.16 | 10.68 | 10.32 | 1.01 | 1.33 |
| FGE | Santiago | 1.08 | 1.30 | 1.29 | 1.41 | 1.68 | 1.60 | 1.48 | 1.58 | 0.45 | 0.69 | 0.61 | 1.09 | 1.38 | 1.62 | 1.96 | 1.97 |
| | Bogotá | 0.52 | 0.64 | 0.49 | 0.53 | 0.59 | 0.80 | 0.31 | 0.24 | 0.38 | 0.21 | 0.89 | 1.28 | 0.60 | 0.62 | 1.99 | 1.99 |
| | Mexico | 1.14 | 1.31 | 1.69 | 1.70 | 1.91 | 1.91 | 1.42 | 1.49 | 0.82 | 1.01 | 0.73 | 0.70 | 1.06 | 1.24 | | |
| | São Paulo | 1.44 | 1.39 | 1.56 | 1.54 | 1.01 | 1.37 | 1.82 | 1.81 | 1.52 | 1.34 | 1.30 | 1.29 | 1.67 | 1.54 | 1.94 | 1.94 |
| R | Santiago | −0.09 | −0.17 | −0.13 | 0.02 | 0.00 | 0.54 | −0.23 | 0.16 | −0.26 | 0.00 | −0.43 | 0.25 | −0.06 | −0.20 | 0.29 | 0.37 |
| | Bogotá | 0.24 | 0.70 | 0.07 | 0.68 | 0.03 | 0.42 | −0.01 | 0.55 | −0.01 | 0.52 | 0.37 | 0.37 | 0.17 | 0.44 | 0.10 | 0.48 |
| | Mexico | 0.24 | 0.18 | 0.17 | 0.16 | 0.10 | −0.03 | 0.20 | −0.04 | 0.19 | 0.21 | 0.14 | 0.04 | 0.11 | 0.04 | | |
| | São Paulo | 0.36 | 0.50 | 0.55 | 0.54 | 0.39 | 0.54 | 0.61 | 0.47 | 0.30 | 0.36 | 0.57 | −0.06 | 0.56 | 0.46 | 0.11 | 0.05 |

\* Ensemble: based on the median value of the models; mean: arithmetic mean of the models; CAMS: Copernicus Atmosphere Monitoring Service's (CAMS); MPI: WRF–Chem executed by MPIM; EMEP: European Monitoring and Evaluation Programme; CHIM: CHIMERE transport model; SILAM: System for Integrated modeling of Atmospheric composition; USP: WRF–Chem executed by University of São Paulo.

**Table A6.** PM$_{2.5}$ model evaluation scores (January and July).

| PM$_{2.5}$ | City | Ensemble Jan | Ensemble Jul | Mean Jan | Mean Jul | CAMS Jan | CAMS Jul | MPI Jan | MPI Jul | EMEP Jan | EMEP Jul | CHIM Jan | CHIM Jul | SILAM Jan | SILAM Jul | USP Jan | USP Jul |
|---|---|---|---|---|---|---|---|---|---|---|---|---|---|---|---|---|---|
| Model / observations | Santiago | 0.61 | 0.40 | 0.78 | 1.45 | 1.15 | 1.74 | 0.66 | 0.39 | 0.20 | 0.34 | 0.62 | 0.17 | 1.66 | 5.55 | 0.34 | 0.10 |
| | Bogotá | 0.34 | 0.42 | 0.55 | 0.67 | 1.34 | 1.54 | 0.85 | 1.09 | 0.18 | 0.20 | 0.26 | 0.34 | 0.42 | 0.39 | 0.19 | 0.37 |
| | Mexico | 1.02 | 1.38 | 1.72 | 2.06 | 4.35 | 5.06 | 0.79 | 1.30 | 0.18 | 0.20 | 0.62 | 0.70 | 2.38 | 2.56 | | |
| | São Paulo | 1.90 | 1.88 | 2.23 | 2.14 | 1.60 | 2.46 | 2.33 | 2.23 | 1.79 | 1.54 | 2.33 | 2.11 | 4.44 | 3.63 | 0.79 | 0.90 |
| MNBIAS [%] | Santiago | −46.9 | −75.3 | −23.1 | 41.4 | 14.0 | 57.5 | −40.3 | −78.9 | −129 | −89.5 | −40.0 | −144 | 48.0 | 138 | −97.9 | −156 |
| | Bogotá | −95.8 | −81.4 | −53.2 | −38.3 | 32.9 | 41.0 | −15.0 | 1.8 | −139 | −133 | −109 | −95.8 | −79.9 | −90.8 | −135 | −93.6 |
| | Mexico | −5.0 | 29.8 | 53.4 | 69.5 | 125 | 133 | −18.0 | 26.9 | −138 | −130 | −46.6 | −34.8 | 69.6 | 79.1 | | |
| | São Paulo | 66.0 | 61.5 | 77.7 | 71.2 | 48.5 | 82.6 | 83.5 | 75.3 | 58.2 | 35.0 | 85.7 | 76.2 | 117 | 92.4 | −23.4 | −7.4 |
| RMSE [μg m$^{-2}$] TS5 | Santiago | 8.61 | 38.52 | 5.92 | 31.44 | 5.47 | 46.62 | 8.22 | 39.07 | 16.27 | 41.00 | 7.48 | 53.00 | 16.82 | 269.80 | 13.80 | 53.60 |
| | Bogotá | 13.64 | 8.02 | 9.98 | 5.17 | 9.25 | 8.64 | 6.84 | 7.28 | 16.60 | 10.97 | 13.77 | 8.83 | 12.42 | 8.50 | 16.47 | 8.87 |
| | Mexico | 7.72 | 12.00 | 18.45 | 23.41 | 83.04 | 87.94 | 7.63 | 9.43 | 21.60 | 17.48 | 9.59 | 9.89 | 39.20 | 39.23 | 10.30 | 10.15 |
| | São Paulo | 16.90 | 20.90 | 23.07 | 27.26 | 12.99 | 32.92 | 23.86 | 28.83 | 16.68 | 19.52 | 26.71 | 28.98 | 69.79 | 78.88 | | |
| FGE | Santiago | 0.47 | 0.87 | 0.25 | 0.42 | 0.20 | 0.57 | 0.41 | 0.90 | 1.30 | 0.94 | 0.40 | 1.44 | 0.48 | 1.38 | 0.98 | 1.57 |
| | Bogotá | 0.96 | 0.81 | 0.53 | 0.41 | 0.38 | 0.41 | 0.32 | 0.42 | 1.40 | 1.34 | 1.09 | 0.96 | 0.80 | 0.91 | 1.35 | 0.94 |
| | Mexico | 0.23 | 0.36 | 0.53 | 0.70 | 1.25 | 1.34 | 0.28 | 0.34 | 1.39 | 1.31 | 0.47 | 0.43 | 0.79 | 0.80 | | |
| | São Paulo | 0.66 | 0.63 | 0.78 | 0.72 | 0.50 | 0.84 | 0.84 | 0.76 | 0.65 | 0.55 | 0.86 | 0.80 | 1.17 | 0.94 | 0.51 | 0.40 |
| R | Santiago | 0.50 | −0.15 | 0.43 | 0.42 | 0.60 | 0.51 | 0.20 | 0.05 | 0.13 | 0.25 | 0.26 | 0.00 | 0.02 | 0.37 | 0.21 | 0.33 |
| | Bogotá | 0.67 | 0.47 | 0.69 | 0.05 | 0.30 | 0.17 | 0.41 | −0.42 | 0.77 | 0.38 | 0.44 | 0.37 | 0.43 | 0.43 | 0.61 | 0.16 |
| | Mexico | 0.83 | 0.35 | 0.89 | 0.66 | 0.69 | 0.62 | 0.84 | 0.11 | 0.74 | 0.32 | 0.75 | −0.19 | 0.83 | 0.63 | | |
| | São Paulo | 0.53 | 0.59 | 0.54 | 0.63 | 0.51 | 0.56 | 0.45 | 0.49 | 0.37 | 0.50 | 0.45 | −0.11 | 0.53 | 0.63 | 0.08 | 0.22 |

* Ensemble: based on the median value of the models; mean: arithmetic mean of the models; CAMS: Copernicus Atmosphere Monitoring Service's (CAMS); MPI: WRF–Chem executed by MPIM; EMEP: European Monitoring and Evaluation Programme; CHIM: CHIMERE transport model; SILAM: System for Integrated modeling of Atmospheric composition; USP: WRF–Chem executed by University of São Paulo.

**Table A7.** PM$_{10}$ model evaluation scores (January and July).

| PM$_{10}$ | City | Ensemble Jan | Ensemble Jul | Mean Jan | Mean Jul | CAMS Jan | CAMS Jul | MPI Jan | MPI Jul | EMEP Jan | EMEP Jul | CHIM Jan | CHIM Jul | SILAM Jan | SILAM Jul | USP Jan | USP Jul |
|---|---|---|---|---|---|---|---|---|---|---|---|---|---|---|---|---|---|
| Model / observations | Santiago | 0.31 | 0.35 | 0.41 | 1.08 | 0.57 | 1.43 | 0.31 | 0.24 | 0.29 | 0.39 | 0.13 | 0.16 | 0.88 | 3.89 | 0.12 | 0.06 |
| | Bogotá | 0.21 | 0.22 | 0.33 | 0.40 | 0.75 | 0.83 | 0.59 | 0.84 | 0.19 | 0.18 | 0.12 | 0.16 | 0.21 | 0.19 | 0.08 | 0.15 |
| | Mexico | 0.57 | 0.87 | 1.10 | 1.31 | 2.96 | 3.30 | 0.42 | 0.91 | 0.41 | 0.51 | 0.28 | 0.35 | 1.21 | 1.16 | | |
| | São Paulo | 1.46 | 1.53 | 1.76 | 1.80 | 1.28 | 2.02 | 1.52 | 1.54 | 2.04 | 1.93 | 1.54 | 1.66 | 3.61 | 3.08 | 0.45 | 0.54 |
| MNBIAS [%] | Santiago | −102 | −85.2 | −81.6 | 13.9 | −53.8 | 39.9 | −103 | −110 | −108 | −79.5 | −111 | −155 | −13.1 | 118 | −156 | −173 |
| | Bogotá | −130 | −127 | −98.4 | −82.9 | −25.6 | −18.4 | −52.3 | −26.6 | −136 | −136 | −153 | −141 | −130 | −137 | −170 | −147 |
| | Mexico | −62.9 | −15.4 | 6.2 | 27.0 | 96.5 | 105 | −79.5 | −11.0 | −89.3 | −65.6 | −115 | −92.6 | 4.2 | 7.2 | | |
| | São Paulo | 39.4 | 42.1 | 54.9 | 53.8 | 24.2 | 64.1 | 42.6 | 40.4 | 64.8 | 45.6 | 46.4 | 53.8 | 104 | 84.1 | −75.1 | −57.0 |
| RMSE [µg m$^{-2}$] TS6 | Santiago | 39.65 | 72.46 | 34.40 | 32.08 | 25.86 | 53.34 | 39.82 | 81.63 | 41.21 | 67.84 | 40.52 | 99.01 | 16.89 | 295.63 | 50.28 | 97.44 |
| | Bogotá | 39.06 | 27.77 | 33.59 | 21.92 | 16.96 | 10.48 | 23.52 | 18.66 | 39.84 | 29.01 | 40.77 | 29.11 | 39.54 | 28.84 | 45.34 | 30.23 |
| | Mexico | 25.44 | 12.05 | 18.74 | 18.74 | 105.46 | 110.21 | 32.93 | 13.46 | 32.46 | 25.05 | 36.82 | 29.92 | 24.00 | 22.35 | | |
| | São Paulo | 39.4 | 42.1 | 53.8 | 53.8 | 24.2 | 64.1 | 42.6 | 40.4 | 64.8 | 45.6 | 33.75 | 33.12 | 102.20 | 84.1 | 20.03 | 20.42 |
| FGE | Santiago | 1.03 | 0.93 | 0.82 | 0.30 | 0.54 | 0.43 | 1.04 | 1.17 | 0.83 | 0.83 | 1.12 | 1.55 | 0.27 | 1.19 | 1.56 | 1.73 |
| | Bogotá | 1.31 | 1.27 | 0.98 | 0.83 | 0.31 | 0.27 | 0.56 | 0.53 | 1.37 | 1.36 | 1.54 | 1.42 | 1.31 | 1.38 | 1.71 | 1.48 |
| | Mexico | 0.64 | 0.25 | 0.20 | 0.31 | 0.96 | 1.05 | 0.80 | 0.27 | 0.90 | 0.66 | 1.16 | 0.93 | 0.38 | 0.34 | | |
| | São Paulo | 0.41 | 0.47 | 0.55 | 0.57 | 0.34 | 0.67 | 0.48 | 0.51 | 0.66 | 0.62 | 0.48 | 0.58 | 1.04 | 0.87 | 0.78 | 0.65 |
| R | Santiago | 0.30 | −0.17 | 0.35 | 0.44 | 0.44 | 0.48 | 0.20 | −0.33 | −0.03 | 0.26 | 0.07 | 0.03 | 0.03 | 0.45 | 0.07 | 0.37 |
| | Bogotá | 0.72 | 0.37 | 0.65 | −0.00 | 0.33 | 0.20 | 0.33 | −0.17 | 0.74 | 0.05 | 0.45 | 0.42 | 0.32 | 0.24 | 0.65 | −0.02 |
| | Mexico | 0.69 | 0.46 | 0.77 | 0.37 | 0.58 | 0.14 | 0.36 | 0.20 | 0.64 | 0.08 | 0.66 | −0.24 | 0.76 | 0.37 | | |
| | São Paulo | 0.51 | 0.45 | 0.56 | 0.50 | 0.43 | 0.51 | 0.12 | 0.13 | 0.41 | 0.37 | 0.45 | −0.01 | 0.58 | 0.52 | 0.11 | 0.19 |

\* Ensemble: based on the median value of the models; mean: arithmetic mean of the models; CAMS: Copernicus Atmosphere Monitoring Service's (CAMS); MPI: WRF–Chem executed by MPIM; EMEP: European Monitoring and Evaluation Programme; CHIM: CHIMERE transport model; SILAM: System for Integrated modeling of Atmospheric composition; USP: WRF–Chem executed by University of São Paulo.

**Table A8.** Coefficient of variation (CV) per city during January and July.

| City | NO$_2$ | | O$_3$ | | CO | | SO$_2$ | | PM$_{2.5}$ | |
|------|------|------|------|------|------|------|------|------|------|------|
| Santiago | 22 % | 29 % | 13 % | 58 % | 10 % | 33 % | 15 % | 20 % | 22 % | 32 % |
| Bogotá | 22 % | 18 % | 23 % | 23 % | 29 % | 17 % | 28 % | 34 % | 33 % | 18 % |
| São Paulo | 35 % | 31 % | 37 % | 38 % | 26 % | 31 % | 37 % | 41 % | 49 % | 45 % |
| Mexico | 24 % | 23 % | 31 % | 20 % | 24 % | 26 % | 60 % | 30 % | 37 % | 24 % |

## Appendix B: Air quality observations

**Table B1.** Station availability and location for Mexico City.

| | | Obs | | Ensemble | | CAMS | | MPI | | EMEP | | CHIM | | SILAM | | USP | |
|---|---|---|---|---|---|---|---|---|---|---|---|---|---|---|---|---|---|
| | | Jan | Jul | Jan | Jul | Jan | Jul | Jan | Jul | Jan | Jul | Jan | Jul | Jan | Jul | Jan | Jul |
| **CO** | | | | | | | | | | | | | | | | | |
| Mexico | number of stations | 21 | 24 | 21 | 24 | 21 | 24 | 21 | 24 | 21 | 24 | 21 | 24 | 21 | 24 | | |
| | availability [%] | 96.67 | 100 | 100 | 100 | 100 | 100 | 100 | 100 | 100 | 100 | 83.33 | 66.67 | 100 | 100 | | |
| | CV | 0.24 | 0.26 | 0.29 | 0.22 | 0.26 | 0.20 | 0.24 | 0.15 | 0.36 | 0.29 | 0.50 | 0.16 | 0.35 | 0.35 | | |
| **NO$_2$** | | | | | | | | | | | | | | | | | |
| Mexico | number of stations | 24 | 24 | 24 | 24 | 24 | 24 | 24 | 24 | 24 | 24 | 24 | 24 | 24 | 24 | | |
| | availability [%] | 100 | 100 | 100 | 100 | 100 | 100 | 100 | 100 | 100 | 100 | 83.33 | 66.67 | 100 | 100 | | |
| | CV | 0.24 | 0.23 | 0.29 | 0.21 | 0.46 | 0.52 | 0.22 | 0.15 | 0.30 | 0.24 | 0.38 | 0.14 | 0.38 | 0.38 | | |
| **O$_3$** | | | | | | | | | | | | | | | | | |
| Mexico | number of stations | 21 | 29 | 28 | 29 | 28 | 29 | 28 | 29 | 28 | 29 | 28 | 29 | 28 | 29 | | |
| | availability [%] | 96.67 | 100 | 100 | 100 | 100 | 100 | 100 | 100 | 100 | 100 | 83.33 | 66.67 | 100 | 100 | | |
| | CV | 0.31 | 0.20 | 0.25 | 0.22 | 0.17 | 0.14 | 0.27 | 0.23 | 0.41 | 0.26 | 0.32 | 0.33 | 0.28 | 0.30 | | |
| **PM$_{10}$** | | | | | | | | | | | | | | | | | |
| Mexico | number of stations | 17 | 24 | 17 | 24 | 17 | 24 | 17 | 24 | 17 | 24 | 17 | 24 | 17 | 24 | | |
| | availability [%] | 96.67 | 100 | 100 | 100 | 100 | 100 | 100 | 100 | 100 | 100 | 83.33 | 66.67 | 100 | 100 | | |
| | CV | 0.27 | 0.20 | 0.55 | 0.27 | 0.28 | 0.23 | 0.24 | 0.26 | 0.48 | 0.31 | 0.52 | 0.25 | 0.48 | 0.43 | | |
| **PM$_{2.5}$** | | | | | | | | | | | | | | | | | |
| Mexico | number of stations | 14 | 16 | 14 | 16 | 14 | 16 | 14 | 16 | 14 | 16 | 14 | 16 | 14 | 16 | | |
| | availability [%] | 96.67 | 100 | 100 | 100 | 100 | 100 | 100 | 100 | 100 | 100 | 83.33 | 66.67 | 100 | 100 | | |
| | CV | 0.37 | 0.24 | 0.52 | 0.32 | 0.28 | 0.22 | 0.25 | 0.19 | 0.34 | 0.19 | 0.52 | 0.30 | 0.48 | 0.46 | | |
| **SO$_2$** | | | | | | | | | | | | | | | | | |
| Mexico | number of stations | 23 | 26 | 23 | 26 | 23 | 26 | 23 | 26 | 23 | 26 | 23 | 26 | 23 | 26 | | |
| | availability [%] | 96.67 | 100 | 100 | 100 | 100 | 100 | 100 | 100 | 100 | 100 | 83.33 | 66.67 | 100 | 100 | | |
| | CV | 0.60 | 0.30 | 0.35 | 0.20 | 0.25 | 0.09 | 0.26 | 0.15 | 0.30 | 0.14 | 0.38 | 0.18 | 0.35 | 0.28 | | |

The observation availability refers to the percentage of days in each period when at least one station records enough data to construct their daily average (minimum of 18 h). Additionally, only stations that maintain at least 75 % of daily availability throughout the entire period are considered (at least 23 d with 18 h minimum). The model availability refers to the percentage of days for which we have modeled data, with CHIMERE being the only one with missing days and USP missing information for Mexico, given their simulation domain did not include it.

**Table B2.** Station availability and location for Bogotá.

| | | Obs | | Ensemble | | CAMS | | MPI | | EMEP | | CHIM | | SILAM | | USP | |
|---|---|---|---|---|---|---|---|---|---|---|---|---|---|---|---|---|---|
| | | Jan | Jul | Jan | Jul | Jan | Jul | Jan | Jul | Jan | Jul | Jan | Jul | Jan | Jul | Jan | Jul |
| **CO** | | | | | | | | | | | | | | | | | |
| Bogotá | number of stations | 7 | 8 | 7 | 8 | 7 | 8 | 7 | 8 | 7 | 8 | 7 | 8 | 7 | 8 | 7 | 8 |
| | availability [%] | 100 | 100 | 100 | 100 | 100 | 100 | 100 | 100 | 100 | 100 | 83.33 | 66.67 | 100 | 100 | 100 | 100 |
| | CV | 0.29 | 0.17 | 0.16 | 0.17 | 0.10 | 0.13 | 0.18 | 0.14 | 0.26 | 0.17 | 0.18 | 0.14 | 0.25 | 0.31 | 0.33 | 0.30 |
| **NO$_2$** | | | | | | | | | | | | | | | | | |
| Bogotá | number of stations | 8 | 7 | 8 | 7 | 8 | 7 | 8 | 7 | 8 | 7 | 8 | 7 | 8 | 7 | 8 | 7 |
| | availability [%] | 100 | 100 | 100 | 100 | 100 | 100 | 100 | 100 | 100 | 100 | 83.33 | 66.67 | 100 | 100 | 100 | 100 |
| | CV | 0.22 | 0.18 | 0.24 | 0.26 | 0.42 | 0.57 | 0.21 | 0.14 | 0.30 | 0.15 | 0.28 | 0.39 | 0.28 | 0.24 | 0.37 | 0.29 |
| **O$_3$** | | | | | | | | | | | | | | | | | |
| Bogotá | number of stations | 10 | 11 | 10 | 11 | 10 | 11 | 10 | 11 | 10 | 11 | 10 | 11 | 10 | 11 | 10 | 11 |
| | availability [%] | 100 | 100 | 100 | 100 | 100 | 100 | 100 | 100 | 100 | 100 | 83.33 | 66.67 | 100 | 100 | 100 | 100 |
| | CV | 0.23 | 0.23 | 0.11 | 0.13 | 0.08 | 0.11 | 0.22 | 0.21 | 0.14 | 0.19 | 0.18 | 0.15 | 0.18 | 0.27 | 0.35 | 0.24 |
| **PM$_{10}$** | | | | | | | | | | | | | | | | | |
| Bogotá | number of stations | 10 | 10 | 10 | 10 | 10 | 10 | 10 | 10 | 10 | 10 | 10 | 10 | 10 | 10 | 10 | 10 |
| | availability [%] | 100 | 100 | 100 | 100 | 100 | 100 | 100 | 100 | 100 | 100 | 83.33 | 66.67 | 100 | 100 | 100 | 100 |
| | CV | 0.27 | 0.22 | 0.31 | 0.19 | 0.16 | 0.21 | 0.33 | 0.52 | 0.37 | 0.17 | 0.26 | 0.14 | 0.37 | 0.40 | 0.33 | 0.34 |
| **PM$_{2.5}$** | | | | | | | | | | | | | | | | | |
| Bogotá | number of stations | 9 | 10 | 9 | 10 | 9 | 10 | 9 | 10 | 9 | 10 | 9 | 10 | 9 | 10 | 9 | 10 |
| | availability [%] | 100 | 100 | 100 | 100 | 100 | 100 | 100 | 100 | 100 | 100 | 83.33 | 66.67 | 100 | 100 | 100 | 100 |
| | CV | 0.33 | 0.18 | 0.29 | 0.22 | 0.16 | 0.22 | 0.29 | 0.40 | 0.40 | 0.18 | 0.28 | 0.11 | 0.40 | 0.42 | 0.34 | 0.33 |
| **SO$_2$** | | | | | | | | | | | | | | | | | |
| Bogotá | number of stations | 7 | 6 | 7 | 6 | 7 | 6 | 7 | 6 | 7 | 6 | 7 | 6 | 7 | 6 | 7 | 6 |
| | availability [%] | 100 | 100 | 100 | 100 | 100 | 100 | 100 | 100 | 100 | 100 | 83.33 | 66.67 | 100 | 100 | 100 | 100 |
| | CV | 0.28 | 0.34 | 0.22 | 0.20 | 0.12 | 0.15 | 0.24 | 0.20 | 0.41 | 0.24 | 0.24 | 0.45 | 0.32 | 0.18 | 0.30 | 0.37 |

The observation availability refers to the percentage of days in each period when at least one station records enough data to construct their daily average (minimum of 18 h). Additionally, only stations that maintain at least 75 % of daily availability throughout the entire period are considered (at least 23 d with 18 h minimum). The model availability refers to the percentage of days for which we have modeled data, with CHIMERE being the only one with missing days.

**Table B3.** Station availability and location for Santiago.

| | | Obs | | Ensemble | | CAMS | | MPI | | EMEP | | CHIM | | SILAM | | USP | |
|---|---|---|---|---|---|---|---|---|---|---|---|---|---|---|---|---|---|
| | | Jan | Jul | Jan | Jul | Jan | Jul | Jan | Jul | Jan | Jul | Jan | Jul | Jan | Jul | Jan | Jul |
| CO | | | | | | | | | | | | | | | | | |
| Santiago | number of stations | 8 | 9 | 8 | 9 | 8 | 9 | 8 | 9 | 8 | 9 | 8 | 9 | 8 | 9 | 8 | 9 |
| | availability [%] | 100 | 100 | 100 | 100 | 100 | 100 | 100 | 100 | 100 | 100 | 83.33 | 66.67 | 100 | 100 | 100 | 100 |
| | CV | 0.10 | 0.33 | 0.10 | 0.19 | 0.27 | 0.29 | 0.13 | 0.10 | 0.13 | 0.16 | 0.12 | 0.29 | 0.17 | 0.25 | 0.13 | 0.20 |
| NO$_2$ | | | | | | | | | | | | | | | | | |
| Santiago | number of stations | 9 | 9 | 10 | 10 | 10 | 10 | 10 | 10 | 10 | 10 | 10 | 10 | 10 | 10 | 10 | 10 |
| | availability [%] | 100 | 100 | 100 | 100 | 100 | 100 | 100 | 100 | 100 | 100 | 83.33 | 66.67 | 100 | 100 | 100 | 100 |
| | CV | 0.22 | 0.29 | 0.11 | 0.16 | 0.36 | 0.32 | 0.17 | 0.14 | 0.14 | 0.13 | 0.11 | 0.28 | 0.17 | 0.26 | 0.21 | 0.27 |
| O$_3$ | | | | | | | | | | | | | | | | | |
| Santiago | number of stations | 9 | 9 | 9 | 9 | 9 | 9 | 9 | 9 | 9 | 9 | 9 | 9 | 9 | 9 | 9 | 9 |
| | availability [%] | 100 | 100 | 100 | 100 | 100 | 100 | 100 | 100 | 100 | 100 | 83.33 | 66.67 | 100 | 100 | 100 | 100 |
| | CV | 0.13 | 0.58 | 0.16 | 0.24 | 0.16 | 0.20 | 0.24 | 0.14 | 0.13 | 0.29 | 0.17 | 0.22 | 0.24 | 0.60 | 0.25 | 0.40 |
| PM$_{10}$ | | | | | | | | | | | | | | | | | |
| Santiago | number of stations | 10 | 10 | 10 | 10 | 10 | 10 | 10 | 10 | 10 | 10 | 10 | 10 | 10 | 10 | 10 | 10 |
| | availability [%] | 100 | 100 | 100 | 100 | 100 | 100 | 100 | 100 | 100 | 100 | 83.33 | 66.67 | 100 | 100 | 100 | 100 |
| | CV | 0.20 | 0.36 | 0.15 | 0.20 | 0.24 | 0.23 | 0.24 | 0.17 | 0.14 | 0.22 | 0.16 | 0.31 | 0.23 | 0.27 | 0.20 | 0.22 |
| PM$_{2.5}$ | | | | | | | | | | | | | | | | | |
| Santiago | number of stations | 10 | 10 | 10 | 10 | 10 | 10 | 10 | 10 | 10 | 10 | 10 | 10 | 10 | 10 | 10 | 10 |
| | availability [%] | 100 | 100 | 100 | 100 | 100 | 100 | 100 | 100 | 100 | 100 | 83.33 | 66.67 | 100 | 100 | 100 | 100 |
| | CV | 0.22 | 0.32 | 0.15 | 0.13 | 0.24 | 0.23 | 0.20 | 0.13 | 0.12 | 0.19 | 0.16 | 0.32 | 0.29 | 0.29 | 0.20 | 0.22 |
| SO$_2$ | | | | | | | | | | | | | | | | | |
| Santiago | number of stations | 4 | 4 | 4 | 4 | 4 | 4 | 4 | 4 | 4 | 4 | 4 | 4 | 4 | 4 | 4 | 4 |
| | availability [%] | 100 | 100 | 100 | 100 | 100 | 100 | 100 | 100 | 100 | 100 | 83.33 | 66.67 | 100 | 100 | 100 | 100 |
| | CV | 0.15 | 0.20 | 0.16 | 0.18 | 0.11 | 0.16 | 0.18 | 0.14 | 0.12 | 0.19 | 0.21 | 0.22 | 0.17 | 0.36 | 0.12 | 0.17 |

The observation availability refers to the percentage of days in each period when at least one station records enough data to construct their daily average (minimum of 18 h). Additionally, only stations that maintain at least 75 % of daily availability throughout the entire period are considered (at least 23 d with 18 h minimum). The model availability refers to the percentage of days for which we have modeled data, with CHIMERE being the only one with missing days.

**Table B4.** Station availability and location for São Paulo.

| | | Obs Jan | Obs Jul | Ensemble Jan | Ensemble Jul | CAMS Jan | CAMS Jul | MPI Jan | MPI Jul | EMEP Jan | EMEP Jul | CHIM Jan | CHIM Jul | SILAM Jan | SILAM Jul | USP Jan | USP Jul |
|---|---|---|---|---|---|---|---|---|---|---|---|---|---|---|---|---|---|
| CO | | | | | | | | | | | | | | | | | |
| São Paulo | number of stations | 17 | 17 | 17 | 17 | 17 | 17 | 17 | 17 | 17 | 17 | 17 | 17 | 17 | 17 | 17 | 17 |
| | availability [%] | 100 | 100 | 100 | 100 | 100 | 100 | 100 | 100 | 100 | 100 | 83.33 | 66.67 | 100 | 100 | 100 | 100 |
| | CV | 0.26 | 0.31 | 0.26 | 0.31 | 0.37 | 0.42 | 0.22 | 0.34 | 0.29 | 0.48 | 0.42 | 0.26 | 0.40 | 0.50 | 0.41 | 0.34 |
| $NO_2$ | | | | | | | | | | | | | | | | | |
| São Paulo | number of stations | 18 | 18 | 18 | 18 | 18 | 18 | 18 | 18 | 18 | 18 | 18 | 18 | 18 | 18 | 18 | 18 |
| | availability [%] | 100 | 100 | 100 | 100 | 100 | 100 | 100 | 100 | 100 | 100 | 83.33 | 66.67 | 100 | 100 | 100 | 100 |
| | CV | 0.35 | 0.31 | 0.28 | 0.27 | 1.13 | 0.87 | 0.27 | 0.33 | 0.26 | 0.36 | 0.34 | 0.20 | 0.45 | 0.53 | 0.49 | 0.45 |
| $O_3$ | | | | | | | | | | | | | | | | | |
| São Paulo | number of stations | 20 | 20 | 20 | 20 | 20 | 20 | 20 | 20 | 20 | 20 | 20 | 20 | 20 | 20 | 20 | 20 |
| | availability [%] | 100 | 100 | 100 | 100 | 100 | 100 | 100 | 100 | 100 | 100 | 83.33 | 66.67 | 100 | 100 | 100 | 100 |
| | CV | 0.37 | 0.38 | 0.35 | 0.44 | 0.32 | 0.36 | 0.33 | 0.48 | 0.35 | 0.57 | 0.38 | 0.43 | 0.47 | 0.85 | 0.43 | 0.41 |
| $PM_{10}$ | | | | | | | | | | | | | | | | | |
| São Paulo | number of stations | 23 | 22 | 23 | 22 | 23 | 22 | 23 | 22 | 23 | 22 | 23 | 22 | 23 | 22 | 23 | 22 |
| | availability [%] | 100 | 100 | 100 | 100 | 100 | 100 | 100 | 100 | 100 | 100 | 83.33 | 66.67 | 100 | 100 | 100 | 100 |
| | CV | 0.39 | 0.37 | 0.27 | 0.33 | 0.34 | 0.36 | 0.27 | 0.36 | 0.40 | 0.75 | 0.39 | 0.33 | 0.56 | 0.76 | 0.47 | 0.38 |
| $PM_{2.5}$ | | | | | | | | | | | | | | | | | |
| São Paulo | number of stations | 9 | 9 | 9 | 9 | 9 | 9 | 9 | 9 | 9 | 9 | 9 | 9 | 9 | 9 | 9 | 9 |
| | availability [%] | 100 | 100 | 100 | 100 | 100 | 100 | 100 | 100 | 100 | 100 | 83.33 | 66.67 | 100 | 100 | 100 | 100 |
| | CV | 0.49 | 0.45 | 0.29 | 0.37 | 0.34 | 0.36 | 0.26 | 0.38 | 0.35 | 0.60 | 0.40 | 0.32 | 0.61 | 0.88 | 0.47 | 0.37 |
| $SO_2$ | | | | | | | | | | | | | | | | | |
| São Paulo | number of stations | 8 | 8 | 8 | 8 | 8 | 8 | 8 | 8 | 8 | 8 | 8 | 8 | 8 | 8 | 8 | 8 |
| | availability [%] | 100 | 100 | 100 | 100 | 100 | 100 | 100 | 100 | 100 | 100 | 83.33 | 66.67 | 100 | 100 | 100 | 100 |
| | CV | 0.37 | 0.41 | 0.28 | 0.31 | 0.28 | 0.36 | 0.26 | 0.40 | 0.31 | 0.48 | 0.30 | 0.39 | 0.35 | 0.46 | 0.34 | 0.34 |

The observation availability refers to the percentage of days in each period when at least one station records enough data to construct their daily average (minimum of 18 h).
Additionally, only stations that maintain at least 75 % of daily availability throughout the entire period are considered (at least 23 d with 18 h minimum). The model availability refers to the percentage of days for which we have modeled data, with CHIMERE being the only one with missing days.

**Table B5.** Station availability and location for Quito

| | | Obs | | Ensemble | | CAMS | | MPI | | EMEP | | CHIM | | SILAM | | USP | |
|---|---|---|---|---|---|---|---|---|---|---|---|---|---|---|---|---|---|
| | | Jan | Jul | Jan | Jul | Jan | Jul | Jan | Jul | Jan | Jul | Jan | Jul | Jan | Jul | Jan | Jul |
| **CO** | | | | | | | | | | | | | | | | | |
| Quito | number of stations | 6 | 6 | 6 | 6 | 6 | 6 | 6 | 6 | 6 | 6 | 6 | 6 | 6 | 6 | 6 | 6 |
| | availability [%] | 100 | 100 | 100 | 100 | 100 | 100 | 100 | 100 | 100 | 100 | 83.33 | 66.67 | 100 | 100 | 100 | 100 |
| | CV | 0.17 | 0.16 | 0.15 | 0.15 | 0.07 | 0.06 | 0.14 | 0.12 | 0.31 | 0.32 | 0.18 | 0.12 | 0.15 | 0.16 | 0.34 | 0.26 |
| **NO$_2$** | | | | | | | | | | | | | | | | | |
| Quito | number of stations | 5 | 5 | 5 | 5 | 5 | 5 | 5 | 5 | 5 | 5 | 5 | 5 | 5 | 5 | 5 | 5 |
| | availability [%] | 100 | 100 | 100 | 100 | 100 | 100 | 100 | 100 | 100 | 100 | 83.33 | 66.67 | 100 | 100 | 100 | 100 |
| | CV | 0.24 | 0.18 | 0.16 | 0.28 | 0.49 | 0.46 | 0.20 | 0.16 | 0.26 | 0.31 | 0.22 | 0.27 | 0.18 | 0.18 | 0.37 | 0.32 |
| **O$_3$** | | | | | | | | | | | | | | | | | |
| Quito | number of stations | 7 | 7 | 7 | 7 | 7 | 7 | 7 | 7 | 7 | 7 | 7 | 7 | 7 | 7 | 7 | 7 |
| | availability [%] | 100 | 100 | 100 | 100 | 100 | 100 | 100 | 100 | 100 | 100 | 83.33 | 66.67 | 100 | 100 | 100 | 100 |
| | CV | 0.17 | 0.20 | 0.13 | 0.13 | 0.08 | 0.07 | 0.21 | 0.21 | 0.26 | 0.20 | 0.20 | 0.12 | 0.24 | 0.27 | 0.33 | 0.18 |
| **PM$_{10}$** | | | | | | | | | | | | | | | | | |
| Quito | number of stations | 3 | 3 | 3 | 3 | 3 | 3 | 3 | 3 | 3 | 3 | 3 | 3 | 3 | 3 | 3 | 3 |
| | availability [%] | 96.67 | 100 | 100 | 100 | 100 | 100 | 100 | 100 | 100 | 100 | 83.33 | 66.67 | 100 | 100 | 100 | 100 |
| | CV | 0.34 | 0.24 | 0.22 | 0.23 | 0.14 | 0.12 | 0.25 | 0.33 | 0.26 | 0.41 | 0.19 | 0.16 | 0.22 | 0.38 | 0.40 | 0.26 |
| **PM$_{2.5}$** | | | | | | | | | | | | | | | | | |
| Quito | number of stations | 5 | 5 | 5 | 5 | 5 | 5 | 5 | 5 | 5 | 5 | 5 | 5 | 5 | 5 | 5 | 5 |
| | availability [%] | 96.67 | 100 | 100 | 100 | 100 | 100 | 100 | 100 | 100 | 100 | 83.33 | 66.67 | 100 | 100 | 100 | 100 |
| | CV | 0.19 | 0.24 | 0.20 | 0.19 | 0.14 | 0.12 | 0.21 | 0.24 | 0.23 | 0.32 | 0.19 | 0.13 | 0.23 | 0.33 | 0.40 | 0.26 |
| **SO$_2$** | | | | | | | | | | | | | | | | | |
| Quito | number of stations | 7 | 7 | 7 | 7 | 7 | 7 | 7 | 7 | 7 | 7 | 7 | 7 | 7 | 7 | 7 | 7 |
| | availability [%] | 100 | 100 | 100 | 100 | 100 | 100 | 100 | 100 | 100 | 100 | 83.33 | 66.67 | 100 | 100 | 100 | 100 |
| | CV | 0.33 | 0.35 | 0.21 | 0.33 | 0.09 | 0.08 | 0.19 | 0.17 | 0.19 | 0.27 | 0.19 | 0.25 | 0.12 | 0.19 | 0.34 | 0.34 |

The observation availability refers to the percentage of days in each period when at least one station records enough data to construct their daily average (minimum of 18 h). Additionally, only stations that maintain at least 75 % of daily availability throughout the entire period are considered (at least 23 d with 18 h minimum). The model availability refers to the percentage of days for which we have modeled data, with CHIMERE being the only one with missing days.

**Table B6.** Station availability and location for Medellín.

| | | Obs Jan | Obs Jul | Ensemble Jan | Ensemble Jul | CAMS Jan | CAMS Jul | MPI Jan | MPI Jul | EMEP Jan | EMEP Jul | CHIM Jan | CHIM Jul | SILAM Jan | SILAM Jul | USP Jan | USP Jul |
|---|---|---|---|---|---|---|---|---|---|---|---|---|---|---|---|---|---|
| **CO** | | | | | | | | | | | | | | | | | |
| Medellín | number of stations | 2 | 2 | 2 | 2 | 2 | 2 | 2 | 2 | 2 | 2 | 2 | 2 | 2 | 2 | 2 | 2 |
| | availability [%] | 76.67 | 100 | 100 | 100 | 100 | 100 | 100 | 100 | 100 | 100 | 83.33 | 66.67 | 100 | 100 | 100 | 100 |
| | CV | 0.14 | 0.11 | 0.10 | 0.09 | 0.08 | 0.13 | 0.13 | 0.13 | 0.34 | 0.30 | 0.19 | 0.09 | 0.14 | 0.12 | 0.18 | 0.14 |
| **$NO_2$** | | | | | | | | | | | | | | | | | |
| Medellín | number of stations | 4 | 4 | 4 | 4 | 4 | 4 | 4 | 4 | 4 | 4 | 4 | 4 | 4 | 4 | 4 | 4 |
| | availability [%] | 100 | 100 | 100 | 100 | 100 | 100 | 100 | 100 | 100 | 100 | 83.33 | 66.67 | 100 | 100 | 100 | 100 |
| | CV | 0.19 | 0.16 | 0.13 | 0.17 | 0.68 | 0.70 | 0.16 | 0.15 | 0.37 | 0.25 | 0.25 | 0.18 | 0.24 | 0.16 | 0.20 | 0.20 |
| **$O_3$** | | | | | | | | | | | | | | | | | |
| Medellín | number of stations | 4 | 3 | 4 | 3 | 4 | 3 | 4 | 3 | 4 | 3 | 4 | 3 | 4 | 3 | 4 | 3 |
| | availability [%] | 100 | 100 | 100 | 100 | 100 | 100 | 100 | 100 | 100 | 100 | 83.33 | 66.67 | 100 | 100 | 100 | 100 |
| | CV | 0.21 | 0.17 | 0.13 | 0.10 | 0.12 | 0.14 | 0.13 | 0.18 | 0.17 | 0.21 | 0.11 | 0.11 | 0.18 | 0.26 | 0.30 | 0.17 |
| **$PM_{10}$** | | | | | | | | | | | | | | | | | |
| Medellín | number of stations | 1 | 1 | 1 | 1 | 1 | 1 | 1 | 1 | 1 | 1 | 1 | 1 | 1 | 1 | 1 | 1 |
| | availability [%] | 100 | 100 | 100 | 100 | 100 | 100 | 100 | 100 | 100 | 100 | 83.33 | 66.67 | 100 | 100 | 100 | 100 |
| | CV | 0.16 | 0.20 | 0.14 | 0.15 | 0.17 | 0.29 | 0.32 | 0.39 | 0.22 | 0.33 | 0.13 | 0.16 | 0.22 | 0.18 | 0.31 | 0.37 |
| **$PM_{2.5}$** | | | | | | | | | | | | | | | | | |
| Medellín | number of stations | 5 | 5 | 5 | 5 | 5 | 5 | 5 | 5 | 5 | 5 | 5 | 5 | 5 | 5 | 5 | 5 |
| | availability [%] | 100 | 100 | 100 | 100 | 100 | 100 | 100 | 100 | 100 | 100 | 83.33 | 66.67 | 100 | 100 | 100 | 100 |
| | CV | 0.15 | 0.14 | 0.14 | 0.15 | 0.18 | 0.30 | 0.27 | 0.30 | 0.18 | 0.25 | 0.14 | 0.13 | 0.22 | 0.20 | 0.27 | 0.35 |
| **$SO_2$** | | | | | | | | | | | | | | | | | |
| Medellín | number of stations | 1 | 1 | 1 | 1 | 1 | 1 | 1 | 1 | 1 | 1 | 1 | 1 | 1 | 1 | 1 | 1 |
| | availability [%] | 10 | 100 | 100 | 100 | 100 | 100 | 100 | 100 | 100 | 100 | 83.33 | 66.67 | 100 | 100 | 100 | 100 |
| | CV | 0.20 | 0.19 | 0.26 | 0.20 | 0.20 | 0.23 | 0.28 | 0.22 | 0.74 | 0.52 | 0.17 | 0.15 | 0.27 | 0.31 | 0.22 | 0.25 |

The observation availability refers to the percentage of days in each period when at least one station records enough data to construct their daily average (minimum of 18 h).
Additionally, only stations that maintain at least 75 % of daily availability throughout the entire period are considered (at least 23 d with 18 h minimum). The model availability refers to the percentage of days for which we have modeled data, with CHIMERE being the only one with missing days.

**Table B7.** Station availability and location for Lima.

| | | Obs | | Ensemble | | CAMS | | MPI | | EMEP | | CHIM | | SILAM | | USP | |
|---|---|---|---|---|---|---|---|---|---|---|---|---|---|---|---|---|---|
| | | Jan | Jul | Jan | Jul | Jan | Jul | Jan | Jul | Jan | Jul | Jan | Jul | Jan | Jul | Jan | Jul |
| **CO** | | | | | | | | | | | | | | | | | |
| Lima | number of stations | 8 | 7 | 8 | 7 | 8 | 7 | 8 | 7 | 8 | 7 | 8 | 7 | 8 | 7 | 8 | 7 |
| | availability [%] | 100 | 100 | 100 | 100 | 100 | 100 | 100 | 100 | 100 | 100 | 83.33 | 66.67 | 100 | 100 | 100 | 100 |
| | CV | 0.21 | 0.17 | 0.08 | 0.09 | 0.08 | 0.10 | 0.11 | 0.09 | 0.10 | 0.08 | 0.36 | 0.13 | 0.33 | 0.19 | 0.20 | 0.14 |
| **$NO_2$** | | | | | | | | | | | | | | | | | |
| Lima | number of stations | 2 | 6 | 2 | 6 | 2 | 6 | 2 | 6 | 2 | 6 | 2 | 6 | 2 | 6 | 2 | 6 |
| | availability [%] | 93.33 | 96.67 | 100 | 100 | 100 | 100 | 100 | 100 | 100 | 100 | 83.33 | 66.67 | 100 | 100 | 100 | 100 |
| | CV | 0.23 | 0.16 | 0.10 | 0.13 | 0.38 | 0.70 | 0.08 | 0.11 | 0.14 | 0.10 | 0.25 | 0.12 | 0.20 | 0.19 | 0.20 | 0.19 |
| **$O_3$** | | | | | | | | | | | | | | | | | |
| Lima | number of stations | 3 | 6 | 3 | 6 | 3 | 6 | 3 | 6 | 3 | 6 | 3 | 6 | 3 | 6 | 3 | 6 |
| | availability [%] | 100 | 100 | 100 | 100 | 100 | 100 | 100 | 100 | 100 | 100 | 83.33 | 66.67 | 100 | 100 | 100 | 100 |
| | CV | 0.47 | 0.29 | 0.14 | 0.13 | 0.18 | 0.15 | 0.17 | 0.08 | 0.12 | 0.10 | 0.12 | 0.12 | 0.19 | 0.18 | 0.31 | 0.25 |
| **$PM_{10}$** | | | | | | | | | | | | | | | | | |
| Lima | number of stations | 8 | 8 | 8 | 8 | 8 | 8 | 8 | 8 | 8 | 8 | 8 | 8 | 8 | 8 | 8 | 8 |
| | availability [%] | 100 | 100 | 100 | 100 | 100 | 100 | 100 | 100 | 100 | 100 | 83.33 | 66.67 | 100 | 100 | 100 | 100 |
| | CV | 0.13 | 0.27 | 0.09 | 0.12 | 0.13 | 0.18 | 0.16 | 0.19 | 0.24 | 0.23 | 0.11 | 0.11 | 0.20 | 0.18 | 0.16 | 0.23 |
| **$PM_{2.5}$** | | | | | | | | | | | | | | | | | |
| Lima | number of stations | 9 | 8 | 9 | 8 | 9 | 8 | 9 | 8 | 9 | 8 | 9 | 8 | 9 | 8 | 9 | 8 |
| | availability [%] | 96.67 | 100 | 100 | 100 | 100 | 100 | 100 | 100 | 100 | 100 | 83.33 | 66.67 | 100 | 100 | 100 | 100 |
| | CV | 0.16 | 0.22 | 0.12 | 0.13 | 0.13 | 0.18 | 0.18 | 0.14 | 0.20 | 0.11 | 0.18 | 0.13 | 0.28 | 0.23 | 0.16 | 0.23 |
| **$SO_2$** | | | | | | | | | | | | | | | | | |
| Lima | number of stations | 4 | 2 | 4 | 2 | 4 | 2 | 4 | 2 | 4 | 2 | 4 | 2 | 4 | 2 | 4 | 2 |
| | availability [%] | 100 | 100 | 100 | 100 | 100 | 100 | 100 | 100 | 100 | 100 | 83.33 | 66.67 | 100 | 100 | 100 | 100 |
| | CV | 0.19 | 0.37 | 0.15 | 0.11 | 0.12 | 0.05 | 0.10 | 0.12 | 0.19 | 0.10 | 0.22 | 0.21 | 0.24 | 0.16 | 0.19 | 0.15 |

The observation availability refers to the percentage of days in each period when at least one station records enough data to construct their daily average (minimum of 18 h).
Additionally, only stations that maintain at least 75 % of daily availability throughout the entire period are considered (at least 23 d with 18 h minimum). The model availability refers to the percentage of days for which we have modeled data, with CHIMERE being the only one with missing days.

**Table B8.** Station availability and location for Guadalajara.

| | | Obs Jan | Obs Jul | Ensemble Jan | Ensemble Jul | CAMS Jan | CAMS Jul | MPI Jan | MPI Jul | EMEP Jan | EMEP Jul | CHIM Jan | CHIM Jul | SILAM Jan | SILAM Jul | USP Jan | USP Jul |
|---|---|---|---|---|---|---|---|---|---|---|---|---|---|---|---|---|---|
| **CO** | | | | | | | | | | | | | | | | | |
| Guadalajara | number of stations | 7 | 7 | 7 | 7 | 7 | 7 | 7 | 7 | 7 | 7 | 7 | 7 | 7 | 7 | | |
| | availability [%] | 96.67 | 100 | 100 | 100 | 100 | 100 | 100 | 100 | 100 | 100 | 83.33 | 66.67 | 100 | 100 | | |
| | CV | 0.20 | 0.12 | 0.22 | 0.14 | 0.20 | 0.13 | 0.14 | 0.18 | 0.28 | 0.20 | 0.32 | 0.16 | 0.30 | 0.27 | | |
| **NO$_2$** | | | | | | | | | | | | | | | | | |
| Guadalajara | number of stations | 8 | 8 | 8 | 8 | 8 | 8 | 8 | 8 | 8 | 8 | 8 | 8 | 8 | 8 | | |
| | availability [%] | 100 | 100 | 100 | 100 | 100 | 100 | 100 | 100 | 100 | 100 | 83.33 | 66.67 | 100 | 100 | | |
| | CV | 0.25 | 0.17 | 0.30 | 0.17 | 0.64 | 0.88 | 0.16 | 0.18 | 0.34 | 0.24 | 0.39 | 0.28 | 0.32 | 0.33 | | |
| **O$_3$** | | | | | | | | | | | | | | | | | |
| Guadalajara | number of stations | 8 | 8 | 8 | 8 | 8 | 8 | 8 | 8 | 8 | 8 | 8 | 8 | 8 | 8 | | |
| | availability [%] | 96.67 | 100 | 100 | 100 | 100 | 100 | 100 | 100 | 100 | 100 | 83.33 | 66.67 | 100 | 100 | | |
| | CV | 0.16 | 0.31 | 0.16 | 0.16 | 0.28 | 0.24 | 0.22 | 0.27 | 0.22 | 0.16 | 0.18 | 0.12 | 0.26 | 0.27 | | |
| **PM$_{10}$** | | | | | | | | | | | | | | | | | |
| Guadalajara | number of stations | 8 | 8 | 8 | 8 | 8 | 8 | 8 | 8 | 8 | 8 | 8 | 8 | 8 | 8 | | |
| | availability [%] | 96.67 | 100 | 100 | 100 | 100 | 100 | 100 | 100 | 100 | 100 | 83.33 | 66.67 | 100 | 100 | | |
| | CV | 0.22 | 0.20 | 0.32 | 0.20 | 0.28 | 0.29 | 0.34 | 0.47 | 0.34 | 0.25 | 0.31 | 0.19 | 0.36 | 0.37 | | |
| **SO$_2$** | | | | | | | | | | | | | | | | | |
| Guadalajara | number of stations | 8 | 8 | 8 | 8 | 8 | 8 | 8 | 8 | 8 | 8 | 8 | 8 | 8 | 8 | | |
| | availability [%] | 96.67 | 100 | 100 | 100 | 100 | 100 | 100 | 100 | 100 | 100 | 83.33 | 66.67 | 100 | 100 | | |
| | CV | 0.52 | 0.41 | 0.33 | 0.12 | 0.26 | 0.17 | 0.20 | 0.21 | 0.26 | 0.16 | 0.28 | 0.23 | 0.30 | 0.28 | | |

The observation availability refers to the percentage of days in each period when at least one station records enough data to construct their daily average (minimum of 18 h). Additionally, only stations that maintain at least 75 % of daily availability throughout the entire period are considered (at least 23 d with 18 h minimum). The model availability refers to the percentage of days for which we have modeled data, with CHIMERE being the only one with missing days and USP missing information for Guadalajara, given their simulation domain did not include it.

**Appendix C:  Particular hourly simulations**

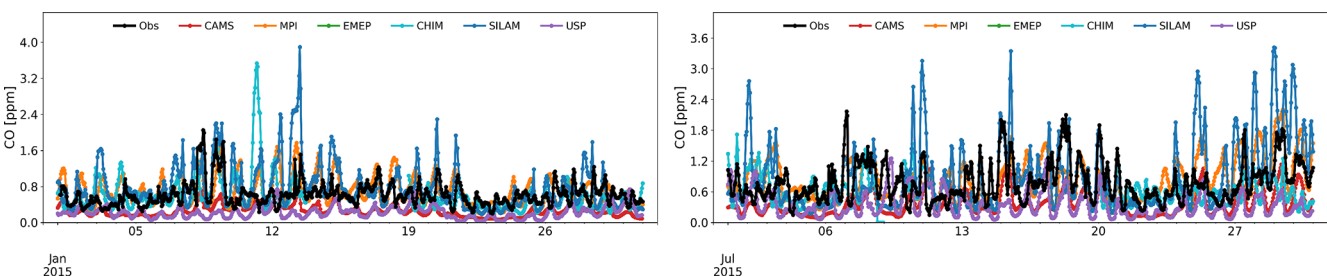

**Figure C1.** Hourly CO simulations in São Paulo for January and July 2015.

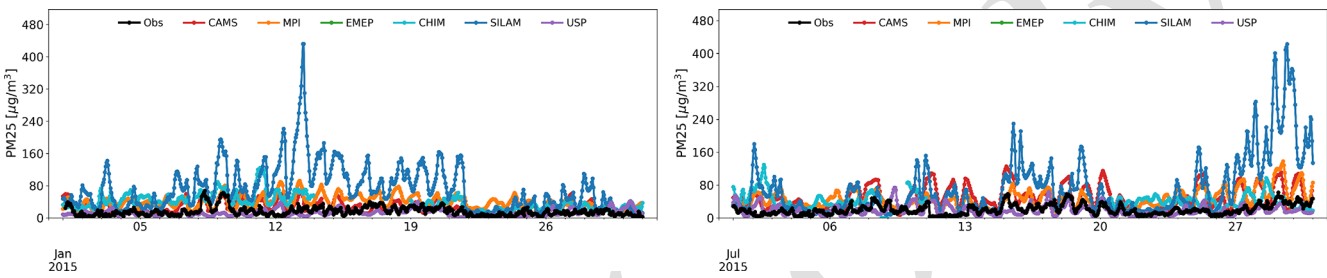

**Figure C2.** Hourly PM$_{2.5}$ simulations in São Paulo for January and July 2015.

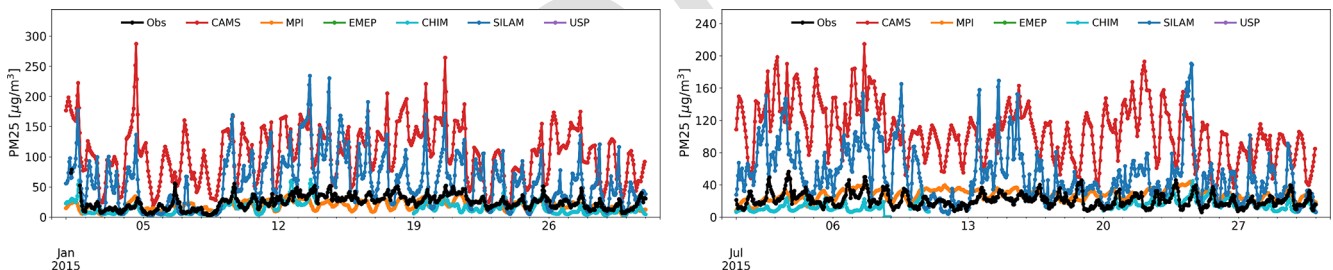

**Figure C3.** Hourly PM$_{2.5}$ simulations in Mexico City for January and July 2015.

**Appendix D: Simulation of all models**

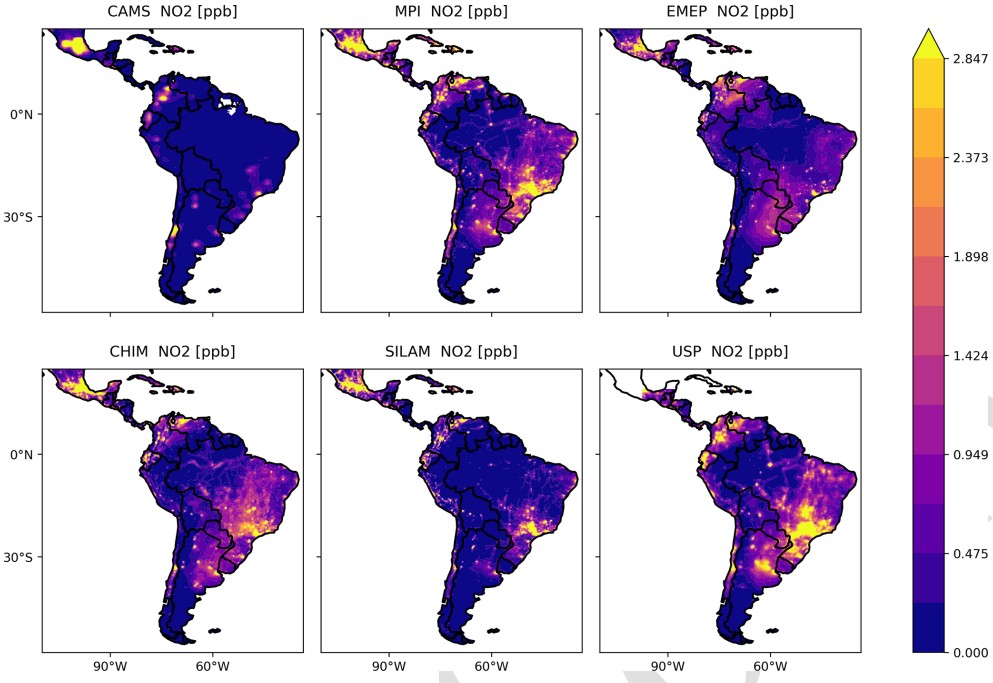

**Figure D1.** NO$_2$ simulations of January 2015 for all models.

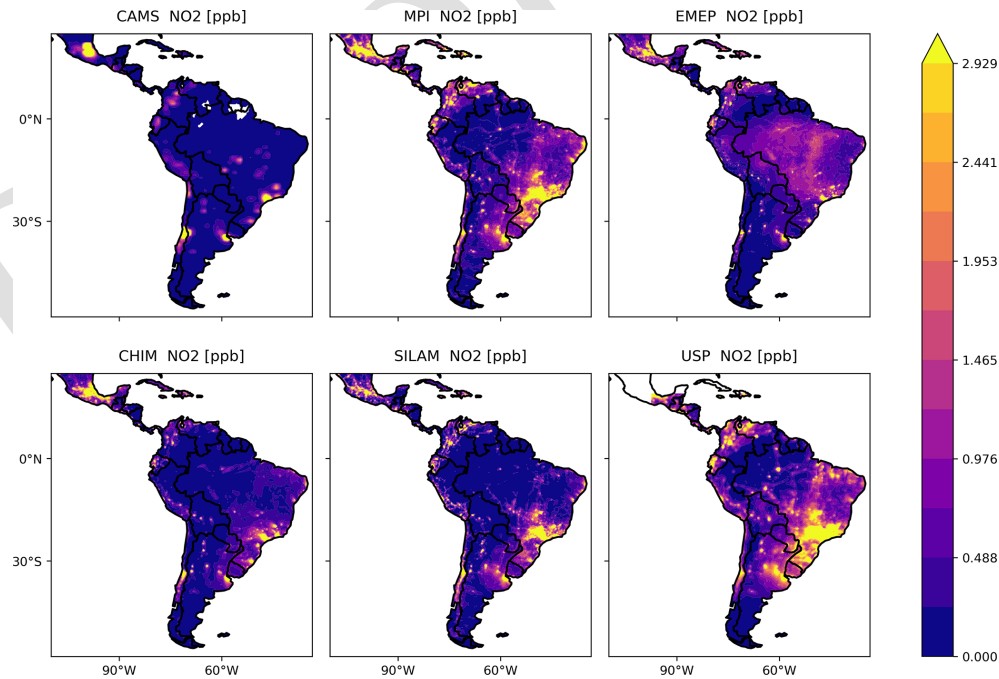

**Figure D2.** NO$_2$ simulations of July 2015 for all models.

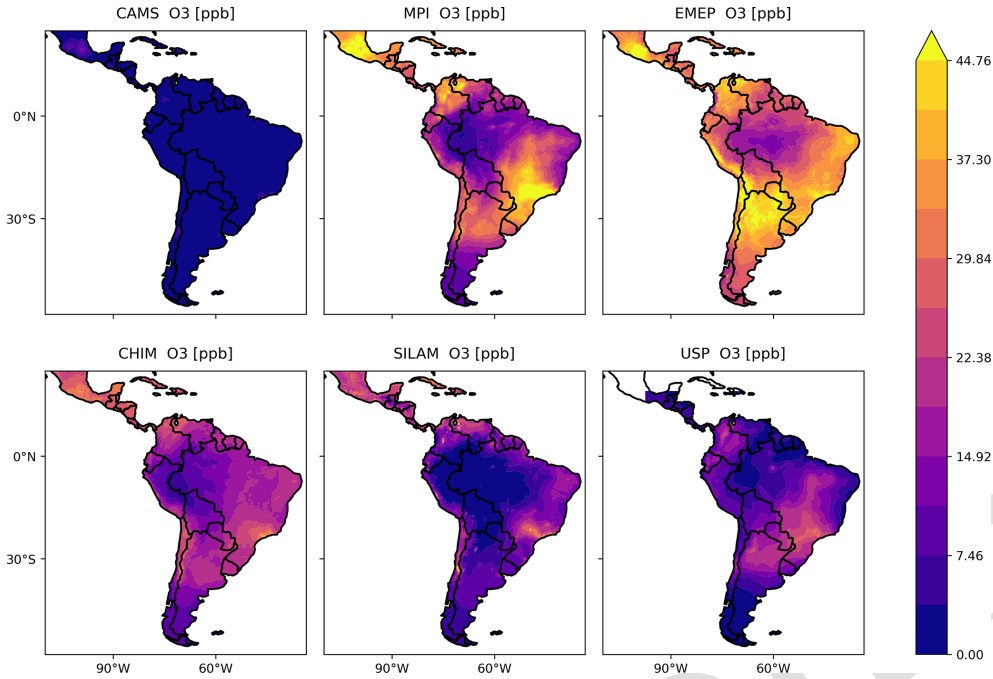

**Figure D3.** O$_3$ simulations of January 2015 for all models.

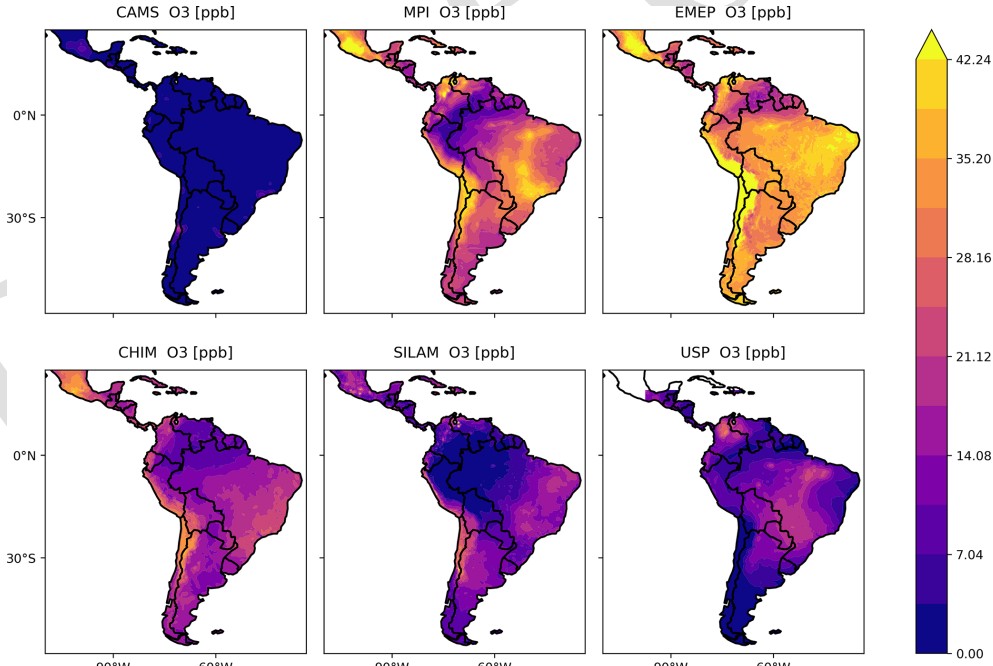

**Figure D4.** O$_3$ simulations of July 2015 for all models.

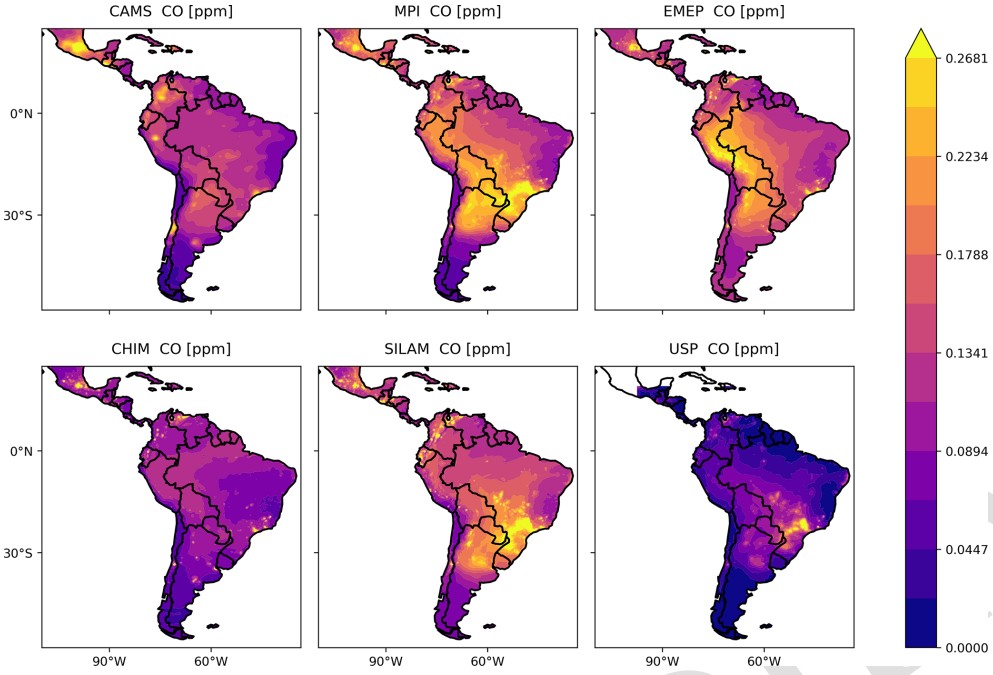

**Figure D5.** CO simulations of January 2015 for all models.

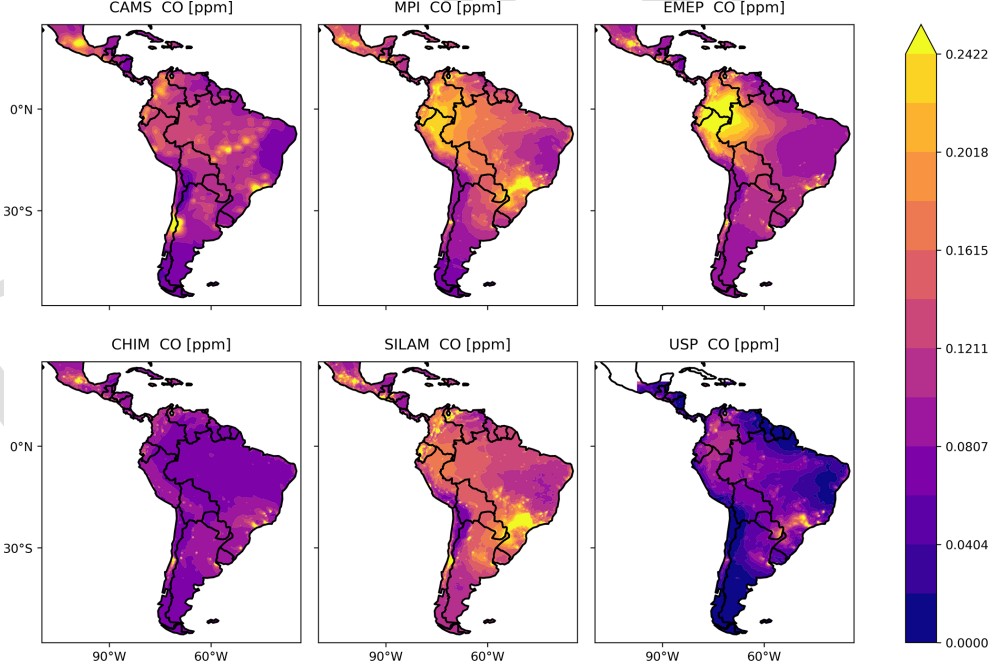

**Figure D6.** CO simulations of July 2015 for all models.

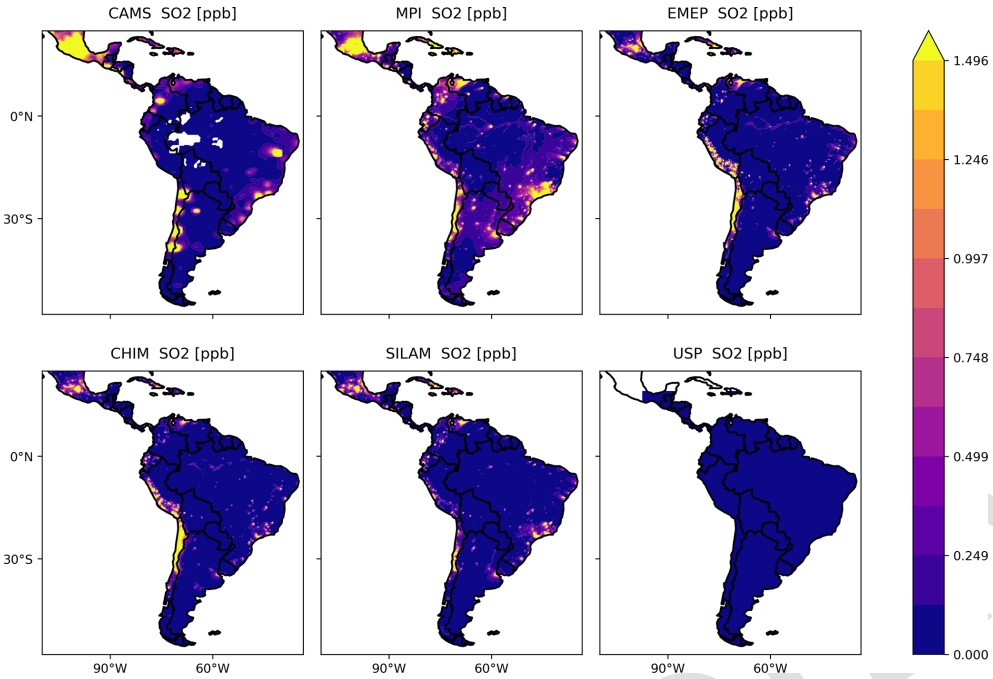

**Figure D7.** SO$_2$ simulations of January 2015 for all models.

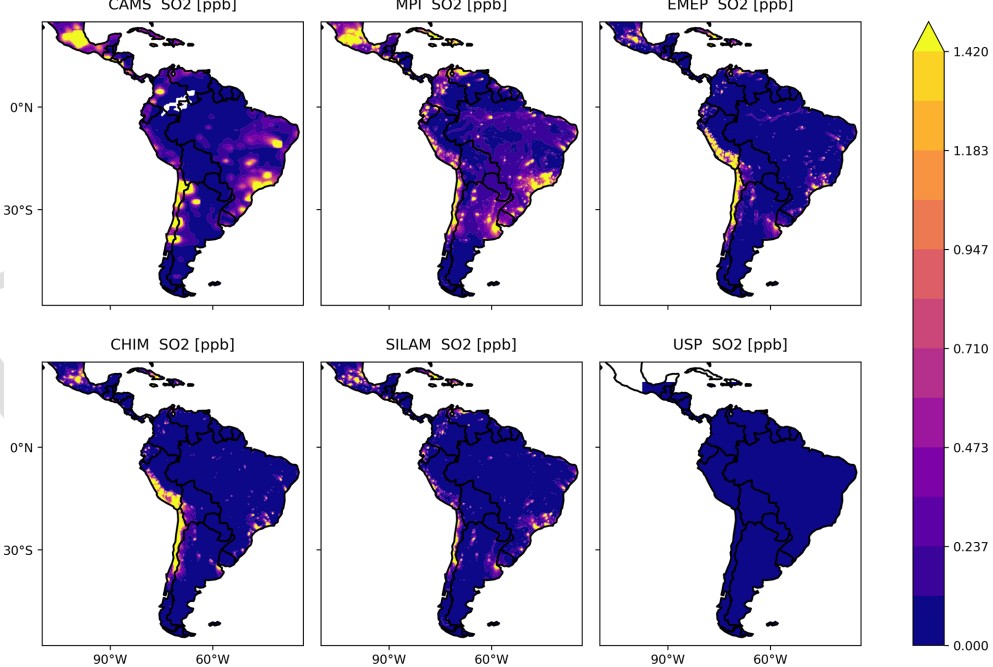

**Figure D8.** SO$_2$ simulations of July 2015 for all models.

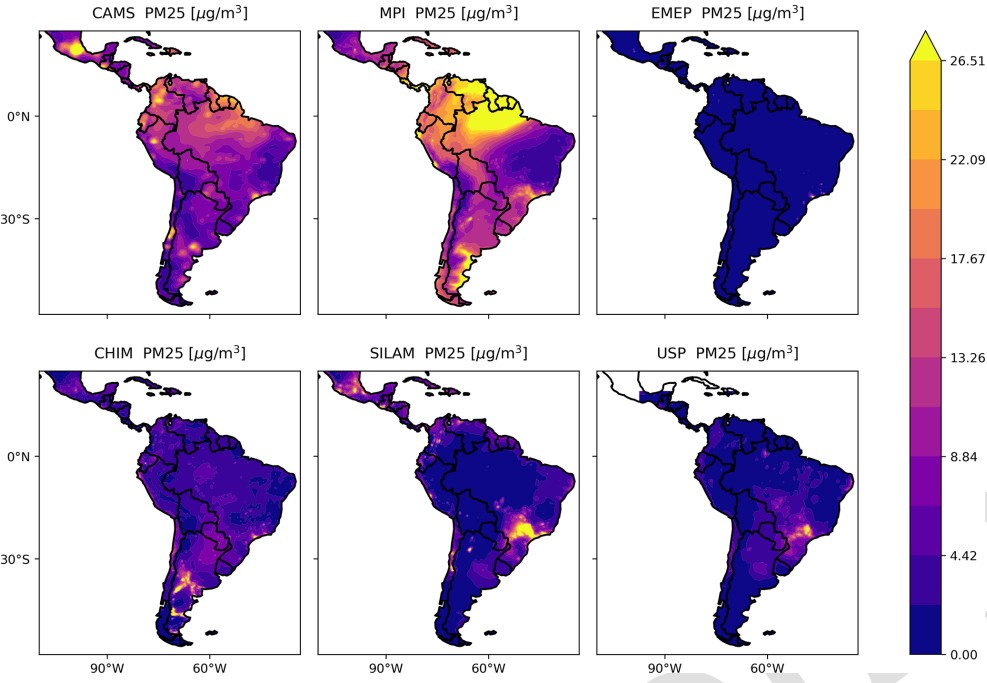

**Figure D9.** PM$_{2.5}$ simulations of January 2015 for all models.

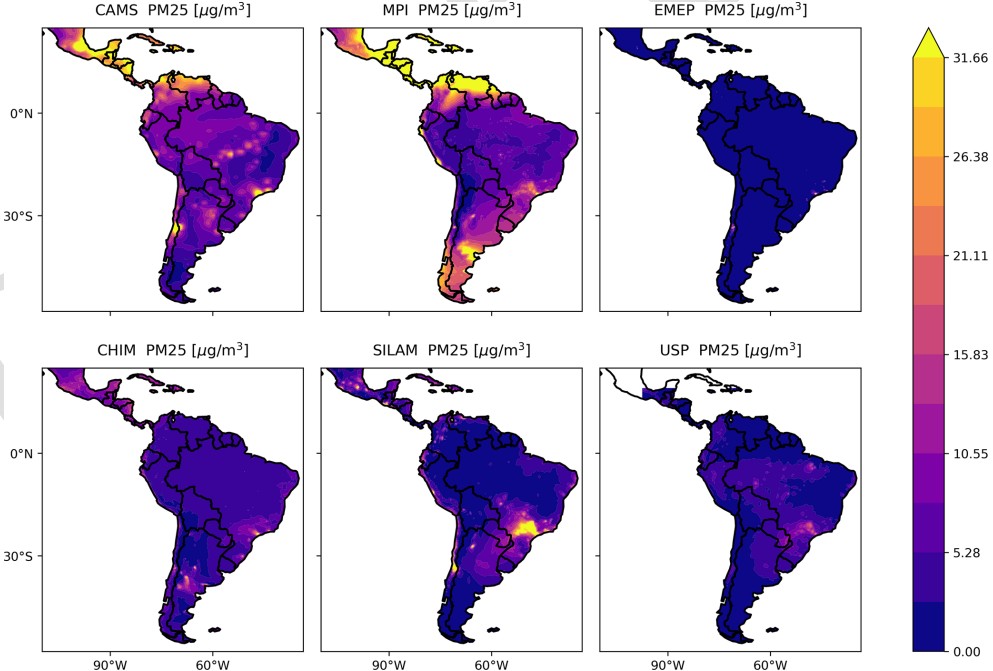

**Figure D10.** PM$_{2.5}$ simulations of July 2015 for all models.

## Appendix E: Model deviations

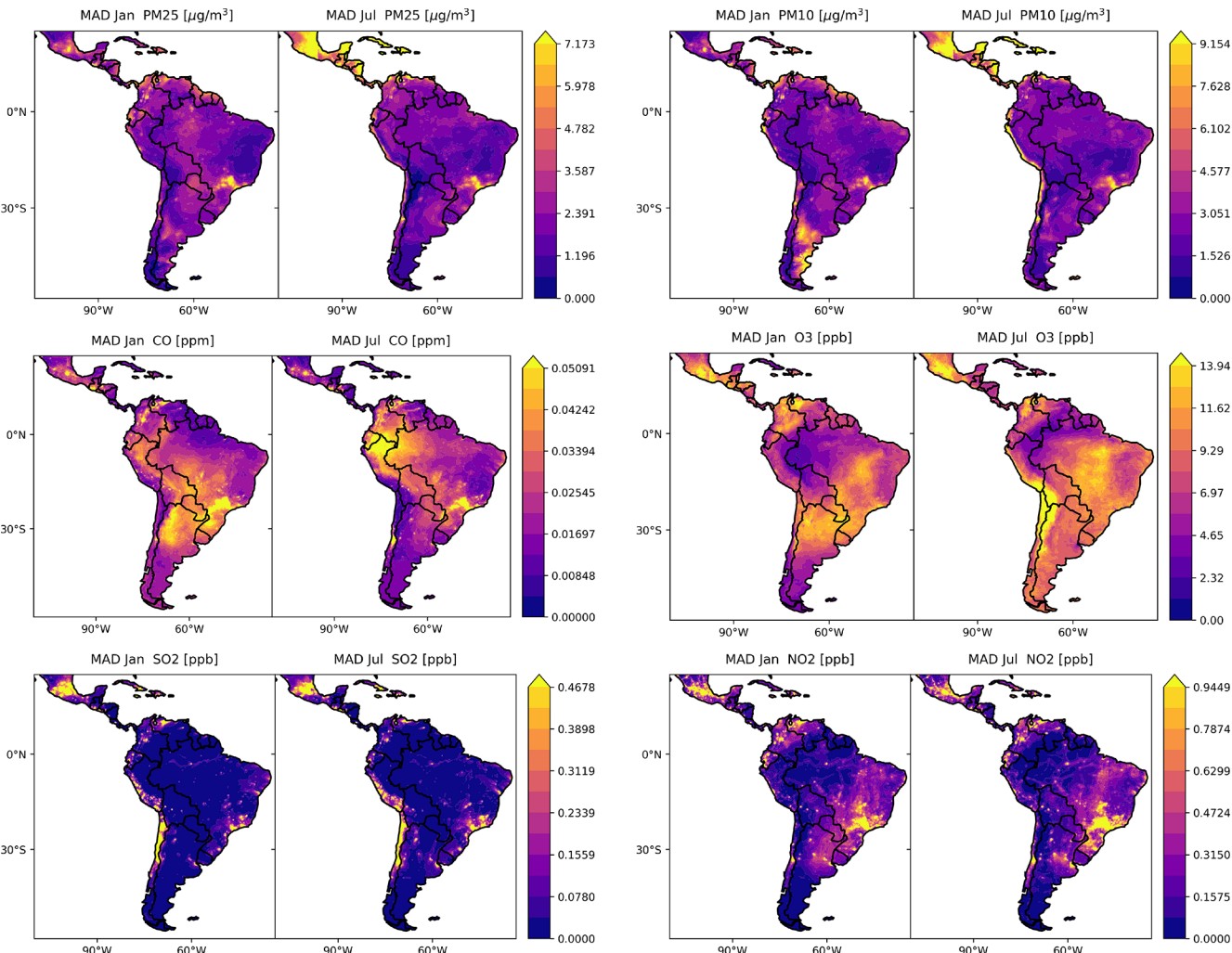

**Figure E1.** Median absolute deviation (MDA) of the models with respect to the ensemble for $PM_{10}$, $PM_{2.5}$, $O_3$, CO, $SO_2$ and $NO_2$ in LAC for January and July 2015.

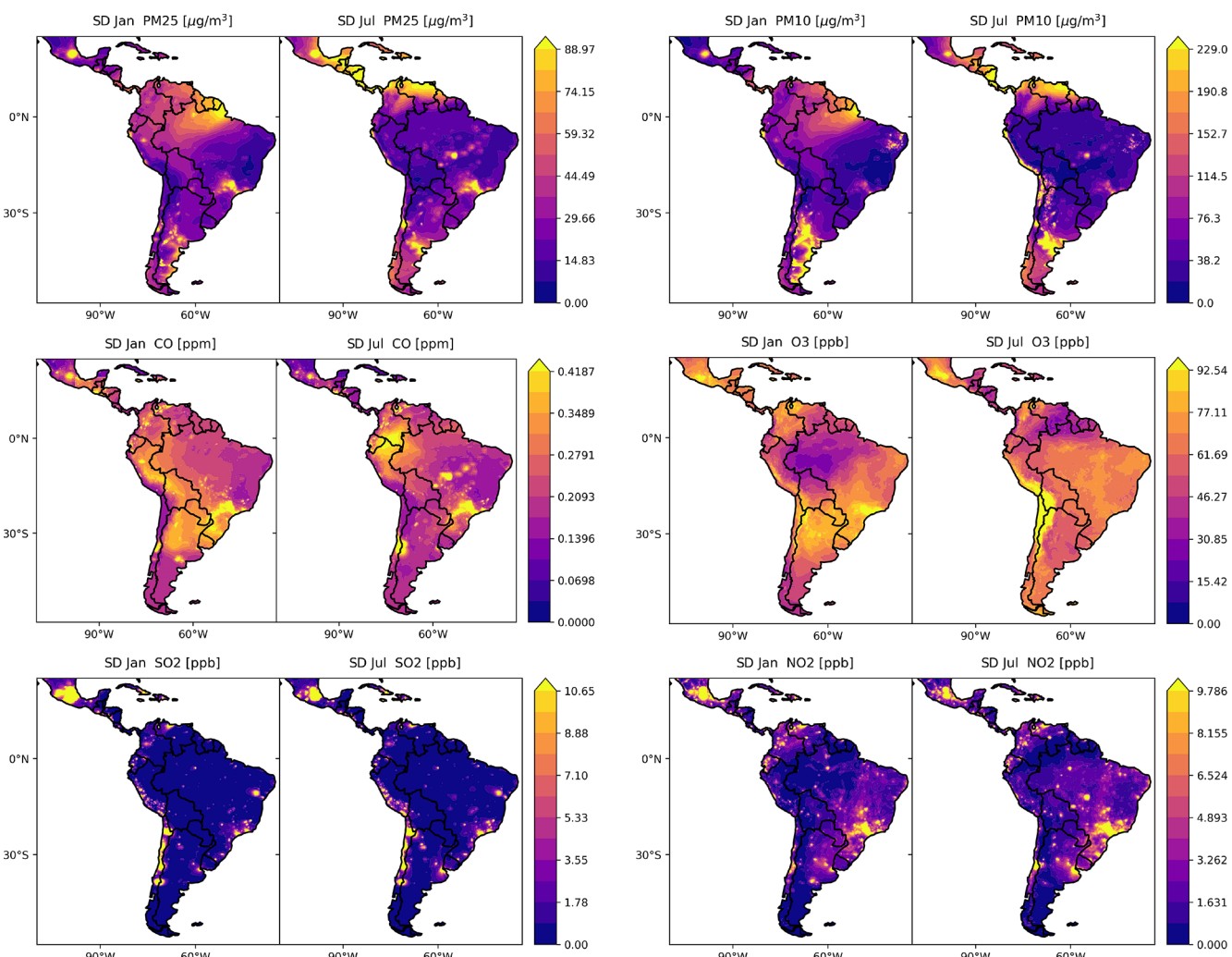

**Figure E2.** Standard deviation (SD) of the models with respect to their mean for $PM_{10}$, $PM_{2.5}$, $O_3$, CO, $SO_2$ and $NO_2$ in LAC for January and July 2015.

*Code and data availability.* All model data analyzed in the intercomparison are archived at https://doi.org/10.5281/zenodo.13151018 (Pachón et al., 2024). The tool to create the plots, MOSPAT, can additionally TS7 be found on GitHub at https://github.com/NeoMOSPAT/NeoMOSPAT_PAPILA.git (Huneeus and Opazo, 2024).

*Author contributions.* JEP and MAO performed the formal analysis of the data; PL, NH, IB, JF, LM, CM, MG, MS, RK, JP, AU, AHDP, MEGC and DS performed the model simulations; JEP, MAO, PL, NH and IB prepared the manuscript with contributions from all co-authors; GB and LG provided the financial support for the project to led to this publication; LD, NYR, NH and MdFA coordinated research activities; CWYL provided technical support; all co-authors reviewed and edited the manuscript.

*Competing interests.* The contact author has declared that none of the authors has any competing interests.

ther geographical representation in this paper. While Copernicus Publications makes every effort to include appropriate place names, the final responsibility lies with the authors.

*Acknowledgements.* The PAPILA (Prediction of Air Pollutants in Latin America) project was funded by the European Commission under the MSCA action for research and innovation staff exchange (grant agreement ID 777544). Support from the Academy of Finland HEATCOST and ACCC flagship projects (grants nos. 334798 and 337552) and the H2020 project AQ-WATCH (grant

no. 870301) is acknowledged. Support from the German Climate Computer Center (Deutsches Klimarechenzentrum, DKRZ) is acknowledged. Support from the Center for Climate and Resilience Research (FONDAP/ANID 1523A0002) is acknowledged.

*Financial support.* This research has been supported by the EU H2020 Marie Skłodowska-Curie Actions (grant no. 777544), Academy of Finland HEATCOST and ACCC flagship projects (grants nos. 334798 and 337552), EU H2020 project AQ-Watch (grant no. 870301), and the Center for Climate and Resilience Research (grant no. FONDAP/ANID 1523A0002).

The article processing charges for this open-access publication were covered by the Max Planck Society.

*Review statement.* This paper was edited by Jason Williams and reviewed by two anonymous referees.

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

**Remarks from the language copy-editor**

CE1    Just to be clear, this is correct as it is and in the short summary?

CE2    Please note the slight adjustments here.

CE3    Please note the slight adjustment to house standard here.

CE4    Please note that a comma was inserted here.

**Remarks from the typesetter**

TS1    Thank you for the information. The name will be corrected in our system automatically once we finalize the production of this article.

TS2    Change inserted, this name will now be abbreviated to "de Fatima Andrade, M." for the citation of the article; please confirm as no further changes can be made once the article has been published.

TS3    Please provide an explanation regarding this correction that can be forwarded by us to the editor. Changes of the scientific content required editor approval.

TS4    Please see comment above regarding editor approval.

TS5    Please see comment regarding editor approval.

TS6    Please see comment regarding editor approval.

TS7    Please confirm.