# Peer review of "Air quality modeling inter-comparison and multi-scale ensemble chain for Latin America"

_EGUsphere, 2024_

## Referee Comment (RC1)

Review of the manuscript

**Air quality modeling intercomparison and multi-scale ensemble chain for Latin America**

Pachon et al., 2024

The submitted manuscript describes an intercomparison of global and regional air quality models operating over Latin America, focusing on the model performance over 4 selected cities. The model results are evaluated against surface measurements of the main air pollutants (NO2, O3, SO2, CO, PM), for each individual model as well as for a model ensemble. The paper in detail describes performance of each model and of the ensemble median by providing the statistical scores for selected pollutants, cities (of different sizes) and seasons (January and July). The study raised up number of interesting points, such as the need for higher model resolution over smaller cities, importance of emission inventories including the local knowledge, the role of wildfires, etc. It also addressed the capabilities and limitations of each of the model and its setup.

As the authors state, this is the first study focusing on the intercomparison and evaluation of the air quality models in this region, and I find this coordinated effort a unique and valuable step forward for the air quality modeling over Latin America.

I recommend the manuscript to be accepted for publication after addressing the following minor comments:

1. Since the paper compares results of different models and their set-ups, I find the Section 2.1 (*Description of the models and modeling set-up*) to be the core part of the manuscript. However, it seems a bit inaccurate or lacking important details. Could the authors please be more specific and include in the paragraph describing each model information on the meteorology driving the model, anthropogenic, biomass burning and biogenic emission inventories, stating the exact name of the emission datasets?
E.g. for SILAM the authors say "the anthropogenic emissions were adopted from the CAMS global emission inventory" (L 74). However, there exist different CAMS inventories and different versions. Also, when the authors say "The biogenic emissions were simulated off-line by the MEGANv2.1 model" (L68, L76, L83), does it mean the MEGAN model runs were performed specifically for this study or did the model use an offline emission inventory calculated by the MEGAN model? Please make clear and if the latter, please specify which biogenic emission datasets were used.
The paragraph describing the WRF-Chem set ups (both for MPIM and USP) is rather brief (L97 – 100). Could the authors be more specific and provide more detail on the emission data used and the difference between MPIM and USP set ups?

   The above mentioned applies also to the summary Table 1. The descriptions seem inaccurate or incomplete.

   - The table is missing vertical resolution for MPIM WRF-Chem and projection for ECMWF-CAMS – please add and if not possible to define, indicate so in the table

- Please define IC-BC abbreviations in the text or in the table footnote
- Please be more specific in description of the emission datasets used and state the name of the emission dataset (including version). E.g. for SILAM the Table states CAMS-REG-AP v3.1 and TNO-MACC which are both regional European inventories. But it is not clear which global anthropogenic dataset was used. Similar for biogenic dataset.

2. South America, esp. the Amazon, is one of the major sources of biogenic VOC emissions globally. I would expect the biogenic VOCs could impact O3 and CO concentrations, esp. in Bogota and Sao Paulo. The paper discusses effect of NO2 on O3, mentions effect of wildfires or excessive OH concentration on CO. But does not mention the possible role of BVOCs. Could the authors please comment on this and where appropriate, include the effect of BVOCs in the discussion? E.g. could the model underestimation of CO be partly explained by possible underestimation of BVOC emissions? The CO January maxima "north of Argentina, south Bolivia, Paraguay and south of Brazil" (L433, Fig. 7) coincide with locations with high isoprene emissions.

Technical comments:

I'd suggest adding a short paragraph at the end of the Introduction section, overviewing the following sections of the manuscript.

L90: please remove *scales* (repetition)

L91: please replace *FINN module* by *FINN dataset*

L115: please replace *Suplhur* by *suplhur*

L116: please add PM10 as well

L133: Please replace *simulate* by *simulated*.

L242: The sentence beginning 'On the other hand' seems incomplete.

L395: Please check the MNBIAS and FGE values in the text. According to the Table A4 these should be 3.6% and 0.1.

L415: Please replace 'hot pollution spots' by 'pollution hot spots'

---

## Author Response (AR2)

Bogotá, August 1st, 2024

Dear Editor
Geoscientific Model Development (GMD)

Regarding the following manuscript:
egusphere-2024-815
Title: Air quality modeling inter-comparison and multi-scale ensemble chain for Latin America
Author(s): Jorge E. Pachon et al.
MS type: Model evaluation paper

Please find below a point-by-point response to the reviewer´s comments. All changes in the revised manuscript has been specified in the response. We sincerely appreciate the reviewer's time and suggestions, they have significantly contributed to enhance the manuscript.

A revised version of the manuscript has been updated in the system, in addition to a marked-up document version showing the changes made. A new author, Andreas Uppstu was included, he provided technical support to reply the reviewers.

Sincerely,

Jorge E. Pachon, Ph.D.
Email: jpachon@unisalle.edu.co
Associate Professor
School of Engineering
Universidad de La Salle

Reviewer 1

*The submitted manuscript describes an intercomparison of global and regional air quality models operating over Latin America, focusing on the model performance over 4 selected cities. The model results are evaluated against surface measurements of the main air pollutants ($NO_2$, $O_3$, $SO_2$, $CO$, PM), for each individual model as well as for a model ensemble. The paper in detail describes performance of each model and of the ensemble median by providing the statistical scores for selected pollutants, cities (of different sizes) and seasons (January and July). The study raised up number of interesting points, such as the need for higher model resolution over smaller cities, importance of emission inventories including the local knowledge, the role of wildfires, etc. It also addressed the capabilities and limitations of each of the model and its setup.*

We thank the reviewer for highlighting important aspects of air quality modeling in Latin America. In fact, this exercise has raised relevant needs, such as improving local emission inventories, elucidating the impact of wildfires and biogenic emissions, achieving higher model resolutions, among others.

*As the authors state, this is the first study focusing on the intercomparison and evaluation of the air quality models in this region, and I find this coordinated effort a unique and valuable step forward for the air quality modeling over Latin America.*

Thanks for the kind appreciation of our work. This study is in fact part of the consolidation of a scientific air quality community in the region with the collaboration of European groups. We are highly motivated with the results of this model inter-comparison effort as a fist stage for the implementation of an Air quality forecasting (AQF) system.

*I recommend the manuscript to be accepted for publication after addressing the following minor comments:*

*1. Since the paper compares results of different models and their set-ups, I find the Section 2.1 (Description of the models and modeling set-up) to be the core part of the manuscript. However, it seems a bit inaccurate or lacking important details. Could the authors please be more specific and include in the paragraph describing each model information on the meteorology driving the model, anthropogenic, biomass burning and biogenic emission inventories, stating the exact name of the emission datasets?*

We thank the suggestion from the reviewer. We carefully revised the description of the models and the modeling set-up. Section 2.1 contains now for each model information on the meteorology driving the model, anthropogenic, biomass burning and biogenic emission inventories, stating the exact name of the emission datasets. More references were added to complement the model description.

*E.g. for SILAM the authors say "the anthropogenic emissions were adopted from the CAMS global emission inventory" (L 74). However, there exist different CAMS inventories and different versions. Also, when the authors say "The biogenic emissions were simulated off-line by the MEGANv2.1 model" (L68, L76, L83), does it mean the MEGAN model runs were performed specifically for this study or did the model use an offline emission inventory calculated by the MEGAN model? Please make clear and if the latter, please specify which biogenic emission datasets were used.*

The reviewer is absolutely right. There exist different CAMS inventories and different versions. For SILAM, anthropogenic emissions were adopted from CAMS-GLOB-ANT v2.1, it was added in the text. With respect to biogenic emissions, only the global models CAMS and SILAM used off-line simulations of MEGAN. CHIMERE uses an on-line approach, so it's the case for EMEP. This situation was clarified in the manuscript. For SILAM, isoprene and monoterpene emissions were computed for the year 2010 as found on the MEGAN website. For CAMS, the biogenic emissions were simulated off-line by the MEGAN model version 2.1 model using an offline emission inventory (ECCAD, 2021).

*The paragraph describing the WRF-Chem set ups (both for MPIM and USP) is rather brief (L97 – 100). Could the authors be more specific and provide more detail on the emission data used and the difference between MPIM and USP set ups?*

We thank the reviewer for this comment. The description of the WRF-Chem setups (both for MPIM and USP) has been expanded including chemical mechanisms, initial and boundary conditions, and emissions datasets for anthropogenic, biogenic and wildfires. Additionally, the difference between MPIM and USP set ups was highlighted.

Lines 155-169 read: "The WRF-Chem is the Weather Research and Forecasting (WRF) model coupled with Chemistry, developed at the National Center for Atmospheric Research (NCAR) with the purpose of simulating urban- to regional-scale fields of trace gasses and particulates. The air quality and meteorological components share the same transport and physics scheme, as well as horizontal and vertical grid (Fast et al., 2006; Grell et al., 2005). The MPIM WRF-Chem uses version 3.6.1 to simulate meteorology and chemistry simultaneously online in South America at ~20 km horizontal resolution and 36 vertical levels extending from the surface to ~21 km altitude. The gas-phase chemistry is represented by the Model for Ozone and Related Chemical Tracers (MOZART-4) chemical scheme (Emmons et al., 2010). The Goddard Chemistry Aerosol Radiation and Transport (GOCART) bulk aerosol module coupled with MOZART is used in this study to consider the aerosol processes (Chin et al., 2002; Ginoux et al., 2001). Boundary and initial conditions for the meteorology were set up from GFS, and for the chemical species concentrations from CAM-Chem. The anthropogenic emissions were from CAMS-GLOB-ANT v4.2, which consists of 0.1° x 0.1° grid maps of several species including CO, SO2, NO, NMVOC, NH3, BC and OC (Granier, 2019). Daily varying emissions of trace species from biomass burning were taken from the (FINN v1.5) dataset (Wiedinmyer et al., 2011). Biogenic emissions of

trace species from terrestrial ecosystems are calculated online using the MEGAN model v2.04 (Guenther et al., 2006). Further details on the MPIM WRF-chem model settings can be found in (Bouarar et al., 2019)."

Lines 170-174 read: "The WRF-Chem run by USP (version 3.9.1) uses similar characteristics as previously described with a horizontal resolution ~22 km and 35 vertical layers. Some differences from the MPIM WRF-Chem configuration are the version of global emissions CAMS-GLOB-ANT v5.3 (ECCAD, 2020), the speciation of the chemical boundary condition from the CAM-Chem model (Buchholz et al., 2019; Emmons et al., 2010) and the speciation of FINN v1.5 emissions which are suitable for simulation over São Paulo."

*The above mentioned applies also to the summary Table 1. The descriptions seem inaccurate or incomplete.*
*- The table is missing vertical resolution for MPIM WRF-Chem and projection for ECMWF-CAMS – please add and if not possible to define, indicate so in the table*
*- Please define IC-BC abbreviations in the text or in the table footnote*

We appreciate this reviewer's comment. Table 1 was completed as follows:
- Vertical resolution was complemented with pressure/height of the lowest and highest layers.
- Number of grid cells (lat/lon) was included.
- Projection for ECMWF-CAMS was added as lat/lon.
- IC-BC abbreviations were defined in the text and populated for all models.
- Emission datasets were augmented including the exact name and versions.

*Please be more specific in description of the emission datasets used and state the name of the emission dataset (including version). E.g. for SILAM the Table states CAMS-REG-AP v3.1 and TNO-MACC which are both regional European inventories. But it is not clear which global anthropogenic dataset was used. Similar for biogenic dataset.*

In fact, there was an error in Table 1. Neither CAMS-REG-AP v3.1 nor TNO-MACC emissions were used in the SILAM simulation. The CAMS global anthropogenic emissions v2.1 were used. It has been corrected. For the rest of the models, emission datasets (including version) have been added to Section 2.1 and Table 1.

*2. South America, esp. the Amazon, is one of the major sources of biogenic VOC emissions globally. I would expect the biogenic VOCs could impact O3 and CO concentrations, esp. in Bogota and Sao Paulo. The paper discusses effect of NO2 on O3, mentions effect of wildfires or excessive OH concentration on CO. But does not mention the possible role of BVOCs. Could the authors please comment on this and where appropriate, include the effect of BVOCs in the discussion? E.g. could the model underestimation of CO be partly explained by possible underestimation of BVOC emissions? The CO January maxima "north of Argentina, south Bolivia, Paraguay and*

*south of Brazil" (L433, Fig. 7) coincide with locations with high isoprene emissions.*

The reviewer makes an excellent point. In fact, the Amazon is the largest rainforest in the world and a significant source of BVOCs. For one part, the oxidation of BVOCs leads to the formation of CO, and for the other, BVOCs and CO are precursors of ozone. Several studies have observed that urban plumes of NOx into the Amazon forest, where BVOCs are abundant, lead to ozone formation (e.g. Kuhn et al., 2010; Nascimento et al., 2022). We complemented the manuscript discussion including the role of BVOCs in the following sections:

Lines 55-57 read: "The Amazon is the largest forest in the world and a significant source of biogenic volatile organic compounds (BVOCs), precursors of CO and secondary ozone (Nascimento et al., 2022; Zimmerman et al., 1988)".

Lines 402-404: "In addition, a major source of atmospheric CO is the oxidation of BVOCs (Worden et al., 2019), which are significantly underestimated in the Southern Hemisphere (Zeng et al, 2015)."

Lines 572-594: "Several studies have shown the influence of urban plumes of $NO_2$ into the Amazon rainforest, rich in BVOCs, with the consequent generation of ozone (Abou Rafee et al., 2017; Nascimiento et al., 2022). In January, simulated $O_3$ concentrations are also large in Mexico City during winter, a situation that has been observed in other studies (Barret and Raga, 2016). There is a maximum of CO in the area between north of Argentina, south of Bolivia, Paraguay and south of Brazil, probably related to fires and the abundance of BVOCs."

Lines 591-593: "In July, CO showed large differences in the Colombian and Peruvian Amazon, mostly driven by the EMEP model. This situation might be related to an incorrect estimation of BVOCs emissions as precursors of CO in forested areas."

*Technical comments:*
*I'd suggest adding a short paragraph at the end of the Introduction section, overviewing the following sections of the manuscript.*

We followed this suggestion from the reviewer. Lines 77-81 now read: "This manuscript presents a retrospective analysis and it's organized as follows: Sect. 2 presents model descriptions, emission inventories utilized in the models, and observations employed for model evaluation. In Sect. 3 we analyze the model performance and conduct inter-comparisons for each pollutant (NO2, O3, CO, SO2, PM2.5). We also discuss the season variability of predictions and the analysis of large vs small urban areas. Finally, Sect. 4 summarizes our findings and outlines directions for future development."

*L90: please remove scales (repetition)*
Thanks, it was removed

*L91: please replace FINN module by FINN dataset*
The sentence has been updated to: "the Fire INventory from NCAR (FINN v1.0)"

*L115: please replace Suplhur by suplhur*
It was replaced
*L116: please add PM10 as well*
It was added

*L133: Please replace simulate by simulated.*
It was replaced

*L242: The sentence beginning 'On the other hand' seems incomplete.*
Thanks, the sentence was rewritten. Lines 370-372 now read: "Additionally, biomass burning from wildfires which begin in July and peak in August and September for the southern part of the Amazon rainforest can bring more CO (Marlier et al., 2020)."

*L395: Please check the MNBIAS and FGE values in the text. According to the Table A4 these should be 3.6% and 0.1.*
The MNBIAS and FGE values, as well as the other metrics, were updated throughout the manuscript due to the following reasons: first, model results and observations were recalculated for 30 days each month (some models were not run on the 31st day of the month) and second, the approach to comparing the model and observations was adjusted to consider the same spatial average across the model and monitors.

*L415: Please replace 'hot pollution spots' by 'pollution hot spots'*
It was replaced

**Reviewer 2**

Summary:
*\* This paper does important work comparing simulations over lesser studied regions.*

Thanks for the kind appreciation of our work. This study of evaluating and comparing different air quality models in Latin America promoted the consolidation of a scientific air quality community in the region with the collaboration of European partners. We are highly motivated with the results of this model inter-comparison effort as a fist stage for the implementation of an Air quality forecasting (AQF) system.

*\* The paper is written as though motivated by forecasting, but the methods seem more focused on historical application and does not provide much discussion of forecasting needs/limitations.*

The motivation of this work and of the scientific community in Latin America is to build an AQF system of air quality, which allows citizens to be adequately informed and reduce the impact on health. To do this, the first step, which this article aims to reflect, is a retrospective (hindcast) model inter-comparison to understand the performance of the models and characterize their errors. Subsequently, the design of the AQF system over the next few days could be implemented.

To consider the reviewer's suggestion, more discussion of forecasting needs/limitations has been included in the conclusions and future development. Lines 621 to 646 now read:

 "This study performed the first inter-comparison and model evaluation effort in Latin America with the idea to develop an AQF system that can inform the public about air pollution episodes and support policy actions. Despite the limitations of air quality and emissions data, as well as computing resources, the scientific community in Latin America, with international support, has achieved significant progress in air quality modeling and in understanding the fate and transport of pollutants in the region. For instance, the impact of Saharan dust, biomass burning from the Orinoco and the Amazon basis, biogenic VOCs of the Amazon rainforest, are becoming better understood through modeling.

Several challenges still exist. In addition to the intricate topography and diverse meteorological conditions, limitations are found in anthropogenic, volcanic and biogenic emissions, spatial and temporal profiles, land use and vegetation types, as well as other data that are relevant for the calculation of wildfire emissions. This last source is crucial in the region under a climate change scenario, for which adequate parametrization of biomass burning is necessary. The boundary conditions of the models can be improved, which are especially important for long-lived species. The experience of local researchers who have been implementing air quality models for several years can greatly benefit international efforts such as global emissions inventories and the recently-launched WMO GAFIS initiative.

At this first stage of development, interesting and insightful findings were identified for the region. Despite the fact that some of the models were still in an early phase for regional implementation, most models could adequately reproduce air quality observations with the best performance observed for nitrogen dioxide in México City and São Paulo. These enormous urban areas (> 3500 km$^2$) outperformed Bogotá and Santiago, which are cities between 500 and 1000 km$^2$. This suggests an accurate portrayal of the temporal and spatial variability in large cities with the current model resolution (0.2° x 0.2°) and the need for a finer model domain in smaller cities that could capture circulation and emission features. At the moment, high-resolution global simulations in the Global South remain rare.

The ensemble median was evaluated on its potential to outperform individual models. In certain periods and cities, the ensemble performed better than any individual models, for example, when the errors of the models compensate for each other, but not when the

errors are recurring in all the models. The results varied per city, pollutant and period. Before defining whether the ensemble is the correct approximation for an AQF system, more research is necessary. This work only looked at two months (one in summer and one in winter), a thorough analysis of one entire annual cycle with sufficient spin-up time should be conducted. More observations should also be included for model calibration and evaluation. For 2015, only eight cities in LAC had data that complied with quality and completeness criteria. In recent years, more AQ networks have been implemented and data is more publicly available."

*The paper has many endpoints and many locations. The current discussion that starts with individual species and all locations was somewhat difficult to read. It would be nice to provide high-level context and specific useful details.*

We acknowledge that the manuscript covers different pollutants in four cities and two periods. This may seem like a lot of information, but as an initial stage of diagnosis we consider it necessary to explore various pollution situations. We have discussed various ways of presenting the information. We believe that the way it is presented is the best possible, because for each pollutant we show the spatial and temporal variability, reflected in four main cities of the region and two periods with different meteorology.

To follow the reviewer´s suggestion, we have provided more context of our work, especially how this effort is the first step towards an air quality forecasting system (AQF) in Latin America. Lines 82-86 now read: "This work conducts the first model inter-comparison effort and ensemble construction for Latin America, which was assembled under the Prediction of Air Pollutants in Latin America (PAPILA) project. The aim of PAPILA was to develop an air quality analysis and forecast system for the region with increasing capabilities in major cities. This objective is in line with the WMO GAFIS initiative that supports the implementation of AQF in countries and regions where they do not exist, such as Africa and South America (WMO, 2022)."

In Section 3.2 Spatial seasonal variability of predictions, the discussion was expanded. Lines 571-585 now read: "During the austral summer, the southeastern part of Brazil (including São Paulo) displays large concentrations of ozone that were simulated mainly by the regional models WRF-Chem, EMEP and the global SILAM (Fig. D3). Several studies have shown the influence of urban plumes of $NO_2$ into the Amazon rainforest, rich in BVOCs, with the consequent generation of ozone (Abou Rafee et al., 2017; Nascimento et al., 2022). In January, simulated $O_3$ concentrations are also large in Mexico City during winter, a situation has been observed in other studies (Barrett and Raga, 2016). There is a maximum of CO in the area between north of Argentina, south of Bolivia, Paraguay and south of Brazil, probably related to fires and the abundance of BVOCs.

In July, during the austral winter, concentrations of CO, PM2.5 and PM10 are significant in Santiago due to transportation and residential heating emissions under adverse meteorological conditions. PM10 concentrations are large in the Caribbean and central

México, primarily due to the transport of Saharan dust into these urban areas (Kramer and Kirtman, 2021; Ramírez-Romero et al., 2021). Similarly, along the Pacific coast between Chile and Peru, increased PM10 is probably explained by anthropogenic emissions of copper smelters in connection with strong eastern wind events (Huneeus et al., 2006). Large concentrations of O3 are visible in México City associated with clear skies under high-pressure atmospheric conditions (Barrett and Raga, 2016). Elevated O3 values in the Andes mountains between northern Chile and central Peru might be explained by the abundance of VOCs from metropolitan regions and industrial zones (Seguel et al., 2024)."

In Section 3.3 Large versus small urban areas, Lines 608-612 now read: "Although the size of cities can influence the performance of the models at coarse resolution, other challenging features for models exist. For instance, Bogotá and Santiago have several challenges in terms of topography and meteorology (Mazzeo et al., 2018b; Nedbor-Gross et al., 2017; Reboredo et al., 2015) and local emissions not always accounted in global inventories (Castesana et al., 2022; Huneeus et al., 2020; Osses et al., 2022; Rojas et al., 2023)."

*Overall, the paper has an impressive scope, but could improve readability/organization/focus.*

We have improved the paper in the following aspects:
Readability: we enhanced the flow of the manuscript; the model evaluation section was reduced to a few metrics and more discussion was included.

Organization: the ensemble description for each pollutant presented in section 3.2 was merged in section 3.1, shortening the manuscript.

Focus: in different parts of the manuscript, we have emphasized the role of this exercise as the first stage for an AQF system in LAC. We have also put in context the need for a local forecast to inform the public about air pollution episodes and support policy actions. This type of effort, especially in regions where they do not exist, is promoted by global initiatives such as the WMO GAFIS (WMO, 2022).

*Though I recommend larger improvements, the article as is represents a substantial effort and could be published with minor updates.*
We thank the reviewer for the recognition of our work. The comments have substantially improved the quality and focus of the paper.

**High level thoughts:**

* 2015 seems like an odd choice for a precursor to a forecast system. Perhaps frame it a bit differently.

We understand the reviewer's point and we have put it in a better context. Our work displays a retrospective analysis of air quality modeling in Latin America with the idea to understand the performance of different models and characterize their errors. In a later stage, the actual forecast could be produced. Lines 25-28 in the abstract now read: "Two global and three regional models were tested and compared in retrospective mode over a shared domain (120W-28W, 60S-30N) for the months of January and July 2015. The objective of this experiment was to understand their performance and characterize their errors."

*Is city size really the determinant factor? Or do Bogota and Santiago have other challenging features for models? While this is interesting, it should be frame as a correlative speculation.*

The reviewer is right, Bogota and Santiago as examples of medium-size urban areas in Latin America, have several challenging features for models, especially in a coarse resolution (0.2 degrees). On one hand, topography and meteorology are complex, Bogota is located at a high altitude surrounded by the Andes mountains where simulation of wind direction has traditionally been a major difficulty (Nedbor-Gross et al., 2017; Reboredo et al., 2015). Santiago is at sea level also surrounded by the Andes cordillera that influences poor circulation and vertical mixing of pollutants, especially during winter (Mazzeo et al., 2018). On the other hand, local emissions sources are abundant and diverse, and not properly accounted for in global inventories (Castesana et al., 2022; Huneeus et al., 2020; Osses et al., 2022; Rojas et al., 2023).

We have mentioned these challenging features in Section 3.3. Lines 610 to 614 now read: "Although the size of cities can influence the performance of the models at coarse resolution, other challenging features for models exist. For instance, Bogota and Santiago have several challenges in terms of topography and meteorology (Mazzeo et al., 2018; Nedbor-Gross et al., 2017; Reboredo et al., 2015) and local emissions not always accounted in global inventories (Castesana et al., 2022; Huneeus et al., 2020; Osses et al., 2022; Rojas et al., 2023)."

*Abstract: what was the suite of endpoints? You say O3+NO2: good; SO2: bad; ?: moderate.*

Section 3.1 expands on the performance of the models per pollutant. $O_3$ and $NO_2$ exhibit the lowest MNBIAS and FGE (Tables A2 and A3) with some models achieving benchmarks, especially in Sao Paulo and Mexico City. For $SO_2$, MNBIAS reach 190% and FGE up to 200% (Table A5) in all cities, except Bogota. To make this clearer, the abstract was rewritten as "$O_3$ and $NO_2$ exhibit the lowest bias and errors, especially in Sao Paulo and Mexico City. For $SO_2$, the bias and error were close to 200%, with exception in Bogota."

*Mediation of outliers doesn't seem like "demonstrating the potential to establish an analysis and forecast system". Was it tractable? Could a single model have performed*

*similarly? If this is setting the stage for future application (ensemble?), I'd like to see more discussion of the application-specific pros/cons.*

In this exercise we observed that the ensemble outperformed individual models in certain cases, for example, when the errors of the models compensate for each other, but not when the errors are recurring in all the models. The results varied per city, pollutant and period. Therefore, we consider that more research is necessary before concluding that the ensemble is the path for an AQF system in Latin America. We thank the reviewer for allowing us to reflect on this topic. Currently, we are conducting a similar model inter-comparison study for a more recent year, where we are further evaluating the ensemble set-up.

Based on the reviewer's comment we have removed the sentence in the abstract and have added the following sentence "The ensemble, created from the median value of the individual models, was evaluated as well. In some cases, the ensemble outperformed individual models and mitigated the extreme over- or underestimation. However, more research is needed before concluding that the ensemble is the path for an AQF system in Latin America".

*\* Did this system out-perform publicly available forecasts from CAMS and GEOS-CF? Since they did not report in 2015, it is hard to say… However, if existing global forecasts are outperforming your ensemble, the goal of forecasting is already solved. I think it would be good to set the stage for the need for a local forecast.*

We understand the reviewer's point. However, much can be learned from local forecasts. From this exercise, we analyzed the performance of global and regional models in Latin America and characterized their errors. We identified needs such as improving emissions in global inventories, developing spatial and temporal profiles, collecting land use and vegetation types, and other data relevant for the calculation of biogenic fluxes and wildfires. High-resolution global models are necessary to resolve the spatial variation in LAC cities, but unfortunately global models at high performance are scarce in the Southern Hemisphere (Zhang et al., 2023).

The model inter-comparison effort also contribute to strengthening the air quality community and enabling a more collaborative platform for drawing together emissions, ambient air quality and meteorology data. In fact, the WMO just launched the GAFIS (Global Air Quality Forecasting and Information System) initiative to promote and enhance air quality forecasts, especially in regions of the world with less experience, such as Africa and South America (WMO, 2022).

To follow the reviewer's suggestion, we set the stage for the need for a local forecast in the manuscript. Lines 45-46 in the introduction read: "Latin America could greatly benefit from an air quality forecasting (AFC) system that inform the public about air pollution episodes

and support policy actions" and lines 83-86 read: "The aim of PAPILA was to develop an air quality analysis and forecast system for the region with increasing capabilities in major cities. This objective is in line with the WMO GAFIS initiative that supports the implementation of AQF in countries and regions where they do not exist (WMO, 2022)."

*Both global models are based on C-IFS meteorology and several regional models too. How does this influence the spread of the ensemble?*

The conceptual approach for this exercise implied that all simulation parameters were left up to the choice of the modeling team, including meteorology and physiography data and input emissions. As a result, some groups used the same meteorology or similar emission databases. However, the models had different configurations (grid cells, projection, IC and BC) and process parameters (e.g., varying chemical mechanisms and reaction rates). The ensemble would also benefit from this variation.

*I suggest including lots of statistics (as you did), but only in the appendix. In the text, the paper would be improved by focusing on a few. Given that you use city averages, you have just 30 points of data so presenting so many statistics seems disproportionate to the data populating them.*

We thank the reviewer for the suggestion. We focused on three statistics in the main document: MNBIAS, FGE and R given that benchmarks exist. The rest of the statistics are still available in Tables A1 though A7.

*Similarly, the per species sections with obs, model, inter-model makes the paper quite long for the value. If the median ensemble was added with the individual models you could reduce the paper length.*

That's a good suggestion, thanks. We merged section 3.2 "Median Ensemble" into section 3.1 "Model evaluation", shortening the manuscript length.

*The comparison of model area mean to monitor mean is not a particularly useful comparison. Monitoring networks are typically located in a spatially biased manner. They tend to be located near high concentrations and near people. As a result, when comparing the observation to the model, only the model is a spatial average. The observation is spatially weighted. A more fair comparison would be to sample the model at observations and then average it. In that way, the model would be spatially weighted in the same way as the observations. Or, you should attempt to remove the monitor location bias. You could do that by averaging the monitors within pixel/polygon intersection. They perform the same weighted average on the observations. In short, right now you are applying meaningfully different spatial averaging on the model and monitors. The more complex method does not seem to address this.*

We fully agree with the reviewer, we were comparing a different spatial averaging on the model and monitors. Therefore, we updated the comparison approach, following the reviewer's suggestion, sampling the model at observations and then averaging it. Lines 185-189 read: "On the other hand, the simulated concentrations for the models were estimated as the average of the models' closest grid point to the location of each station that is within the city's polygon for every city and pollutant considered in this study. This results in a weighted average of the model where the weight is given by the number of stations that measure the pollutant closest to each grid point, resulting in the same geographical sampling for the observations and the models, thus reducing any potential station's sampling bias to the best of our abilities."

We prepared new tables and graphs to assess the changes. In general, results were very similar, but a major change was observed in the performance of the SILAM and EMEP models in Santiago. This is likely due to the number of stations assigned to a model cell with a particularly high concentration of the pollutant. As an example, for $NO_2$ in Santiago, previously all model underestimated this pollutant, SILAM (MNBIAS ~ -15%), EMEP (MNBIAS ~ -50%), and with the new comparison method SILAM overestimates (MNBIAS ~ 37%) and EMEP underestimates with less severity in one period (MNBIAS ~ 7% in Jan / -24.5% in Jul). As a result of this change, the ensemble in Santiago achieved better metrics (MNBIAS -19% Jan / -74% Jul) in comparison to the previous method (MNBIAS -55% Jan / -88% Jul). The FGE and RMSE were also reduced, especially in January. In Bogota, EMEP is closer to observations (MNBIAS ~ -45%) previously -65%. However, the changes in the ensemble in Bogota are negligible. In Sao Paulo, EMEP model further overestimates (MNBIAS ~ 40% Jan / 18% Jul) with respect to the previous estimate (MNBIAS ~ 70% Jan / 44% Jul).

**Dataset Methods:**

*CAMS emissions are described in detail by specific reports. You should consider citing those documents rather than MEGAN generally.*

  *https://eccad.aeris-data.fr/essd-surf-emis-cams-bio/*

  *https://eccad.aeris-data.fr/essd-surf-emis-cams-ant/*

  *https://eccad.aeris-data.fr/essd-surf-emis-cams-soil/*

We thank the reviewer for this suggestion. The citing for the Global anthropogenic emissions was included. However, for the biogenic and soil database we prefer to keep the references provided by the modeling groups.

**Table 1:**

*Global models do have initial conditions (IC), so "global model" is not sufficient. How long were they spun up? From what?*

The reviewer is correct, global models do have initial conditions (IC). For CAMS and SILAM models, IC for meteorology are from the ECMWF´s operational analysis and for chemistry from the previous forecast. Additionally, SILAM has boundary conditions (BC) from CAMS-Global IFS. This information has been augmented in Table 1.

*Vertical structure is important, but perhaps most relevant is the depth of the first layer which directly influences the comparability of the model to the measurement (at a few meters). Also, I recommend being consistent. 25 layers up to what? 60 levels up to what? 35 levels up to what? MPIM vertical structure?*

The information was added in Table 1 and consistency was checked. All models now include the depth of the first layer and highest level (in hPa).

*Recommend adding cell size to the projection cell. Because distortion varies by projection type, this would be useful to understand where and when ~0.2 degree is achieved.*

The outputs from all models were regridded to a shared grid. The number of grids (lat / lon) for each model was included in Table 1. Unfortunately, the cell size is no longer available.

**Large versus small areas:**

*The discussion of city-wide means would benefit from intra-urban variability of model performance. Are large cities (e.g., Mexico) simply averaging high and low biases in the city-wide mean? The large cities also have complex topography/coastal issues that could lead to problems from 0.2 degree resolved models. * You likely have sufficient observations within Mexico and Sao Paulo to say something more than simply observing a difference in means.*

We understand the reviewer's point of view. However, the evaluation of intra-urban variability is beyond the scope of this work. This experiment was designed to evaluate the overall performance of the models in the region, not the models at the monitoring station. The coarse resolution (0.2 degrees) of this effort is not sufficient to explain the variability of pollutants at the city scale. For instance, Bogota and Santiago are represented by four to six cells in the models and Mexico City and Sao Paulo by at least nine cells (Fig. 1 in the manuscript).

**Specific questions:**

*line 118, "modified normalized bias" should probably be "mean normalized bias" as described in Table A1.*

We adopted the definition of "modified normalized bias" from previous studies (Petersen et al., 2019), Equation 3. We corrected MNBIAS definition in Table A1.

*line 126, 75% completeness for the entire dataset? Was there any completeness applied to specific days (e.g., 18 of 24h)? Was there any minimum number of sites per city? Per species?*

We make this information clearer in the text. Lines 204-206 now read: "Only stations with a minimum of 75% data completeness were considered when calculating the city average of the observations, resulting in eight cities with enough data to use for this study. This data completeness requirement considers a minimum of 75% of days available for each period, as well as a minimum of 75% of hourly data to construct their daily average."

*line 128-129, if all the data is used, then the appendix should include data completeness and number of monitors for all cities. I do not see that. Guadalajara, Medellin, Quito, Lima*

We thank the reviewer for this suggestion. Tables B5 through B8 have been included with information from Quito, Medellín, Lima and Guadalajara.

*line 139, "In all cities the data availability was 100%." It sounds like you're saying that all stations in all 8 cities never missed a single hour of measurement for the simulation period. That sounds amazing, but I could be misinterpreting the sentence.*

We apologize for this sentence, there was an error. We updated the availability of pollutants at all cities and included the next footnote in Tables B1 through B8 to explain the meaning of data availability "The observations availability refers to the percentage of days in each period when at least one station records enough data to construct their daily average (minimum of 18 hours). Additionally, only stations that maintain at least 75% of daily availability throughout the entire period are considered (at least 23 days with 18 hours minimum). The model availability refers to the percentage of days for which we have modeled data, being CHIMERE the only one with missing days, and USP missing information for México given their simulation domain did not include it."

*Figure 2 shows considerable missing data from CHIM. I didn't see anything about models being incomplete, so this seems odd. How does this affect the interpretation of statistics from CHIM to other models? After reading the appendices, I believe that missing data is described there.*

We apologize for this omission. Indeed, there are some missing days in the outputs of CHIMERE. We clarified this situation in the manuscript. Lines 138-139 read: "For this exercise, CHIMERE were run for the 31 days of January and July of 2015, however due to

problems in the output files 15 days were missing (5 days from January 14th to 18th and 10 days from July 11th to 19th and July 9th).”

To answer the reviewer's question about the impact on the results, we compare the statistics based on the 30 days of January or July vs the statistics estimated only in the days when CHIMERE outputs were available. Tables A2 to A7 for the CHIMERE period are in the Appendix of this document. There are no major changes in the metrics with respect to the 30-days estimates that would impact on the analysis.

*Figure 2 shows that no model has a prediction on July 31, but two models in the appendix claim 100% data coverage. One of these two things is not true.*

The reviewer is absolutely right. January 31 and July 31 were removed from the analysis, both from models and observations. Tables B1 through B8 now show that all models have 30 days of execution in January and July, with the exception of CHIMERE.

*line 147, [the] NO2 [mean] is underestimated [by all ensemble members]*

This sentence is no longer valid. In the updated comparison method, SILAM overestimates $NO_2$ in Santiago.

*line 149, what was the Bogota value?*

The value was included and updated. Lines 232-233 now read: “Similarly, in Bogotá the mean of the modeled values is 6.6 ppb, much lower than observations.”

*line 149, "the model fields are above and below" do you mean that the ensemble members both over and underpredict?*

We thank the reviewer for this comment, the grammar has been adjusted.

*line 165, does this suggest the correlation is related to meteorology and the magnitude to emissions?*

The reviewer makes an excellent point. However, we think that our work is too preliminary to support this conclusion. For example, the meteorology from the IFS model was utilized by SILAM, CAMS, EMEP and CHIMERE and these models do not always display a similar correlation with observations. The same situation is observed with the WRF-Chem implemented by MPI-M and USP and driven by the same meteorology. On the other hand, the same CAMS emission inventory was used by SILAM and CAMS models with large differences in the magnitude of some of the simulated pollutants. To answer the reviewer's questions, we would need to optimize the configuration of the models, improve model inputs and conduct a thorough analysis of one entire annual cycle with sufficient spin-up time.

*line 186, Please rewrite this sentence. Dispersion is used for ventilation*

We thank the reviewer. The term dispersion has been replaced by variation throughout the manuscript.

*Table A1: I wonder if Coefficient of Variation has a typo. I am used to this being the std dev divided by the mean, which given a constant mean is larger when the variance is a larger. Because your table does not report standard deviation and the description does not either, I can't check what the results are.*

We apologize for this typo. The Coefficient of Variation has the traditional definition of std dev divided by the mean. The notation has been corrected in Table A1.

*line 189-191, does dispersion mean the variation between ensemble members? I'd recommend not using the word dispersion because air quality scientists use this word to describe ventilation.*

We thank the reviewer for this suggestion. The term dispersion has been replaced by variation throughout the manuscript.

*line 194, the number of sites with obs should be added to figure 1 using different shapes for ozone, no2, co, so2, and pm. It seems odd that a reader would have to go to the appendix to know. In the text, at least provide a range for the four cities that are the focus of the paper. Is it between 1 and 10?*

We thank the reviewer for this suggestion. Figure 1 has been updated showing the number of sites per pollutant (using different layers). The range of sites per pollutant for the four cities was added in the text. For example, for $O_3$, lines 300-301 now read: "The number of stations per city recording $O_3$ during January and July of 2015 varies between 9 in Santiago and 29 in Mexico City (Appendix B)"
*line 196-197, ozone season for Mexico? Are all the areas the same?*

The reviewer is right. The ozone season in Mexico City corresponds to the dry-hot months of March, April and May. July is indeed outside the spring season but rather part of the wet summer months. However, elevated ozone concentrations are also recorded in Mexico City in summer when clear skies affect the basin (Barrett & Raga, 2016). A sentence clarifying the situation was included in the manuscript. Lines 801-803 now read: "The highest observed ozone concentration was in México City in July with an average of 31 ppb. However, this value is significantly lower than the surface ozone concentrations reported in the MAM (March-April-May) season with values larger than 70 ppb (Barrett and Raga, 2016; Silva-Quiroz et al., 2019)"

*Figure 3, I recommend using a common y-minimum (e.g., 0 ppb)*

Figures 2 through 6 were adjusted to include the y-minimum at 0 ppb (0 ug/m3 for PM2.5 in Fig. 6).

*line 207, is that high or low for ozone in Bogota? I shouldn't have to check the figure to understand the text and vice versa.*

The concentration is typical for Bogotá. We make this sentence clearer in Line 315: "Similarly, in Bogotá, models estimate an average of 17 ppb which is in the same order of magnitude as the observations".

*line 386, which one?*

The model was CHIMERE. It has been added to the manuscript.

**References**

Barrett, B. S., & Raga, G. B. (2016). Variability of winter and summer surface ozone in Mexico City on the intraseasonal timescale. *Atmos. Chem. Phys.*, *16*(23), 15359–15370. https://doi.org/10.5194/acp-16-15359-2016

Castesana, P., Diaz Resquin, M., Huneeus, N., Puliafito, E., Darras, S., Gómez, D., Granier, C., Osses Alvarado, M., Rojas, N., & Dawidowski, L. (2022). PAPILA dataset: a regional emission inventory of reactive gases for South America based on the combination of local and global information. *Earth System Science Data*, *14*(1), 271–293. https://doi.org/10.5194/essd-14-271-2022

ECCAD. (2021). *Global Biogenic VOC emissions CAMS-GLOB-BIO v3.0 Snapshot*. https://doi.org/10.24380/xs64-gj42

Huneeus, N., Denier van der Gon, H., Castesana, P., Menares, C., Granier, C., Granier, L., Alonso, M., de Fatima Andrade, M., Dawidowski, L., Gallardo, L., Gomez, D., Klimont, Z., Janssens-Maenhout, G., Osses, M., Puliafito, E., Rojas, N., Ccoyllo, O. S.-, Tolvett, S., & Ynoue, R. Y. (2020). Evaluation of anthropogenic air pollutant emission inventories for South America at national and city scale. *Atmospheric Environment*, 117606. https://doi.org/https://doi.org/10.1016/j.atmosenv.2020.117606

Mazzeo, A., Huneeus, N., Ordoñez, C., Orfanoz-Cheuquelaf, A., Menut, L., Mailler, S., Valari, M., Denier van der Gon, H., Gallardo, L., Muñoz, R., Donoso, R., Galleguillos, M., Osses, M., & Tolvett, S. (2018). Impact of residential combustion and transport

emissions on air pollution in Santiago during winter. *Atmospheric Environment*, *190*, 195–208. https://doi.org/https://doi.org/10.1016/j.atmosenv.2018.06.043

Nedbor-Gross, R., Henderson, B. H., Davis, J. R., Pachón, J. E., Rincón, A., Guerrero, O. J., & Grajales, F. (2017). Comparing Standard to Feature-Based Meteorological Model Evaluation Techniques in Bogotá, Colombia. *Journal of Applied Meteorology and Climatology*. https://doi.org/10.1175/jamc-d-16-0058.1

Osses, M., Rojas, N., Ibarra, C., Valdebenito, V., Laengle, I., Pantoja, N., Osses, D., Basoa, K., Tolvett, S., Huneeus, N., Gallardo, L., & Gómez, B. (2022). High-resolution spatial-distribution maps of road transport exhaust emissions in Chile, 1990--2020. *Earth System Science Data*, *14*(3), 1359–1376. https://doi.org/10.5194/essd-14-1359-2022

Petersen, A. K., Brasseur, G. P., Bouarar, I., Flemming, J., Gauss, M., Jiang, F., Kouznetsov, R., Kranenburg, R., Mijling, B., Peuch, V.-H., Pommier, M., Segers, A., Sofiev, M., Timmermans, R., van der A, R., Walters, S., Xie, Y., Xu, J., & Zhou, G. (2019). Ensemble forecasts of air quality in eastern China – Part 2: Evaluation of the MarcoPolo–Panda prediction system, version 1. *Geoscientific Model Development*, *12*(3), 1241–1266. https://doi.org/10.5194/gmd-12-1241-2019

Reboredo, B., Arasa, R., & Codina, B. (2015). Evaluating Sensitivity to Different Options and Parameterizations of a Coupled Air Quality Modelling System over Bogotá, Colombia. Part I: WRF Model Configuration. *Open Journal of Air Pollution*, *4*, 47–64. https://doi.org/10.4236/ojap.2015.42006.

Rojas, N. Y., Mangones, S. C., Osses, M., Granier, C., Laengle, I., Alfonso A., J. V, & Mendez, J. A. (2023). Road transport exhaust emissions in Colombia. 1990–2020 trends and spatial disaggregation. *Transportation Research Part D: Transport and Environment*, *121*, 103780. https://doi.org/https://doi.org/10.1016/j.trd.2023.103780

WMO. (2022). *Global Air Quality Forecasting and Information System (GAFIS)*. https://community.wmo.int/en/activity-areas/gaw/science-for-services/gafis